# GENERALIZATION GUARANTEES FOR NEURAL NETS VIA HARNESSING THE LOW-RANKNESS OF JACOBIAN

## ABSTRACT

Modern neural network architectures often generalize well despite containing many more parameters than the size of the training dataset. This paper explores the generalization capabilities of neural networks trained via gradient descent. We develop a data-dependent optimization and generalization theory which leverages the low-rank structure of the Jacobian matrix associated with the network. Our results help demystify why training and generalization is easier on clean and structured datasets and harder on noisy and unstructured datasets as well as how the network size affects the evolution of the train and test errors during training. Specifically, we use a control knob to split the Jacobian spectrum into "information" and "nuisance" spaces associated with the large and small singular values. We show that over the information space learning is fast and one can quickly train a model with zero training loss that can also generalize well. Over the nuisance space training is slower and early stopping can help with generalization at the expense of some bias. We also show that the overall generalization capability of the network is controlled by how well the labels are aligned with the information space. A key feature of our results is that even constant width neural nets can provably generalize for sufficiently nice datasets. We conduct various numerical experiments on deep networks that corroborate our theoretical findings and demonstrate that: (i) the Jacobian of typical neural networks exhibit low-rank structure with a few large singular values and many small ones leading to a low-dimensional information space, (ii) over the information space learning is fast and most of the labels falls on this space, and (iii) label noise falls on the nuisance space and impedes optimization/generalization.

## 1 INTRODUCTION

### 1.1 MOTIVATION AND CONTRIBUTIONS

Deep neural networks (DNN) are ubiquitous in a growing number of domains ranging from computer vision to healthcare. State-of-the-art DNN models are typically overparameterized and contain more parameters than the size of the training dataset. It is well understood that in this overparameterized regime, DNNs are highly expressive and have the capacity to (over)fit arbitrary training datasets including pure noise Zhang et al. (2016). Mysteriously however neural network models trained via simple algorithms such as (stochastic) gradient descent continue to predict well or *generalize* on yet unseen test data. In this paper we wish to take a step towards demystifying this phenomenon and help explain why neural nets can overfit to noise yet have the ability to generalize when real data sets are used for training. In particular we explore the generalization dynamics of neural nets trained via gradient descent. Using the Jacobian mapping associated with the neural network we characterize directions where learning is fast and generalizable versus directions where learning is slow and leads to overfitting. The main contributions of this work are as follows.

• **Leveraging dataset structure:** We develop new optimization and generalization results that can harness the low-rank representation of semantically meaningful datasets via the Jacobian mapping of the neural net. This sheds light as to why training and generalization is easier using datasets where the features and labels are semantically linked versus others where there is no meaningful relationship between the features and labels (even when the same network is used for training).

• **Bias–variance tradeoffs:** We develop a generalization theory based on the Jacobian which decouples the learning process into *information* and *nuisance* spaces. We show that gradient descent almost perfectly interpolates the data over the information space (incurring only a small bias). In contrast, optimization over the nuisance space is slow and results in overfitting due to higher variance.

• **Network size vs prediction bias:** We obtain data-dependent tradeoffs between the network size and prediction bias. Specifically, we show that larger networks result in smaller prediction bias, but small networks can still generalize well, especially when the dataset is sufficiently structured, but typically incur a larger bias. This compares favorably with recent literature on optimization and generalization of neural networks Jacot et al. (2018); Arora et al. (2019); Du et al. (2018b); Allen-Zhu et al. (2018b); Cao & Gu (2019); Ma et al. (2019); Allen-Zhu et al. (2018a); Brutzkus et al. (2017) where guarantees only hold for very wide networks with the width of the network growing inversely proportional to the class margins or related notions. See Section 3 for further detail.

• **Pretrained models:** Our framework does not require random initialization and our results continue to apply even with arbitrary initialization. Therefore, our results may shed light on the generalization capabilities of networks initialized with pre-trained models commonly used in meta/transfer learning. Our extensive experiments strongly suggest Jacobian adapts over time in a favorable and data-dependent fashion shedding light on the properties of (pre)trained models.

## 1.2 MODEL AND TRAINING

Our theoretical analysis will focus on neural networks consisting of one hidden layer with $d$ input features, $k$ hidden neurons and $K$ outputs as depicted in Figure 1. We use $\boldsymbol{W} \in \mathbb{R}^{k \times d}$ and $\boldsymbol{V} \in \mathbb{R}^{K \times k}$ to denote the input-to-hidden and hidden-to-output weights. The overall input-output relationship of the network is a function $f(\cdot; \boldsymbol{W}) : \mathbb{R}^d \to \mathbb{R}^K$ that maps an input $\boldsymbol{x} \in \mathbb{R}^d$ to an output via

$$\boldsymbol{x} \mapsto f(\boldsymbol{x}; \boldsymbol{W}) := \boldsymbol{V}\phi(\boldsymbol{W}\boldsymbol{x}). \qquad (1.1)$$

Given a dataset consisting of $n$ feature/label pairs $(\boldsymbol{x}_i, \boldsymbol{y}_i)$ with $\boldsymbol{x}_i \in \mathbb{R}^d$ representing the features and $\boldsymbol{y}_i \in \mathbb{R}^K$ the associated labels representing one of $K$ classes with one-hot encoding (i.e. $\boldsymbol{y}_i \in \{\boldsymbol{e}_1, \boldsymbol{e}_2, \ldots, \boldsymbol{e}_K\}$ where $\boldsymbol{e}_\ell \in \mathbb{R}^K$ has all zero entries except for the $\ell$th entry which is equal to one).

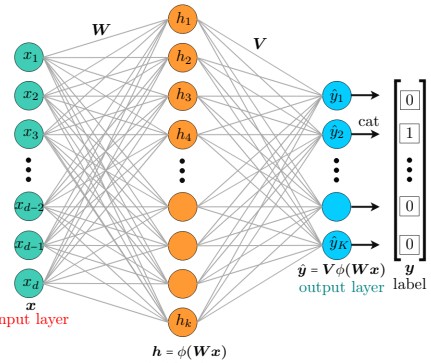

Figure 1: Illustration of a one-hidden layer neural net with $d$ inputs, $k$ hidden units and $K$ outputs along with a one-hot encoded label.

To learn this dataset, we fix the output layer and train over $\boldsymbol{W}$ via[1]

$$\min_{\boldsymbol{W} \in \mathbb{R}^{k \times d}} \mathcal{L}(\boldsymbol{W}) := \frac{1}{2} \sum_{i=1}^{n} \|\boldsymbol{V}\phi(\boldsymbol{W}\boldsymbol{x}_i) - \boldsymbol{y}_i\|_{\ell_2}^2. \qquad (1.2)$$

It will be convenient to concatenate the labels and prediction vectors as follows

$$\boldsymbol{y} = \begin{bmatrix} \boldsymbol{y}_1 \\ \vdots \\ \boldsymbol{y}_n \end{bmatrix} \in \mathbb{R}^{nK} \quad \text{and} \quad f(\boldsymbol{W}) = \begin{bmatrix} \boldsymbol{V} f(\boldsymbol{x}_1; \boldsymbol{W}) \\ \vdots \\ \boldsymbol{V} f(\boldsymbol{x}_n; \boldsymbol{W}) \end{bmatrix} \in \mathbb{R}^{nK}. \qquad (1.3)$$

Using this shorthand we can rewrite the loss (1.2) as

$$\min_{\boldsymbol{W} \in \mathbb{R}^{k \times d}} \mathcal{L}(\boldsymbol{W}) := \frac{1}{2} \|f(\boldsymbol{W}) - \boldsymbol{y}\|_{\ell_2}^2. \qquad (1.4)$$

To optimize this loss starting from an initialization $\boldsymbol{W}_0$ we run gradient descent iterations of the form

$$\boldsymbol{W}_{\tau+1} = \boldsymbol{W}_\tau - \eta \nabla \mathcal{L}(\boldsymbol{W}_\tau), \qquad (1.5)$$

with a step size $\eta$. In this paper we wish to explore the theoretical properties of the model found by such iterative updates with an emphasis on the generalization ability.

## 1.3 INFORMATION AND NUISANCE SPACES

In order to understand the generalization capabilities of models trained via gradient descent we need to develop better insights into the form of the gradient updates and how it affects the training dynamics. To this aim let us aggregate the weights at each iteration into one large vector $\boldsymbol{w}_\tau := \text{vect}(\boldsymbol{W}_\tau) \in \mathbb{R}^{kd}$, define the misfit/residual vector $\boldsymbol{r}(\boldsymbol{w}) := f(\boldsymbol{w}) - \boldsymbol{y}$ and note that the gradient updates take the form

$$\boldsymbol{w}_{\tau+1} = \boldsymbol{w}_\tau - \eta \nabla \mathcal{L}(\boldsymbol{w}_\tau) \quad \text{where} \quad \nabla \mathcal{L}(\boldsymbol{w}) = \nabla \mathcal{L}(\boldsymbol{w}) = \mathcal{J}(\boldsymbol{w})^T \boldsymbol{r}(\boldsymbol{w}).$$

---

[1]For clarity of exposition, we focus only on optimizing the input layer. However, as shown in the supplementary, the technical approach is quite general and applies to arbitrary multiclass nonlinear least-squares problems. In particular, the proofs are stated so as to apply to one-hidden layer networks where both layers are trained.

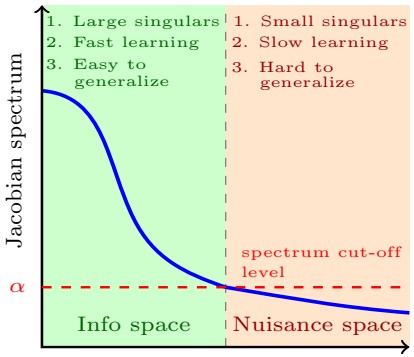

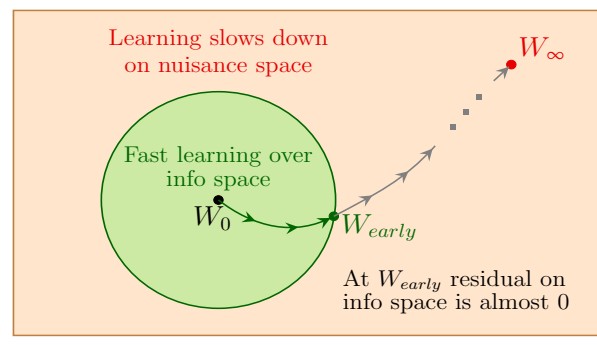

(a) Depiction via the Jacobian spectrum

(b) Depiction in parameter space

Figure 2: Depiction of the training and generalization dynamics of gradient methods based on the information and nuisance spaces associated with the neural net Jacobian.

Here, $\mathcal{J}(\boldsymbol{w}) \in \mathbb{R}^{nK \times kd}$ denotes the Jacobian mapping associated with $f$ defined as $\mathcal{J}(\boldsymbol{w}) = \frac{\partial f(\boldsymbol{w})}{\partial \boldsymbol{w}}$. Due to the form of the gradient updates the dynamics of training is dictated by the spectrum of the Jacobian matrix as well as the interaction between the residual vector and the Jacobian. If the residual vector is very well aligned with the singular vectors associated with the top singular values of $\mathcal{J}(\boldsymbol{w})$, the gradient update significantly reduces the misfit allowing substantial reduction in the train error. Thus to provide a more precise understanding of the training dynamics and generalization capabilities of neural networks it is crucial to develop a better understanding of the interaction between the Jacobian and the misfit and label vectors. To capture these interactions we require a few definitions.

**Definition 1.1 (Information & Nuisance Spaces)** *Consider a matrix $\boldsymbol{J} \in \mathbb{R}^{nK \times p}$ with singular value decomposition given by*

$$\boldsymbol{J} = \sum_{s=1}^{nK} \lambda_s \boldsymbol{u}_s \boldsymbol{v}_s^T = \boldsymbol{U} diag\left(\lambda_1, \lambda_2, \ldots, \lambda_{nK}\right) \boldsymbol{V}^T,$$

*with $\lambda_1 \geq \lambda_2 \geq \ldots \geq \lambda_{nK}$ denoting the singular values of $\boldsymbol{J}$ in decreasing order and $\{\boldsymbol{u}_s\}_{s=1}^{nK} \in \mathbb{R}^{nK}$ and $\{\boldsymbol{v}_s\}_{s=1}^{nK} \in \mathbb{R}^p$ the corresponding left and right singular vectors forming the orthonormal basis matrices $\boldsymbol{U} \in \mathbb{R}^{nK \times nK}$ and $\boldsymbol{V} \in \mathbb{R}^{p \times nK}$. For a spectrum cutoff $\alpha$ obeying $0 \leq \alpha \leq \lambda_1$ let $r := r(\alpha)$ denote the index of the smallest singular value above $\alpha$. We define the information and nuisance spaces associated with $\boldsymbol{J}$ as $\mathcal{I} := span(\{\boldsymbol{u}_s\}_{s=1}^r)$ and $\mathcal{N} := span(\{\boldsymbol{u}_s\}_{s=r+1}^{Kn})$.*

In this paper we shall use either the expected value of the Jacobian at the random initialization or the Jacobian at one of the iterates to define the matrix $\boldsymbol{J}$ and the corresponding information/nuisance spaces. More, specifically we will set $\boldsymbol{J}$ to either $\boldsymbol{J} = \left(\mathbb{E}[\mathcal{J}(\boldsymbol{W}_0)\mathcal{J}^T(\boldsymbol{W}_0)]\right)^{1/2}$ or $\boldsymbol{J} = \mathcal{J}(\boldsymbol{W}_\tau)$. Therefore, one can effectively think of the information space as the span of the prominent singular vectors of the Jacobian and the nuisance space as its complement. In particular, as we demonstrate in Section 4 the Jacobian mapping associated with neural networks exhibit low-rank structure with a few large singular values and many small ones leading to natural choices for the cut-off value $\alpha$ as well as the information and nuisance spaces. Furthermore, we demonstrate both (empirically and theoretically) that learning is fast over the information space leading to a significant reduction in both train/test accuracy in the early stages of training. However, after a certain number of iterations learning shifts to the nuisance space and reduction in the training error significantly slows down (see Fig. 2). Furthermore, subsequent iterations in this stage lead to a slight increase in test error.

## 2 MAIN RESULTS

Our main results establish multi-class generalization bounds for neural networks trained via gradient descent. First, we will focus on networks where both layers are randomly initialized. Next we will provide guarantees for arbitrary initialization with the goal of characterizing the generalization ability of subsequent iterative updates for a given (possibly pre-trained) network in terms of its Jacobian mapping. In this paper we focus on activations $\phi$ which are smooth and have bounded first and

second order derivatives. This would for instance apply to the softplus activation $\phi(z) = \log(1 + e^z)$. We note that utilizing a proof technique developed in Oymak & Soltanolkotabi (2019) for going from smooth to ReLU activations it is possible to extend our results to ReLU activations with proper modifications. We avoid doing this in the current paper for clarity of exposition. Before we begin discussing our main results we discuss some notation used throughout the paper. For a matrix $X \in \mathbb{R}^{n \times d}$ we use $s_{\min}(X)$ and $s_{\max}(X) = \|X\|$ to denote the minimum and maximum singular value of $X$. For two matrices $A$ and $B$ we use $A \odot B$ and $A \otimes B$ to denote their Hadamard and Kronecker products, respectively. For a PSD matrix $A \in \mathbb{R}^{n \times n}$ with eigenvalue decomposition $A = \sum_{i=1}^{n} \lambda_i u_i u_i^T$, the square root matrix is defined as $A^{1/2} := \sum_{i=1}^{n} \sqrt{\lambda_i} u_i u_i^T$. We also use $A^\dagger$ to denote the pseudo-inverse of $A$. In this paper we mostly focus on label vectors $y$ which are one-hot encoded i.e. all entries are zero except one of them. For a subspace $\mathcal{S} \subset \mathbb{R}^n$ and point $x \in \mathbb{R}^n$, $\Pi_{\mathcal{S}}(x)$ denotes the projection of $x$ onto $\mathcal{S}$. Finally, before stating our results we need to provide a quantifiable measure of performance for a trained model. Given a sample $(x, y) \in \mathbb{R}^d \times \mathbb{R}^K$ from a distribution $\mathcal{D}$, the classification error of the network $W$ with respect to $\mathcal{D}$ is defined as

$$\text{Err}_{\mathcal{D}}(W) = \mathbb{P}\Big\{ \arg \max_{1 \leq \ell \leq K} y_\ell \neq \arg \max_{1 \leq \ell \leq K} f_\ell(x; W) \Big\}. \tag{2.1}$$

## 2.1 Results for random initialization

To explore the generalization of randomly initialized networks, we utilize the neural tangent kernel.

**Definition 2.1 (Multiclass Neural Tangent Kernel (M-NTK) Jacot et al. (2018))** *Let $w \in \mathbb{R}^d$ be a vector with $\mathcal{N}(0, I_d)$ distribution. Consider a set of $n$ input data points $x_1, x_2, \ldots, x_n \in \mathbb{R}^d$ aggregated into the rows of a data matrix $X \in \mathbb{R}^{n \times d}$. Associated to the activation $\phi$ and the input data matrix $X$ we define the multiclass kernel matrix as*

$$\Sigma(X) := I_K \otimes \mathbb{E}\Big[ \Big( \phi'(Xw) \phi'(Xw)^T \Big) \odot \big( XX^T \big) \Big],$$

*where $I_K$ is the identity matrix of size $K$. Here, the $\ell$ th diagonal block of $\Sigma(X)$ corresponds to the kernel matrix associated with the $\ell$ th network output for $1 \leq \ell \leq K$. This kernel is intimately related to the multiclass Jacobian mapping. In particular, suppose the initial input weights $W_0$ are distributed i.i.d. $\mathcal{N}(0, 1)$ and the output layer $V$ has i.i.d. zero-mean entries with $\nu^2 / K$ variance. Then $\mathbb{E}[\mathcal{J}(W_0) \mathcal{J}(W_0)^T] = \nu^2 \Sigma(X)$. We use the square root of this multiclass kernel matrix (i.e. $\Sigma(X)^{1/2}$) to define the information and nuisance spaces for our random initialization result.*

The following theorem is a (non-rigorous) simplification of our main result Theorem 6.24 where we ignore constants and log factors, and state a weaker but simpler generalization bound.

**Theorem 2.2** *Fix numbers $\Gamma \geq 1$ and $\alpha > 0$. Consider an i.i.d. training dataset $\{(x_i, y_i)\}_{i=1}^n \in \mathbb{R}^d \times \mathbb{R}^K$ with unit length input samples and one-hot encoded labels. Consider the neural net in (1.1) parameterized by $W$ and initialized with $W_0 \overset{i.i.d.}{\sim} \mathcal{N}(0, 1)$ entries. Set $V$ with i.i.d. Rademacher entries (properly scaled). Define the information $\mathcal{I}$ and nuisance $\mathcal{N}$ spaces with respect to $\Sigma(X)^{1/2}$ with spectrum cutoff $\alpha \sqrt{nK}$ per Definition 1.1. Furthermore, assume*

$$k \gtrsim \frac{\Gamma^4 \log n}{\alpha^8}. \tag{2.2}$$

*Then after $T \propto \Gamma / \alpha^2$ gradient iterations of (1.5), with high probability, training loss obeys*

$$\|f(W_T) - y\|_{\ell_2} \lesssim \|\Pi_{\mathcal{N}}(y)\|_{\ell_2} + e^{-\Gamma} \sqrt{n}, . \tag{2.3}$$

*Furthermore, the classification error obeys*

$$\text{Err}_{\mathcal{D}}(W_T) \lesssim \frac{\|\Pi_{\mathcal{N}}(y)\|_{\ell_2}}{\sqrt{n}} + e^{-\Gamma} + \frac{\Gamma}{\alpha \sqrt{n}}.$$

This theorem shows that even networks of moderate width can achieve a small generalization error if (1) the data has low-dimensional representation i.e. the kernel is approximately low-rank and (2) the inputs and labels are semantically-linked i.e. the label vector $y$ mostly lies on the information space.

• **Generalization bound:** The generalization error has two core components: bias and variance. The bias component $\|\Pi_{\mathcal{N}}(\boldsymbol{y})\|_{\ell_2}/\sqrt{n} + \mathrm{e}^{-\Gamma}$ arises from the training loss and corresponds to the portion of the labels that falls over the nuisance space. The variance component $\Gamma/\alpha\sqrt{n}$ corresponds to the Rademacher complexity of the model space which connects to the distance $\|\boldsymbol{W}_T - \boldsymbol{W}_0\|_F$.

If $\boldsymbol{y}$ is aligned with the information space, the bias term $\Pi_{\mathcal{N}}(\boldsymbol{y})$ will be small. Additionally, if the kernel matrix is low-rank, we can pick a large $\alpha$ to ensure small variance as well as small network width. In particular with a constant $\alpha$ the **required network width is logarithmic in $n$.**

We note however that our results continue to apply even when the kernel is not approximately low-rank. In particular, consider the extreme case where we select $\alpha\sqrt{nK} = \sqrt{\lambda} \coloneqq \sqrt{\lambda_{\min}\left(\boldsymbol{\Sigma}(\boldsymbol{X})\right)}$. This sets $\mathcal{I} = \mathbb{R}^{Kn}$ and $\|\Pi_{\mathcal{N}}(\boldsymbol{y})\|_{\ell_2} = 0$. For this case, the more general Theorem 6.24 yields

$$\mathrm{Err}_{\mathcal{D}}(\boldsymbol{W}_T) \lesssim \frac{\sqrt{K}}{\sqrt{n}}\sqrt{\boldsymbol{y}^T\boldsymbol{\Sigma}^{-1}(\boldsymbol{X})\boldsymbol{y}} \quad \text{while requiring a width of} \quad k \gtrsim \frac{K^4 n^4 \log n}{\lambda^4}. \tag{2.4}$$

We note that in this special case our results improve upon the required width in recent literature Arora et al. (2019)[2] that focuses on $K = 1$ and a conclusion of the form (2.4). However, as we demonstrate in our numerical experiments in practice $\lambda$ is very small or even zero (e.g. see the toy model in Section 2.3) so that requirements of the form (2.4) may require unrealistically (or even infinitely) wide networks. In contrast, our results apply to all Jacobian spectrums, however can further harness the low-rank structure of the Jacobian to give even stronger bounds.

• **Small width is sufficient for generalization:** Based on our simulations the M-NTK indeed has low-rank structure with a few large eigenvalues and many smaller ones. As a result a reasonable scaling choice of $\alpha$ is constant. In that case our result states that as soon as the number of hidden nodes are logarithmic in $n$, good generalization can be achieved. This favorably compares to related works Jacot et al. (2018); Arora et al. (2019); Du et al. (2018b); Allen-Zhu et al. (2018b); Cao & Gu (2019) where network size is required to grow polynomial with $n$ and inversely with the distance between the inputs or other notions of margin.

• **Network size–Bias tradeoff:** Based on the requirement (2.2) if the network is large (in terms of # of hidden units $k$), we can choose a small cut-off $\alpha$. This in turn allows us to enlarge the information space and reduce the training bias further. In summary, as the network capacity grows, we can gradually interpolate finer detail and reduce bias.

• **Fast convergence:** Note that the number of gradient iterations is upper bounded by $\Gamma/\alpha^2$. Hence, the training speed is dictated by and is inversely proportional to the the smallest singular value over the information space. Specifically, picking $\alpha$ to be a constant, convergence on the information space will be fast requiring only a *constant number of iterations* to reach any fixed accuracy (see (2.3)).

## 2.2 Generalization guarantees with arbitrary initialization

Our next result provides generalization guarantees from an arbitrary initialization which applies to pre-trained networks (e.g. those that arise in transfer learning applications) as well as intermediate gradient iterates as the weights evolve. This result has a similar flavor to Theorem 2.2 with the key difference that the information and nuisance spaces are defined with respect to any arbitrary initial Jacobian. This shows that if a pre-trained model[3] provides a better low-rank representation of the data in terms of its Jacobian, it is more likely to generalize well. Furthermore, given its deterministic nature the theorem can be applied at any iteration, implying that if the Jacobians of any of the iterates provides a better low-rank representation of the data then one can provide sharper generalization guarantees. The following theorem is a (non-rigorous) simplification of Theorem 6.21.

**Theorem 2.3** *Let $\Gamma \geq 1, \alpha$ be arbitrary scalars. Consider i.i.d. training data $\{(\boldsymbol{x}_i, \boldsymbol{y}_i)\}_{i=1}^n \in \mathbb{R}^d \times \mathbb{R}^K$ with unit length inputs and one-hot encoded labels. Also consider a neural net with $k$ hidden nodes as in (1.1) parameterized by $\boldsymbol{W}$. Let $\boldsymbol{W}_0$ be an **arbitrary initial weight matrix** and assume the output matrix has bounded entries obeying $\|\boldsymbol{V}\|_{\ell_\infty} \leq \frac{1}{\sqrt{kK}}$. Define the nuisance space $\mathcal{N}$ associated with $\mathcal{J}(\boldsymbol{W}_0)$ based on spectrum cutoff $\alpha\sqrt{n}$. Set the initial residual $\boldsymbol{r}_0 = f(\boldsymbol{W}_0) - \boldsymbol{y} \in \mathbb{R}^{nK}$ and assume $\|\boldsymbol{r}_0\|_{\ell_2} \lesssim \sqrt{n}$. Suppose $k \gtrsim \Gamma^4/\alpha^8$. After $T \propto \Gamma/\alpha^2$ iterations (1.5) with constant learning rate, training loss obeys: $\|f(\boldsymbol{W}_T) - \boldsymbol{y}\|_{\ell_2} \lesssim \|\Pi_{\mathcal{N}}(\boldsymbol{r}_0)\|_{\ell_2} + \mathrm{e}^{-\Gamma}\sqrt{n}$.*

---

[2]Based on our understanding Arora et al. (2019) requires the number of hidden units to be on the order of $k \gtrsim n^8/\lambda^6$. Hence our result reduces the dependence on width by a factor of at least $n^4/\lambda^2$.

[3]e.g. obtained by training with data in a related problem as is common in transfer learning.

*Also with high probability, classification error obeys: $Err_{\mathcal{D}}(\boldsymbol{W}_T) \lesssim \frac{\|\Pi_{\mathcal{N}}(\boldsymbol{r}_0)\|_{\ell_2}}{\sqrt{n}} + e^{-\Gamma} + \frac{\Gamma}{\alpha\sqrt{n}}$.*

As with the random initialization result, this theorem shows that as long as the initial residual is sufficiently correlated with the information space, then high accuracy can be achieved for neural networks with reasonable size. As with its randomized counterpart this result also allows us to study various tradeoffs between bias-variance and network size-bias. Crucially however this result does not rely on random initialization. The reason this is particularly important is two fold. First, in many scenarios neural networks are not initialized at random. For instance, in transfer learning the network is pre-trained via data from a different domain. Second, as we demonstrate in Section 4 as the iterates progress the Jacobian mapping develops more favorable properties with the labels/initial residuals becoming more correlated with the information space of the Jacobian. As mentioned earlier, due to its deterministic nature the theorem above applies in both of these scenarios. In particular, if a pre-trained model provides a better low-rank representation of the data via its Jacobian, it is more likely to generalize well. Furthermore, given its deterministic nature the theorem can be applied at any iteration by setting $\boldsymbol{W}_0 = \boldsymbol{W}_\tau$, implying that if the Jacobians of any of the iterates provides a better low-rank representation then one can provide better generalization guarantees. Our numerical experiments demonstrate that the Jacobian of the neural network adapts to the dataset over time with a more substantial amount of the labels lying on the information space. While we defer the rigorous theory of this adaptation to future, Section D provides a proof sketch of evolution of Jacobian rank for a simple dataset model. Such a result when combined with our result above can potentially provide significantly tighter bounds. This is particularly important in light of recent literature Chizat & Bach (2018b); Ghorbani et al. (2019c); Yehudai & Shamir (2019) suggesting a significant generalization gap between kernel methods/linearized neural nets when compared with neural nets operating beyond the linearized regime (e.g. mean field regime). As a result we view our deterministic result as a first step towards moving beyond the NTK regime.

## 2.3 CASE STUDY: GAUSSIAN MIXTURE MODEL

To illustrate a concrete example, we consider a distribution based on multiclass mixture models.

**Definition 2.4 (Gaussian mixture model)** *Consider a size $n$ dataset $\{(\boldsymbol{x}_i, \boldsymbol{y}_i)\}_{i=1}^n \in \mathbb{R}^d \times \mathbb{R}^K$. We assume this dataset consists of $K$ classes each comprising of $C$ clusters with a total of $KC$ clusters. We index each cluster with $(\ell, \widetilde{\ell})$ denoting the $\widetilde{\ell}$th cluster from the $\ell$th class. We assume the dataset in cluster $(\ell, \widetilde{\ell})$ is centered around a cluster center $\boldsymbol{\mu}_{\ell, \widetilde{\ell}} \in \mathbb{R}^d$ with unit Euclidian norm. We assume the dataset is generated i.i.d. with the cluster membership assigned uniformly of the clusters with probability $\frac{1}{KC}$ and the input samples associated with the cluster $(\ell, \widetilde{\ell})$ are generated i.i.d. according to $\mathcal{N}(\boldsymbol{\mu}_{\ell, \widetilde{\ell}}, \sigma^2 \boldsymbol{I}_d/d)$ with the corresponding label set to the one-hot encoding of the class $\ell$ i.e. $\boldsymbol{e}_\ell$. Note that the cluster indexed by $(\ell, \widetilde{\ell})$ contains $\widetilde{n}_{\ell, \widetilde{\ell}}$ data points satisfying $\mathbb{E}[\widetilde{n}_{\ell, \widetilde{\ell}}] = \widetilde{n} = n/KC$.*

This distribution is an ideal candidate to demonstrate why the Jacobian of the network exhibits low-rank or bimodal structure. Let us consider the extreme case $\sigma = 0$ where we have a discrete input distribution over the cluster centers. In this scenario, the multi-class Jacobian matrix is at most rank

$$K^2 C = \text{\# of output nodes } \times \text{ \# of distinct inputs.}$$

as there are (i) only $KC$ distinct input vectors and (ii) $K$ output nodes. We can thus set the information space to be the top $K^2 C$ eigenvectors of the multiclass kernel matrix $\boldsymbol{\Sigma}(\boldsymbol{X})$. As formalized in the appendix, it can be shown that

- The singular values of the information space grow proportionally with $n/KC$.
- The concatenated label vector $\boldsymbol{y}$ perfectly lies on the information space.

In Figure 3 we numerically verify that the approximate rank and singular values of the Jacobian indeed scale as above even when $\sigma > 0$. The following informal theorem leverages these observations to establish a generalization bound for this mixture model. This informal statement is for exposition purposes. See Theorem A.3 in Appendix A for a more detailed result capturing the exact dependencies (e.g. $\zeta, B, \log n$). In this theorem we use $\gtrsim$ to denote inequality up to constant/logarithmic factors.

**Theorem 2.5 (Generalization for Gaussian Mixture Models-simplified)** *Consider a data set of size $n$ consisting of input/label pairs $\{(\boldsymbol{x}_i, \boldsymbol{y}_i)\}_{i=1}^n \in \mathbb{R}^d \times \mathbb{R}^K$ generated according to Def. 2.4 with the standard deviation obeying $\sigma \lesssim \frac{K}{n}$. Let $\boldsymbol{M} = [\boldsymbol{\mu}_{1,1} \ \ldots \ \boldsymbol{\mu}_{K,C}]^T$ be the matrix obtained by aggregating all the cluster centers and let $\boldsymbol{g} \sim \mathcal{N}(0, \boldsymbol{I}_d)$. Also let $\boldsymbol{\Sigma}(\boldsymbol{M}) \in \mathbb{R}^{KC \times KC}$ be the M-NTK*

*associated with the cluster centers $M$ per Def. 2.1. Furthermore, set $\lambda_M = \lambda_{\min}(\Sigma(M))$, and assume $\lambda_M > 0$. If the number of hidden nodes obeys $k \gtrsim \frac{\Gamma^4 K^8 C^4}{\lambda_M^4}$. after $T = \frac{2\Gamma K^2 C}{\lambda_M}$ gradient iterations, with high probability, the model obeys $Err_{\mathcal{D}}(W_T) \lesssim \Gamma\sqrt{\frac{K^2 C}{n\lambda_M}}$.*

We note that $\lambda_M$ captures how diverse the cluster centers are. In this sense $\lambda_M > 0$ intuitively means that neural network, specifically the neural tangent kernel, is sufficiently expressive to interpolate the cluster centers. In fact when the cluster centers are in generic position $\lambda_M$ scales like a constant Oymak & Soltanolkotabi (2019). This theorem focuses on the regime where the noise level $\sigma$ is small. In this case one can achieve good generalization as soon as the sample size scales as $n \gtrsim K^2 C$ which is the effective rank of the M-NTK matrix. This result follows from our main result with random initialization by setting the cutoff at $\alpha^2 \sim \frac{\lambda_M}{K^2 C}$. This demonstrates that in this model $\alpha$ does indeed scale as a constant. Finally, the required network width is independent of $n$ and only depends on $K$ and $C$ specifically $k \gtrsim K^8 C^4$. This compares favorably with Arora et al. (2019) which concerns the $K = 1$ case. In particular, Arora et al. (2019) requires $k \gtrsim n^8/\lambda_X^6$ which depends on $n$ (in lieu of $K$ and $C$) and the minimum eigenvalue $\lambda_X$ of the NTK matrix $\Sigma(X)$ (rather than $\lambda_M$). Furthermore, as $\sigma \to 0$, $\Sigma(X)$ becomes rank deficient and $\lambda_X \to 0$ so that Arora et al. (2019) requires infinite width.

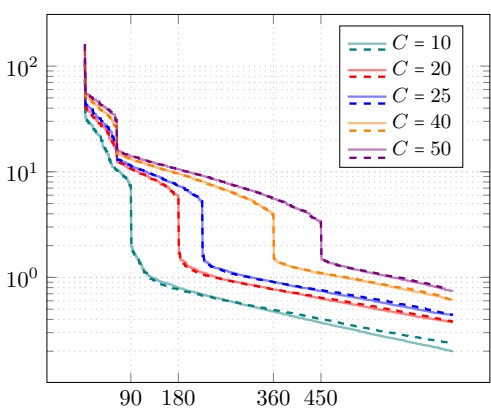

Singular value index (x-axis marks $K^2 C$)

Figure 3: The singular values of the normalized Jacobian spectrum $\sqrt{\frac{KC}{n}}\mathcal{J}(W_0)$ of a neural network with $K = 3$. Here, the data is generated according to the Def. 2.4 with $K$ classes and $\sigma = 0.1$. The cluster centers are picked so that the distance between any two is at least 0.5. We consider two cases: $n = 30C$ (solid line) and $n = 60C$ (dashed line). These plots demonstrate that the top $KC$ singular values grow proportional to $\sqrt{n}$.

## 3 PRIOR ART

Neural networks have impressive generalization abilities even when they are trained with more parameters than the data Zhang et al. (2016). Thus, optimization/generalization properties of neural nets have been the topic of recent literature Zhang et al. (2016). Below we discuss the works on statistical learning, optimization, and implicit bias.

**Statistical learning theory:** Statistical properties of neural networks have been studied since 1990's Anthony & Bartlett (2009); Bartlett et al. (1999); Bartlett (1998). With the success of deep networks, there is a renewed interest in understanding capacity of the neural networks under different norm constraints or network architectures Dziugaite & Roy (2017); Arora et al. (2018); Neyshabur et al. (2017b); Golowich et al. (2017). Bartlett et al. (2017); Neyshabur et al. (2017a) established tight sample complexity results for deep networks based on spectral norms. See also Nagarajan & Kolter (2019) for improvements via leveraging various properties of the inter-layer Jacobian and Long & Sedghi (2019) for results with convolutional networks. Related, Arora et al. (2018) leverages compression techniques for constructing tighter bounds. Yin et al. (2018) jointly studies statistical learning and adversarial robustness. These interesting results, provide generalization guarantees for the optimal solution to the empirical risk minimizer.

**Properties of gradient descent:** There is a growing understanding that solutions found by first-order methods such as gradient descent have often favorable properties. Generalization properties of stochastic gradient descent is extensively studied empirically Keskar et al. (2016); Hardt et al. (2015); Sagun et al. (2017); Chaudhari et al. (2016); Hoffer et al. (2017); Goel & Klivans (2017); Goel et al. (2018). For linearly separable datasets, Soudry et al. (2018); Gunasekar et al. (2018); Brutzkus et al. (2017); Ji & Telgarsky (2018a;b) show that first-order methods find solutions that generalize well without an explicit regularization for logistic regression. An interesting line of work establish connection between kernel methods and neural networks and study the generalization abilities of kernel methods when the model interpolates the training data Dou & Liang (2019); Belkin et al. (2018a;b; 2019); Liang & Rakhlin (2018); Belkin et al. (2018c). Chizat & Bach (2018a); Song et al. (2018); Mei et al. (2018); Sirignano & Spiliopoulos (2018); Rotskoff & Vanden-Eijnden (2018) relate the distribution of the network weights to Wasserstein gradient flows using mean field analysis.

**Global convergence and generalization of neural nets:** Closest to our work, recent literature Cao & Gu (2019); Arora et al. (2019); Ma et al. (2019); Allen-Zhu et al. (2018a) provides generalization bounds for overparameterized networks trained via gradient descent. Also see Li et al. (2018); Huang et al. (2019) for interesting visualization of the optimization and generalization landscape. Jacot et al. (2018) introduced NTK and observed that principal directions of NTK is optimized faster than smaller eigendirections for infinitely wide networks. In connection to this, our Def 1.1 helps quantify the low-rankness and bimodality of the Jacobian spectrum (same as NTK for random initialization). Similar to Thm 2.2, Arora et al. (2019) uses the NTK to provide generalization guarantees in a similar framework to Jacot et al. (2018) (see (2.4) for comparison). Li et al. (2019a) leverages low-rank Jacobian structure to establish robustness to label noise. Very recent work Su & Yang (2019) uses low-rankness to better capture approximation power of neural nets. These works build on global convergence results of randomly initialized networks Du et al. (2018b;a); Allen-Zhu et al. (2018b); Chizat & Bach (2018b); Zhang et al. (2019); Nitanda & Suzuki (2019); Oymak & Soltanolkotabi (2018); Zou et al. (2018) which study the gradient descent trajectory via comparisons to a NTK linearization. These results however typically require unrealistically wide networks for optimization where the width grows poly in $n$ and poly-inversely proportional to the distance between the input samples. Example distance measures are class margin for logistic loss and minimum eigenvalue of the NTK matrix for least-squares. Our work circumvents this issue by allowing a capacity-dependent interpolation. We prove that even small networks (e.g. of constant width) can interpolate the data over a low-dimensional information space without making restrictive assumptions on the input. This approach also leads to faster convergence rates. In terms of generalization, our work has three distinguishing features: (a) bias-variance tradeoffs by identifying information/nuisance spaces, (b) no margin/distance/minimum eigenvalue assumptions on data, (c) the bounds apply to multiclass classification as well as pre-trained networks (Theorem 2.3).

Finally, low-rankness of the Jacobian plays a central role in this work. Hessian and Jacobian of neural nets are investigated by multiple papers which contain related findings on the bimodal (approximately low-rank) spectrum Papyan (2018); Ghorbani et al. (2019b); Papyan (2019b); Sagun et al. (2017); Li et al. (2019b); Javadi et al. (2019). Our key empirical contribution is establishing (in great detail) that multiclass Jacobian adapts over time to align its information space with the labels to better represent the data. This alignment leads to tighter generalization bounds in our analysis shedding light on representation learning and gradient dynamics beyond NTK.

## 4 NUMERICAL EXPERIMENTS

We present experiments demonstrating our theoretical findings on two popular image classification datasets. In this section we focus on a set of CIFAR-10 experiments and discuss how our theory is strongly supported by what we observe in practice. To provide more detail and show that our theory holds across different datasets, in addition to the experiments discussed in this section we perform additional experiments on a modified 3-class version of CIFAR-10 and MNIST in Appendix C.

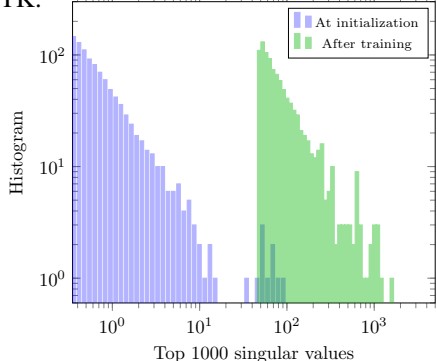

Figure 4: Histogram of the top 1000 Jacobian singular values on the CIFAR10 dataset.

**Experimental setup.** The CIFAR-10 dataset consists of $50k$ training images and $10k$ test images in 10 classes. We demonstrate our results on ResNet20, a state-of-the-art architecture with a fairly low test error on this dataset ($8.75\%$ test error reported) and relatively few parameters ($0.27M$). In all of our experiments we set the information space to be the span of the top 50 singular vectors (out of total dimension of $Kn \approx 500000$) of the neural network Jacobian. In order to be consistent with our theoretical formulation we make the following modifications to the default ResNet20 architecture: (1) we scale the output of the final fully connected layer to ensure that the output is small, consistent with Theorem 2.2 (2) we turn off batch normalization and (3) we do not pass the network output through a soft-max function. We train the network with SGD on least-squares loss with batch size 128 and without any form of data augmentation. We set the initial learning rate to $0.1$ and adjust the learning rate schedule and number of epochs depending on the particular experiment so as to achieve a good fit to the training data quickly. The figures in this section depict the minimum error over a window consisting of the last 10 epochs for visual clarity. We also conduct two sets of experiments to illustrate the results on

| | $\dfrac{\|\Pi_{\mathcal{I}}(\boldsymbol{y})\|_{\ell_2}}{\|\boldsymbol{y}\|_{\ell_2}}$ | $\dfrac{\|\Pi_{\mathcal{N}}(\boldsymbol{y})\|_{\ell_2}}{\|\boldsymbol{y}\|_{\ell_2}}$ | $\dfrac{\|\Pi_{\mathcal{I}}(\boldsymbol{r}_0)\|_{\ell_2}}{\|\boldsymbol{r}_0\|_{\ell_2}}$ | $\dfrac{\|\Pi_{\mathcal{N}}(\boldsymbol{r}_0)\|_{\ell_2}}{\|\boldsymbol{r}_0\|_{\ell_2}}$ |
|---|---|---|---|---|
| $\boldsymbol{J}^{train}_{init}$ | 0.38081 | 0.92465 | 0.37114 | 0.92858 |
| $\boldsymbol{J}^{train}_{final}$ | 0.9869 | 0.16131 | 0.98669 | 0.1626 |

Table 1: Depiction of the alignment of the initial residual with the information/nuisance space using uncorrupted data and a Multi-class ResNet20 model trained with SGD.

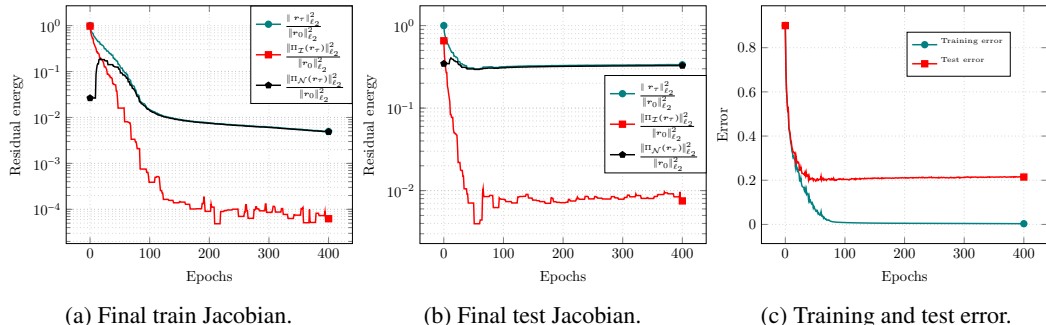

(a) Final train Jacobian.     (b) Final test Jacobian.     (c) Training and test error.

Figure 5: Evolution of the residual ($\boldsymbol{r}_\tau = f(\boldsymbol{W}_\tau) - \boldsymbol{y}$) along the information/nuisance spaces of the final Jacobian on (a) the training data and (b) the test data and c) misclassification error on training and test. This experiment uses uncorrupted labels.

uncorrupted and corrupted data. In this section we highlight some of these results and relate them to our theoretical framework. For the complete set of experiments we refer the reader to Appendix C.

**Jacobian eigenstructure.** Calculating the exact full singular value decomposition of the Jacobian at this scale ($500k \times 270k$) is not tractable due to computation/memory limitations. In order to verify the bimodal structure of the Jacobian with exact singular values we plot the histogram of the top 1000 singular values of the Jacobian mapping at initalization and after training in Figure 4. This figure clearly demonstrates that the Jacobian has low-rank structure. In both cases we observe that singular values are concentrated around zero with a relatively small density distributed over higher singular values. This observation serves as a natural basis for decomposition of the label space into information $\mathcal{I}$ (large singular values, low-dimensional) and nuisance space $\mathcal{N}$ (small singular values, high-dimensional). We note that while calculating all the eigenvalues is not possible, we verify the bimodal structure of the entire Jacobian spectrum by approximating its spectral density in App. C.

**Experiments without label corruption.** First, we present experiments on the original training data described above with no label corruption. We train the network for 400 epochs to achieve a good fit to the training data. Our theory predicts that the sum of $\left\|\boldsymbol{J}^{\dagger}_{\mathcal{I}}\boldsymbol{y}\right\|_{\ell_2}$ and $\|\Pi_{\mathcal{N}}(\boldsymbol{y})\|_{\ell_2}$ determines the classification error (Theorems 2.2 and 6.24). Table 1 collects these values for the initial and final Jacobian. These values demonstrate that the label vector is indeed correlated with the top eigenvectors of both the initial and final Jacobians. An interesting aspect of these results is that this correlation increases from the initial to the final Jacobian so that more of the label energy lies on the information space of the final Jacobian in comparison with the initial Jacobian. Stated differently, we observe a significant adaptation of the Jacobian to the labels after training compared to the initial Jacobian so that our predictions become more and more accurate as the iterates progress. In particular, the first column of Table 1 shows that the fraction of label energy lying on the information subspace of the Jacobian drastically increases after training (from 0.38 to 0.99). Consequentially, less energy falls on the nuisance space (decreases from 0.92 to 0.16 after training), while $\left\|\boldsymbol{J}^{\dagger}_{\mathcal{I}}\boldsymbol{y}\right\|_{\ell_2}$ remains relatively small resulting in better generalization. Towards explaining these, Section D provides a preliminary analysis showing Jacobian spectrum indeed adapts to data.

We also track the projection of the residual $\boldsymbol{r}_\tau$ on the information and nuisance subspaces throughout training on both training and test data and depict the results in Figures 5a and 5b. In agreement with our theory, these plots show that learning on $\mathcal{I}$ is fast and the residual energy decreases rapidly on this space. On the other hand, residual energy on $\mathcal{N}$ goes down rather slowly and the decrease in total residual energy is overwhelmingly governed by $\mathcal{I}$, suggesting that most information relevant to learning lies in this space. We also plot the training and test error in Figure 5c. We observe that as learning progresses, the residual on both spaces decrease in tandem with training and test error.

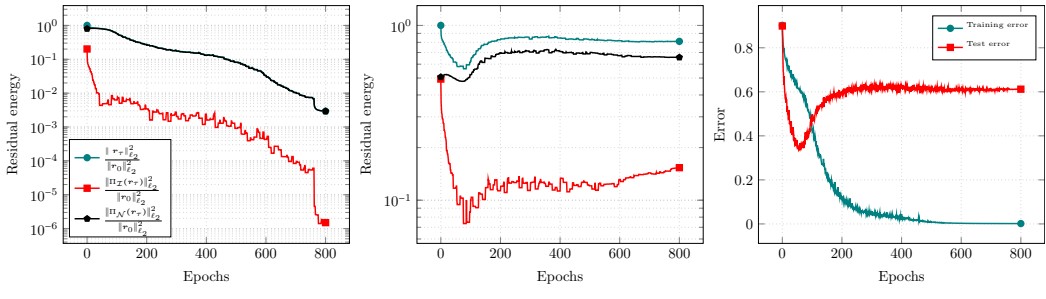

(a) 50 epochs train Jacobian.  (b) 50 epochs test Jacobian.  (c) Training and test error

Figure 6: Evolution of the residual ($r_\tau = f(W_\tau) - y$) along the information/nuisance spaces of the Jacobian at $50$ epochs on (a) training data and (b) test data and (c) misclassification error on training and test data. $50\%$ of the labels have been corrupted.

| | $\frac{\|\Pi_{\mathcal{I}}(\boldsymbol{y})\|_{\ell_2}}{\|\boldsymbol{y}\|_{\ell_2}}$ | $\frac{\|\Pi_{\mathcal{N}}(\boldsymbol{y})\|_{\ell_2}}{\|\boldsymbol{y}\|_{\ell_2}}$ | $\frac{\|\Pi_{\mathcal{I}}(\boldsymbol{r}_0)\|_{\ell_2}}{\|\boldsymbol{r}_0\|_{\ell_2}}$ | $\frac{\|\Pi_{\mathcal{N}}(\boldsymbol{r}_0)\|_{\ell_2}}{\|\boldsymbol{r}_0\|_{\ell_2}}$ |
|---|---|---|---|---|
| $\boldsymbol{J}_{init}^{train}$ | 0.32762 | 0.94481 | 0.32152 | 0.9469 |
| $\boldsymbol{J}_{final}^{train}$ | 0.8956 | 0.44487 | 0.89597 | 0.44412 |

Table 2: Depiction of the alignment of the initial residual with the information/nuisance space using $50\%$ label corrupted data and a Multi-class ResNet20 trained with SGD.

**Experiments with label corruption.** Our next experiments study the effect of corruption. Specifically, we corrupt $50\%$ of the labels by randomly picking a label from a (strictly) different class. We train the network for $800$ epochs and divide the learning rate by $10$ at epoch $760$ to fit to the training data. We again track the projection of the residual $r_\tau$ on the information/nuisance spaces throughout training on both training and test data and depict the results in Figs 6a and 6b. We also track the train and test errors in Figure 6c. From Figure 6c it is evident that while the training error steadily decreases, test error exhibits a very different behavior compared to the uncorrupted experiment. In the first phase, test error drops rapidly as the network learns from information contained in the uncorrupted data, accompanied by a corresponding decrease in residual energy on the information subspace on the training data (Figure 6a). The lowest test error is observed at epoch 50 after which a steady increase follows. In the second phase, the network overfits to the corrupted data resulting in more test error on the uncorrupted test data (Figure 6b). More importantly, the increase of the test error is due to the nuisance space as the error over information space is stable while it

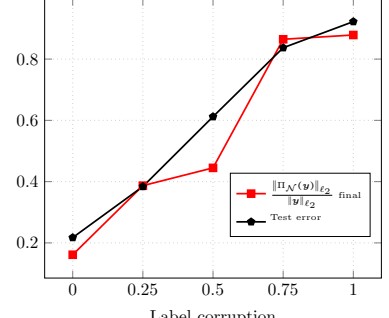

Figure 7: Fraction of the energy of the label vector that lies on the nuisance space of the initial Jacobian and final Jacobian as well as the test error as a function of the amount of label corruption.

increases over the nuisance. In particular the residual on $\mathcal{N}$ slowly increases while residual on $\mathcal{I}$ drops sharply creating a dip in both test error and total residual energy around epoch 50. This phenomenon is further explained in the appendix (see Sec. 5.1) via a linear model.

In Table 2 we again depict the fraction of the energy of the labels and the initial residual that lies on the information/nuisance spaces. The Jacobian continues to adapt to the labels/initial residual even in the presence of label corruption, albeit to a smaller degree. We note that due to corruption, labels are less correlated with the information space of the Jacobian and the fraction of the energy on the nuisance space is higher which results in worse generalization (as predicted by our theory).

To demonstrate the connection between generalization and information/nuisance spaces, we repeat the experiment with $25\%$, $75\%$ and $100\%$ label corruption and depict the results after 800 epochs in Fig. 7. As expected, the test error increases with the corruption. Furthermore, the corrupted labels become less correlated with the information space with more of the label energy falling onto the nuisance space. This is consistent with our theory which predicts worse generalization in this case.

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

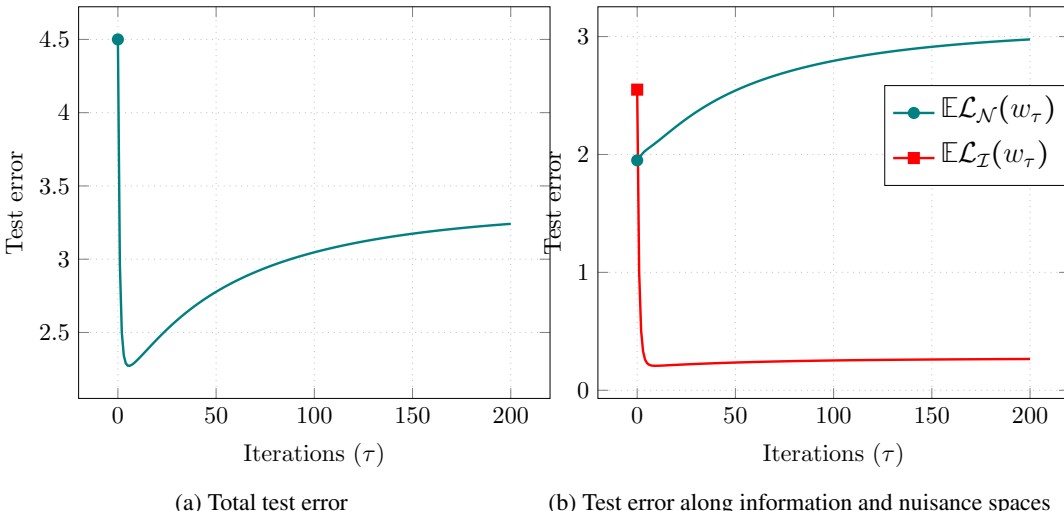

(a) Total test error                 (b) Test error along information and nuisance spaces

Figure 8: Plots of the (a) total test error and (b) the test error components for the model in Section 5.1. The test error decreases rapidly over the information subspace but slowly increases over the nuisance subspace.

## 5 TECHNICAL APPROACH AND GENERAL THEORY

### 5.1 PRELUDE: FITTING A LINEAR MODEL

To gain better insights into what governs the generalization capability of gradient based iterations let us consider the simple problem of fitting a linear model via gradient descent. This model maps an input/feature vector $\boldsymbol{x} \in \mathbb{R}^d$ into a one-dimensional output/label via $\boldsymbol{x} \mapsto f(\boldsymbol{x}, \boldsymbol{w}) \coloneqq \boldsymbol{w}^T \boldsymbol{x}$. We wish to fit a model of this form to $n$ training data consisting of input/label pairs $\{(\boldsymbol{x}_i, y_i)\}_{i=1}^n \in \mathbb{R}^d \times \mathbb{R}$. Aggregating this training data as rows of a feature matrix $\boldsymbol{X} \in \mathbb{R}^{n \times d}$ and label vector $\boldsymbol{y} \in \mathbb{R}^n$, the training problem takes the form

$$\mathcal{L}(\boldsymbol{w}) = \frac{1}{2} \|\boldsymbol{X}\boldsymbol{w} - \boldsymbol{y}\|_{\ell_2}^2. \tag{5.1}$$

We focus on an overparameterized model where there are fewer training data than the number of parameters i.e. $n \le d$. We assume the feature matrix can be decomposed into the form $\boldsymbol{X} = \overline{\boldsymbol{X}} + \boldsymbol{Z}$ where $\overline{\boldsymbol{X}}$ is low-rank (i.e. $\text{rank}(\overline{\boldsymbol{X}}) = r \ll n$) with singular value decomposition $\overline{\boldsymbol{X}} = \boldsymbol{U}\boldsymbol{\Sigma}\boldsymbol{V}^T$ with $\boldsymbol{U} \in \mathbb{R}^{n \times r}$, $\boldsymbol{\Sigma} \in \mathbb{R}^{r \times r}$, $\boldsymbol{V} \in \mathbb{R}^{d \times r}$, and $\boldsymbol{Z} \in \mathbb{R}^{n \times d}$ is a matrix with i.i.d. $\mathcal{N}(0, \sigma_x^2/n)$ entries. We shall also assume the labels are equal to $\boldsymbol{y} = \overline{\boldsymbol{y}} + \boldsymbol{z}$ with $\overline{\boldsymbol{y}} = \overline{\boldsymbol{X}}\boldsymbol{w}^*$ for some $\boldsymbol{w}^* \in \text{Range}(\boldsymbol{V})$ and $\boldsymbol{z} \in \mathbb{R}^n$ a Gaussian random vector with i.i.d. $\mathcal{N}(0, \sigma_y^2/n)$ entries. One can think of this as a linear regression model where the features and labels are corrupted with Gaussian noise. The goal of course is to learn a model which fits to the clean uncorrupted data and not the corruption. In this case the population loss (i.e. test error) takes the form

$$\mathbb{E}\left[\mathcal{L}(\boldsymbol{w})\right] = \frac{1}{2} \|\overline{\boldsymbol{X}}\boldsymbol{w} - \overline{\boldsymbol{y}}\|_{\ell_2}^2 + \frac{1}{2}\sigma_x^2 \|\boldsymbol{w}\|_{\ell_2}^2 + \frac{1}{2}\sigma_y^2,$$

Now let us consider gradient descent iterations with a step size $\eta$ which take the form

$$\boldsymbol{w}_{\tau+1} = \boldsymbol{w}_\tau - \eta \nabla \mathcal{L}(\boldsymbol{w}_\tau) = \left(\boldsymbol{I} - \eta \boldsymbol{X}^T \boldsymbol{X}\right) \boldsymbol{w}_\tau + \eta \boldsymbol{X}^T \boldsymbol{y}. \tag{5.2}$$

To gain further insight into the generalization capabilities of the gradient descent iterations we shall consider an instance of this problem where the subspaces $\boldsymbol{U}$ and $\boldsymbol{V}$ are chosen uniformly at random, $\boldsymbol{\Sigma} = \boldsymbol{I}_r$ with $n = 200$, $d = 500$, $r = 5$, and $\sigma_x = 0.2$, $\sigma_y = 2$. In Figure 8a we plot the population loss evaluated at different iterations. We observe an interesting phenomenon, in the first few iterations the test error goes down quickly but it then slowly increases. To better understand this behavior we decompose the population loss into two parts by tracking the projection of the misfit $\boldsymbol{X}\boldsymbol{w} - \boldsymbol{y}$ on the column space of the uncorrupted portion of the input data ($\boldsymbol{U}$) and its complement. That is,

$$\mathbb{E}\,\mathcal{L}(\boldsymbol{w}) = \mathbb{E}\,\mathcal{L}_{\mathcal{I}}(\boldsymbol{w}) + \mathbb{E}\,\mathcal{L}_{\mathcal{N}}(\boldsymbol{w}).$$

where

$$\mathbb{E}\,\mathcal{L}_{\mathcal{I}}(\boldsymbol{w}) := \mathbb{E}\Big[\, \|\Pi_{\mathcal{I}}\left(\boldsymbol{X}\boldsymbol{w} - \boldsymbol{y}\right)\|_{\ell_2}^2 \Big] = \big\|\overline{\boldsymbol{X}}\boldsymbol{w} - \overline{\boldsymbol{y}}\big\|_{\ell_2}^2 + \frac{r}{2n}\sigma_x^2\|\boldsymbol{w}\|_{\ell_2}^2 + \frac{r}{2n}\sigma_y^2,$$

$$\mathbb{E}\,\mathcal{L}_{\mathcal{N}}(\boldsymbol{w}) := \mathbb{E}\Big[\, \|\Pi_{\mathcal{N}}\left(\boldsymbol{X}\boldsymbol{w} - \boldsymbol{y}\right)\|_{\ell_2}^2 \Big] = \frac{1}{2}\left(1 - \frac{r}{n}\right)\left(\sigma_x^2\|\boldsymbol{w}\|_{\ell_2}^2 + \sigma_y^2\right),$$

with $\Pi_{\mathcal{I}} = \boldsymbol{U}\boldsymbol{U}^T$ and $\Pi_{\mathcal{N}} = \boldsymbol{I} - \boldsymbol{U}\boldsymbol{U}^T$. In Figure 8b we plot these two components. This plot clearly shows that $\mathbb{E}\,\mathcal{L}_{\mathcal{I}}(\boldsymbol{w})$ goes down quickly while $\mathbb{E}\,\mathcal{L}_{\mathcal{N}}(\boldsymbol{w})$ slowly increases with their sum creating the dip in the test error. Since $\boldsymbol{U}$ is a basis for the range of the uncorrupted portion of the features ($\overline{\boldsymbol{X}}$) one can think of span($\boldsymbol{U}$) as the "information" subspace and $\mathbb{E}\,\mathcal{L}_{\mathcal{I}}(\boldsymbol{w})$ as the test error on this information subspace. Similarly, one can think of the complement of this subspace as the "nuisance" subspace and $\mathbb{E}\,\mathcal{L}_{\mathcal{N}}(\boldsymbol{w})$ as the test error on this nuisance subspace. Therefore, one can interpret Figure 8a as the test error decreasing rapidly in the first few iterations over the information subspace but slowly increasing due to the contributions of the nuisance subspace.

To help demystify this behavior note that using the gradient descent updates from (5.2) the update in terms of the misfit/residual $\boldsymbol{r}_\tau = \boldsymbol{X}\boldsymbol{w}_\tau - \boldsymbol{y}$ takes the form

$$\boldsymbol{r}_{\tau+1} = \left(\boldsymbol{I} - \eta\boldsymbol{X}\boldsymbol{X}^T\right)\boldsymbol{r}_\tau = \left(\boldsymbol{I} - \eta\overline{\boldsymbol{X}}\,\overline{\boldsymbol{X}}^T\right)\left(\overline{\boldsymbol{X}}\boldsymbol{w}_\tau - \overline{\boldsymbol{y}}\right) + noise$$

Based on the form of this update when the information subspace is closely aligned with the prominent singular vectors of $\boldsymbol{X}$ the test error on the information subspace ($\mathbb{E}\,\mathcal{L}_{\mathcal{I}}(\boldsymbol{w}) \approx \big\|\overline{\boldsymbol{X}}\boldsymbol{w}_\tau - \overline{\boldsymbol{y}}\big\|_{\ell_2}^2$) quickly decreases in the first few iterations. However, the further we iterate the parts of the residual aligned with the less prominent eigen-directions of $\boldsymbol{X}$ (which correspond to the nuisance subspace) slowly pick up more energy contributing to a larger total test error. In this section, we outline our approach to proving robustness of over-parameterized neural networks. Towards this goal, we consider a general formulation where we aim to fit a general nonlinear model of the form $\boldsymbol{x} \mapsto f(\boldsymbol{x}; \boldsymbol{\theta})$ with $\boldsymbol{x} \in \mathbb{R}^d$ denoting the input features, $\boldsymbol{\theta} \in \mathbb{R}^p$ denoting the parameters, and $f(\boldsymbol{x}; \boldsymbol{\theta}) \in \mathbb{R}^K$ the $K$ outputs of the model denoted by $f_1(\boldsymbol{x}; \boldsymbol{\theta}), f_2(\boldsymbol{x}; \boldsymbol{\theta}), \ldots, f_K(\boldsymbol{x}; \boldsymbol{\theta})$. For instance in the case of neural networks $\boldsymbol{\theta}$ represents its weights. Given a data set of $n$ input/label pairs $\{(\boldsymbol{x}_i, \boldsymbol{y}_i)\}_{i=1}^n \subset \mathbb{R}^d \times \mathbb{R}^K$, we fit to this data by minimizing a nonlinear least-squares loss of the form

$$\mathcal{L}(\boldsymbol{\theta}) = \frac{1}{2}\sum_{i=1}^n \|f(\boldsymbol{x}_i; \boldsymbol{\theta}) - \boldsymbol{y}_i\|_{\ell_2}^2. \tag{5.3}$$

To continue let us first aggregate the predictions and labels into larger vectors based on class. In particular define

$$f_\ell(\boldsymbol{\theta}) = \begin{bmatrix} f_\ell(\boldsymbol{x}_1; \boldsymbol{\theta}) \\ \vdots \\ f_\ell(\boldsymbol{x}_n; \boldsymbol{\theta}) \end{bmatrix} \in \mathbb{R}^n \quad \text{and} \quad \boldsymbol{y}^{(\ell)} = \begin{bmatrix} (\boldsymbol{y}_1)_\ell \\ \vdots \\ (\boldsymbol{y}_n)_\ell \end{bmatrix} \in \mathbb{R}^n \quad \text{for} \quad \ell = 1, 2, \ldots, K.$$

Concatenating these vectors we arrive at

$$f(\boldsymbol{\theta}) = \begin{bmatrix} f_1(\boldsymbol{\theta}) \\ \vdots \\ f_K(\boldsymbol{\theta}) \end{bmatrix} \in \mathbb{R}^{Kn} \quad \text{and} \quad \boldsymbol{y} = \begin{bmatrix} \boldsymbol{y}^{(1)} \\ \vdots \\ \boldsymbol{y}^{(K)} \end{bmatrix} \in \mathbb{R}^{Kn}. \tag{5.4}$$

Using the latter we can rewrite the optimization problem (5.3) into the more compact form

$$\mathcal{L}(\boldsymbol{\theta}) = \frac{1}{2}\|f(\boldsymbol{\theta}) - \boldsymbol{y}\|_{\ell_2}^2. \tag{5.5}$$

To solve this problem we run gradient descent iterations with a learning rate $\eta$ starting from an initial point $\boldsymbol{\theta}_0$. These iterations take the form

$$\boldsymbol{\theta}_{\tau+1} = \boldsymbol{\theta}_\tau - \eta\nabla\mathcal{L}(\boldsymbol{\theta}_\tau) \quad \text{with} \quad \nabla\mathcal{L}(\boldsymbol{\theta}) = \mathcal{J}^T(\boldsymbol{\theta})\left(f(\boldsymbol{\theta}) - \boldsymbol{y}\right). \tag{5.6}$$

As mentioned earlier due to the form of the gradient the convergence/generalization of gradient descent naturally depends on the spectral properties of the Jacobian. To capture these spectral properties we will use a reference Jacobian $\boldsymbol{J}$ (formally defined below) that is close to the Jacobian at initialization $\mathcal{J}(\boldsymbol{\theta}_0)$.

**Definition 5.1 (Reference Jacobian and its SVD)** *Consider an initial point $\boldsymbol{\theta}_0 \in \mathbb{R}^p$ and the Jacobian mapping $\mathcal{J}(\boldsymbol{\theta}_0) \in \mathbb{R}^{Kn \times p}$. For $\varepsilon_0, \beta > 0$, we call $\boldsymbol{J} \in \mathbb{R}^{Kn \times \max(Kn,p)}$ an $(\varepsilon_0, \beta)$ reference Jacobian matrix if it obeys the following conditions,*

$$\|\boldsymbol{J}\| \leq \beta, \quad \left\|\mathcal{J}(\boldsymbol{\theta}_0)\mathcal{J}^T(\boldsymbol{\theta}_0) - \boldsymbol{J}\boldsymbol{J}^T\right\| \leq \varepsilon_0^2, \quad and \quad \left\|\overline{\mathcal{J}}(\boldsymbol{\theta}_0) - \boldsymbol{J}\right\| \leq \varepsilon_0.$$

*where $\overline{\mathcal{J}}(\boldsymbol{\theta}_0) \in \mathbb{R}^{Kn \times \max(Kn,p)}$ is a matrix obtained by augmenting $\mathcal{J}(\boldsymbol{\theta}_0)$ with $\max(0, Kn - p)$ zero columns. Furthermore, consider the singular value decomposition of $\boldsymbol{J}$ given by*

$$\boldsymbol{J} = \boldsymbol{U} diag(\boldsymbol{\lambda}) \boldsymbol{V}^T = \sum_{s=1}^{Kn} \lambda_s \boldsymbol{u}_s \boldsymbol{v}_s^T. \tag{5.7}$$

*where $\boldsymbol{\lambda} \in \mathbb{R}^{Kn}$ are the vector of singular values and $\boldsymbol{u}_s \in \mathbb{R}^{Kn}$ and $\boldsymbol{v}_s \in \mathbb{R}^p$ are the left/right singular vectors.*

One natural choice for this reference Jacobian is $\boldsymbol{J} = \overline{\mathcal{J}}(\boldsymbol{\theta}_0)$. However, we shall also use other reference Jacobians in our results. We will compare the gradient iterations (5.6) to the iterations associated with fitting a linearized model around $\boldsymbol{\theta}_0$ defined as $f_{\text{lin}}(\widetilde{\boldsymbol{\theta}}) = f(\boldsymbol{\theta}_0) + \boldsymbol{J}(\widetilde{\boldsymbol{\theta}} - \overline{\boldsymbol{\theta}}_0)$, where $\overline{\boldsymbol{\theta}}_0 \in \mathbb{R}^{\max(Kn,p)}$ is obtained from $\boldsymbol{\theta}_0$ by adding $\max(Kn - p, 0)$ zero entries at the end of $\boldsymbol{\theta}_0$. The optimization problem for fitting the linearized problem has the form

$$\mathcal{L}_{lin}(\boldsymbol{\theta}) = \frac{1}{2} \|f_{\text{lin}}(\boldsymbol{\theta}) - \boldsymbol{y}\|_{\ell_2}^2. \tag{5.8}$$

Thus starting from $\widetilde{\boldsymbol{\theta}}_0 = \overline{\boldsymbol{\theta}}_0$ the iterates $\widetilde{\boldsymbol{\theta}}_\tau$ on the linearized problem take the form

$$\begin{aligned}
\widetilde{\boldsymbol{\theta}}_{\tau+1} &= \widetilde{\boldsymbol{\theta}}_\tau - \eta \nabla \mathcal{L}_{lin}(\widetilde{\boldsymbol{\theta}}_\tau), \\
&= \widetilde{\boldsymbol{\theta}}_\tau - \eta \boldsymbol{J}^T(f(\boldsymbol{\theta}_0) + \boldsymbol{J}(\widetilde{\boldsymbol{\theta}}_\tau - \boldsymbol{\theta}_0) - \boldsymbol{y}), \\
&= \widetilde{\boldsymbol{\theta}}_\tau - \eta \boldsymbol{J}^T \boldsymbol{J}(\widetilde{\boldsymbol{\theta}}_\tau - \overline{\boldsymbol{\theta}}_0) - \eta \boldsymbol{J}^T(f(\boldsymbol{\theta}_0) - \boldsymbol{y}).
\end{aligned} \tag{5.9}$$

The iterates based on the linearized problem will provide a useful reference to keep track of the evolution of the original iterates (5.6). Specifically we study the evolution of misfit/residuals associated with the two problems

$$\text{Original residual: } \boldsymbol{r}_\tau = f(\boldsymbol{\theta}_\tau) - \boldsymbol{y}. \tag{5.10}$$

$$\text{Linearized residual: } \widetilde{\boldsymbol{r}}_\tau = f_{\text{lin}}(\widetilde{\boldsymbol{\theta}}_\tau) - \boldsymbol{y} = (\boldsymbol{I} - \eta \boldsymbol{J}\boldsymbol{J}^T)^\tau \boldsymbol{r}_0. \tag{5.11}$$

To better understand the dynamics of convergence of the linearized iterates next we define two subspaces associated with the reference Jacobian and its spectrum.

**Definition 5.2 (Information/Nuisance Subspaces)** *Let $\boldsymbol{J}$ denote the reference Jacobian per Definition 5.1 with eigenvalue decomposition $\boldsymbol{J} = \boldsymbol{U} diag(\boldsymbol{\lambda}) \boldsymbol{V}^T$ per (5.7). For a spectrum cutoff $\alpha$ obeying $0 \leq \alpha \leq \lambda_1$ let $r(\alpha)$ denote the index of the smallest singular value above the threshold $\alpha$, that is,*

$$r(\alpha) = \min\left(\{s \in \{1, 2, \ldots, nK\} \quad such \ that \quad \lambda_s \geq \alpha\}\right).$$

*We define the information and nuisance subspaces associated with $\boldsymbol{J}$ as $\mathcal{I} := span(\{\boldsymbol{u}_s\}_{s=1}^r)$ and $\mathcal{N} := span(\{\boldsymbol{u}_s\}_{s=r+1}^{Kn})$. We also define the truncated reference Jacobian*

$$\boldsymbol{J}_{\mathcal{I}} = \begin{bmatrix} \boldsymbol{u}_1 & \boldsymbol{u}_2 & \ldots & \boldsymbol{u}_r \end{bmatrix} diag\left(\lambda_1, \lambda_2, \ldots, \lambda_r\right) \begin{bmatrix} \boldsymbol{v}_1 & \boldsymbol{v}_2 & \ldots & \boldsymbol{v}_r \end{bmatrix}^T$$

*which is the part of the reference Jacobian that acts on the information subspace $\mathcal{I}$.*

We will show rigorously that the information and nuisance subspaces associated with the reference Jacobian dictate the directions where learning is fast and generalizable versus the directions where learning is slow and overfitting occurs. Before we make this precise we list two assumptions that will be utilized in our result.

**Assumption 1 (Bounded spectrum)** *For any $\boldsymbol{\theta} \in \mathbb{R}^p$ the Jacobian mapping associated with the nonlinearity $f : \mathbb{R}^p \mapsto \mathbb{R}^n$ has bounded spectrum, i.e. $\|\mathcal{J}(\boldsymbol{\theta})\| \leq \beta$.*

**Assumption 2 (Bounded perturbation)** *Consider a point $\boldsymbol{\theta}_0 \in \mathbb{R}^p$ and positive scalars $\varepsilon, R > 0$. Assume that for any $\boldsymbol{\theta}$ obeying $\|\boldsymbol{\theta} - \boldsymbol{\theta}_0\|_{\ell_2} \leq R$, we have*

$$\|\mathcal{J}(\boldsymbol{\theta}) - \mathcal{J}(\boldsymbol{\theta}_0)\| \leq \frac{\varepsilon}{2}.$$

With these assumptions in place we are now ready to discuss our meta theorem that demonstrates that the misfit/residuals associated to the original and linearized iterates do in fact track each other rather closely.

**Theorem 5.3 (Meta Theorem)** *Consider a nonlinear least squares problem of the form $\mathcal{L}(\boldsymbol{\theta}) = \frac{1}{2}\|f(\boldsymbol{\theta}) - \boldsymbol{y}\|_{\ell_2}^2$ with $f : \mathbb{R}^p \mapsto \mathbb{R}^{nK}$ the multi-class nonlinear mapping, $\boldsymbol{\theta} \in \mathbb{R}^p$ the parameters of the model, and $\boldsymbol{y} \in \mathbb{R}^{nK}$ the concatenated labels as in (5.4). Let $\overline{\boldsymbol{\theta}}$ be zero-padding of $\boldsymbol{\theta}$ till size $\max(Kn,p)$. Also, consider a point $\boldsymbol{\theta}_0 \in \mathbb{R}^p$ with $\boldsymbol{J}$ an $(\epsilon_0, \beta)$ reference Jacobian associated with $\mathcal{J}(\boldsymbol{\theta}_0)$ per Definition 5.1 and fitting the linearized problem $f_{lin}(\widetilde{\boldsymbol{\theta}}) = f(\boldsymbol{\theta}_0) + \boldsymbol{J}(\widetilde{\boldsymbol{\theta}} - \overline{\boldsymbol{\theta}}_0)$ via the loss $\mathcal{L}_{lin}(\boldsymbol{\theta}) = \frac{1}{2}\|f_{lin}(\boldsymbol{\theta}) - \boldsymbol{y}\|_{\ell_2}^2$. Furthermore, define the information $\mathcal{I}$ and nuisance $\mathcal{N}$ subspaces and the truncated Jacobian $\boldsymbol{J}_{\mathcal{I}}$ associated with the reference Jacobian $\boldsymbol{J}$ based on a cut-off spectrum value of $\alpha$ per Definition 5.2. Furthermore, assume the Jacobian mapping $\mathcal{J}(\boldsymbol{\theta}) \in \mathbb{R}^{nK \times p}$ associated with $f$ obeys Assumptions 1 and 2 for all $\boldsymbol{\theta} \in \mathbb{R}^p$ obeying*

$$\|\boldsymbol{\theta} - \boldsymbol{\theta}_0\|_{\ell_2} \leq R := 2\left(\left\|\boldsymbol{J}_{\mathcal{I}}^\dagger \boldsymbol{r}_0\right\|_{\ell_2} + \frac{\Gamma}{\alpha}\left\|\Pi_{\mathcal{N}}\left(\boldsymbol{r}_0\right)\right\|_{\ell_2} + \delta\frac{\Gamma}{\alpha}\left\|\boldsymbol{r}_0\right\|_{\ell_2}\right), \tag{5.12}$$

*around a point $\boldsymbol{\theta}_0 \in \mathbb{R}^p$ for a tolerance level $\delta$ obeying $0 < \delta \leq 1$ and stopping time $\Gamma$ obeying $\Gamma \geq 1$. Finally, assume the following inequalities hold*

$$\varepsilon_0 \leq \frac{\min(\delta\alpha, \sqrt{\delta\alpha^3/\Gamma\beta})}{5} \quad and \quad \varepsilon \leq \frac{\delta\alpha^3}{5\Gamma\beta^2}. \tag{5.13}$$

*We run gradient descent iterations of the form $\boldsymbol{\theta}_{\tau+1} = \boldsymbol{\theta}_\tau - \eta\nabla\mathcal{L}(\boldsymbol{\theta}_\tau)$ and $\widetilde{\boldsymbol{\theta}}_{\tau+1} = \widetilde{\boldsymbol{\theta}}_\tau - \eta\nabla\mathcal{L}_{lin}(\widetilde{\boldsymbol{\theta}}_\tau)$ on the original and linearized problems starting from $\boldsymbol{\theta}_0$ with step size $\eta$ obeying $\eta \leq 1/\beta^2$. Then for all iterates $\tau$ obeying $0 \leq \tau \leq T := \frac{\Gamma}{\eta\alpha^2}$ the iterates of the original ($\boldsymbol{\theta}_\tau$) and linearized ($\widetilde{\boldsymbol{\theta}}_\tau$) problems and the corresponding residuals $\boldsymbol{r}_\tau := f(\boldsymbol{\theta}_\tau) - \boldsymbol{y}$ and $\widetilde{\boldsymbol{r}}_\tau := f_{lin}(\widetilde{\boldsymbol{\theta}}_\tau) - \boldsymbol{y}$ closely track each other. That is,*

$$\|\boldsymbol{r}_\tau - \widetilde{\boldsymbol{r}}_\tau\|_{\ell_2} \leq \frac{3}{5}\frac{\delta\alpha}{\beta}\|\boldsymbol{r}_0\|_{\ell_2} \quad and \quad \|\overline{\boldsymbol{\theta}}_\tau - \widetilde{\boldsymbol{\theta}}_\tau\|_{\ell_2} \leq \delta\frac{\Gamma}{\alpha}\|\boldsymbol{r}_0\|_{\ell_2} \tag{5.14}$$

*Furthermore, for all iterates $\tau$ obeying $0 \leq \tau \leq T := \frac{\Gamma}{\eta\alpha^2}$*

$$\|\boldsymbol{\theta}_\tau - \boldsymbol{\theta}_0\|_{\ell_2} \leq \frac{R}{2} = \left\|\boldsymbol{J}_{\mathcal{I}}^\dagger \boldsymbol{r}_0\right\|_{\ell_2} + \frac{\Gamma}{\alpha}\left\|\Pi_{\mathcal{N}}\left(\boldsymbol{r}_0\right)\right\|_{\ell_2} + \delta\frac{\Gamma}{\alpha}\left\|\boldsymbol{r}_0\right\|_{\ell_2}. \tag{5.15}$$

*and after $\tau = T$ iteration we have*

$$\|\boldsymbol{r}_T\|_{\ell_2} \leq e^{-\Gamma}\left\|\Pi_{\mathcal{I}}(\boldsymbol{r}_0)\right\|_{\ell_2} + \left\|\Pi_{\mathcal{N}}(\boldsymbol{r}_0)\right\|_{\ell_2} + \frac{\delta\alpha}{\beta}\|\boldsymbol{r}_0\|_{\ell_2}. \tag{5.16}$$

# 6 PROOFS

Before we proceed with the proof let us briefly discuss some notation used throughout. For a matrix $\boldsymbol{W} \in \mathbb{R}^{k \times d}$ we use $\text{vect}(\boldsymbol{W}) \in \mathbb{R}^{kd}$ to denote a vector obtained by concatenating the rows $\boldsymbol{w}_1, \boldsymbol{w}_2, \ldots, \boldsymbol{w}_k \in \mathbb{R}^d$ of $\boldsymbol{W}$. That is, $\text{vect}(\boldsymbol{W}) = \begin{bmatrix} \boldsymbol{w}_1^T & \boldsymbol{w}_2^T & \ldots & \boldsymbol{w}_k^T \end{bmatrix}^T$. Similarly, we use $\text{mat}(\boldsymbol{w}) \in \mathbb{R}^{k \times d}$ to denote a $k \times d$ matrix obtained by reshaping the vector $\boldsymbol{w} \in \mathbb{R}^{kd}$ across its rows. Throughout, for a differentiable function $\phi : \mathbb{R} \mapsto \mathbb{R}$ we use $\phi'$ and $\phi''$ to denote the first and second derivative.

## 6.1 PROOFS FOR GENERAL THEORY (PROOF OF THEOREM 5.3)

In this section we prove our result for general nonlinearities. We begin with a few notations and definitions and preliminary lemmas in Section 6.1.1. Next in Section 6.1.2 we prove some key lemmas regarding the evolution of the linearized residuals $\widetilde{\boldsymbol{r}}_\tau$. In Section 6.3 we establish some key Rademacher complexity results used in our generalization bounds. Finally, in Section 6.1.3 we use these results to complete the proof of Theorem 5.3.

### 6.1.1 PRELIMINARY DEFINITIONS AND LEMMAS

Throughout we use

$$\boldsymbol{U}_\mathcal{I} = \begin{bmatrix} \boldsymbol{u}_1 & \boldsymbol{u}_2 & \dots & \boldsymbol{u}_r \end{bmatrix} \in \mathbb{R}^{nK \times r} \quad \text{and} \quad \boldsymbol{U}_\mathcal{N} = \begin{bmatrix} \boldsymbol{u}_{r+1} & \boldsymbol{u}_{r+2} & \dots & \boldsymbol{u}_{nK} \end{bmatrix} \in \mathbb{R}^{nK \times (nK-r)}.$$

to denote the basis matrices for the information and nuisance subspaces from Definition 5.2. Similarly, we define the information and nuisance spectrum as

$$\boldsymbol{\lambda}_\mathcal{I} = \begin{bmatrix} \lambda_1 & \lambda_2 & \dots & \lambda_r \end{bmatrix}^T \quad \text{and} \quad \boldsymbol{\lambda}_\mathcal{N} = \begin{bmatrix} \lambda_{r+1} & \lambda_{r+2} & \dots & \lambda_{nK} \end{bmatrix}^T.$$

We also define the diagonal matrices

$$\boldsymbol{\Lambda} = \mathrm{diag}(\boldsymbol{\lambda}), \quad \boldsymbol{\Lambda}_\mathcal{I} = \mathrm{diag}(\boldsymbol{\lambda}_\mathcal{I}), \quad \text{and} \quad \boldsymbol{\Lambda}_\mathcal{N} = \mathrm{diag}(\boldsymbol{\lambda}_\mathcal{N}).$$

**Definition 6.1 (early stopping value and distance)** *Consider Definition 5.2 and let $\Gamma > 0$ be a positive scalar. Associated with the initial residual $\boldsymbol{r}_0 = f(\boldsymbol{\theta}_0) - \boldsymbol{y}$ and the information/nuisance subspaces of the reference Jacobian $\boldsymbol{J}$ (with a cut-off level $\alpha$) we define the $(\alpha, \Gamma)$ early stopping value as*

$$\mathcal{B}_{\alpha,\Gamma} = \left( \sum_{s=1}^r \frac{\alpha^2}{\lambda_s^2} \left( \langle \boldsymbol{u}_s, \boldsymbol{r}_0 \rangle \right)^2 + \Gamma^2 \sum_{s=r+1}^{nK} \frac{\lambda_s^2}{\alpha^2} \left( \langle \boldsymbol{u}_s, \boldsymbol{r}_0 \rangle \right)^2 \right)^{1/2}. \tag{6.1}$$

*We also define the early stopping distance as*

$$\mathcal{D}_{\alpha,\Gamma} = \frac{\mathcal{B}_{\alpha,\Gamma}}{\alpha}.$$

The goal of early stopping value/distance is understanding the behavior of the algorithm at a particular stopping time that depends on $\Gamma$ and the spectrum cutoff $\alpha$. In particular, as we will see later on the early stopping distance characterizes the distance from initialization at an appropriate early stopping time. We continue by stating and proving a few simple lemmas. The first Lemma provides upper/lower bounds on the early stopping value.

**Lemma 6.2 (Bounds on Early-Stopping Value)** *The early stopping value $\mathcal{B}_{\alpha,\Gamma}$ from Definition 6.1 obeys*

$$\mathcal{B}_{\alpha,\Gamma} \le \left( \|\Pi_\mathcal{I}(\boldsymbol{r}_0)\|_{\ell_2}^2 + \Gamma^2 \|\Pi_\mathcal{N}(\boldsymbol{r}_0)\|_{\ell_2}^2 \right)^{1/2} \le \Gamma \|\boldsymbol{r}_0\|_{\ell_2} \tag{6.2}$$

$$\mathcal{B}_{\alpha,\Gamma} \ge \frac{\alpha}{\lambda_1} \|\Pi_\mathcal{I}(\boldsymbol{r}_0)\|_{\ell_2}. \tag{6.3}$$

**Proof** To prove the upper bound we use the fact that $\alpha \le \lambda_s$ for $s \le r$ and $\alpha \ge \lambda_s$ for $s \ge r$ to conclude that

$$\mathcal{B}_{\alpha,\Gamma} \le \left( \sum_{s=1}^r \left( \langle \boldsymbol{u}_s, \boldsymbol{r}_0 \rangle \right)^2 + \Gamma^2 \sum_{s=r+1}^{nK} \left( \langle \boldsymbol{u}_s, \boldsymbol{r}_0 \rangle \right)^2 \right)^{1/2}$$

$$= \left( \|\Pi_\mathcal{I}(\boldsymbol{r}_0)\|_{\ell_2}^2 + \Gamma^2 \|\Pi_\mathcal{N}(\boldsymbol{r}_0)\|_{\ell_2}^2 \right)^{1/2}$$

$$\le \Gamma \|\boldsymbol{r}_0\|_{\ell_2}.$$

To prove the lower bound, we use the facts that $\alpha^2/\lambda_s^2 \ge \alpha^2/\lambda_1^2$ to conclude that

$$\mathcal{B}_{\alpha,\Gamma} = \left( \sum_{s=1}^r \frac{\alpha^2}{\lambda_s^2} \left( \langle \boldsymbol{u}_s, \boldsymbol{r}_0 \rangle \right)^2 + \Gamma^2 \sum_{s=r+1}^{nK} \frac{\lambda_s^2}{\alpha^2} \left( \langle \boldsymbol{u}_s, \boldsymbol{r}_0 \rangle \right)^2 \right)^{1/2},$$

$$\ge \left( \sum_{s=1}^r \frac{\alpha^2}{\lambda_s^2} \left( \langle \boldsymbol{u}_s, \boldsymbol{r}_0 \rangle \right)^2 \right)^{1/2},$$

$$\ge \frac{\alpha}{\lambda_1} \|\Pi_\mathcal{I}(\boldsymbol{r}_0)\|_{\ell_2}.$$

■

It is of course well known that the mapping $\left( \boldsymbol{I} - \eta \boldsymbol{A}\boldsymbol{A}^T \right)$ is a contraction for sufficiently small values of $\eta$. The next lemma shows that if we replace one of the $\boldsymbol{A}$ matrices with a matrix $\boldsymbol{B}$ which is close to $\boldsymbol{A}$ the resulting matrix $\left( \boldsymbol{I} - \eta \boldsymbol{A}\boldsymbol{B}^T \right)$, while may not be contractive, is not too expansive.

**Lemma 6.3 (Asymmetric PSD increase)** *Let $A, B \in \mathbb{R}^{n \times p}$ be matrices obeying*

$$\|A\| \le \beta, \quad \|B\| \le \beta, \quad \text{and} \quad \|B - A\| \le \varepsilon.$$

*Then, for all $r \in \mathbb{R}^n$ and $\eta \le 1/\beta^2$ we have*

$$\left\|\left(I - \eta A B^T\right) r\right\|_{\ell_2} \le \left(1 + \eta \varepsilon^2\right) \|r\|_{\ell_2}.$$

**Proof** Note that using $\eta \le 1/\beta^2$ and $\|B - A\| \le \varepsilon$ we conclude that

$$
\begin{aligned}
\left\|\left(I - \eta A B^T\right) r\right\|_{\ell_2}^2 &= \left\|\left(I - \eta B B^T - \eta (A - B) B^T\right) r\right\|_{\ell_2}^2 \\
&= \left\|r - \eta (A - B + B) B^T r\right\|_{\ell_2}^2 \\
&= \|r\|_{\ell_2}^2 - 2\eta r^T (A - B + B) B^T r + \eta^2 \left\|A B^T r\right\|_{\ell_2}^2 \\
&\le \|r\|_{\ell_2}^2 - 2\eta \|B^T r\|_{\ell_2}^2 + 2\eta \left\|(A - B)^T r\right\|_{\ell_2} \|B^T r\|_{\ell_2} + \eta^2 \|A\|^2 \|B^T r\|_{\ell_2}^2 \\
&= \|r\|_{\ell_2}^2 - \eta \|B^T r\|_{\ell_2}^2 + 2\eta \left\|(A - B)^T r\right\|_{\ell_2} \|B^T r\|_{\ell_2} + \left(\eta^2 \|A\|^2 \|B^T r\|_{\ell_2}^2 - \eta \|B^T r\|_{\ell_2}^2\right) \\
&\overset{\eta \le 1/\beta^2}{\le} \|r\|_{\ell_2}^2 - \eta \|B^T r\|_{\ell_2}^2 + 2\eta \left\|(A - B)^T r\right\|_{\ell_2} \|B^T r\|_{\ell_2} \\
&\overset{\|A - B\| \le \epsilon}{\le} \|r\|_{\ell_2}^2 - \eta \|B^T r\|_{\ell_2}^2 + 2\eta\varepsilon \|B^T r\|_{\ell_2} \|r\|_{\ell_2} \\
&= (1 + \eta\varepsilon^2) \|r\|_{\ell_2}^2 - \eta \left(\varepsilon \|r\|_{\ell_2} - \|B^T r\|_{\ell_2}\right)^2 \\
&\le (1 + \eta\varepsilon^2) \|r\|_{\ell_2}^2,
\end{aligned}
$$

completing the proof. ∎

The next lemma shows that if two PSD matrices are close to each other then an appropriate square root of these matrices will also be close.

**Lemma 6.4** *Let $A$ and $B$ be $n \times n$ positive semi-definite matrices satisfying $\|A - B\| \le \alpha^2$ for a scalar $\alpha \ge 0$. Then for any $X \in \mathbb{R}^{n \times p}$ with $p \ge n$ obeying $A = X X^T$, there exists a matrix $Y \in \mathbb{R}^{n \times p}$ obeying $B = Y Y^T$ such that*

$$\|Y - X\| \le 2\alpha$$

**Proof** First we note that for any two PSD matrices $A_+, B_+ \in \mathbb{R}^{n \times n}$ obeying $A_+, B_+ \ge \frac{\alpha^2}{4} I_n$, Lemma 2.2 of Schmitt (1992) guarantees that

$$\left\|A_+^{1/2} - B_+^{1/2}\right\| \le \frac{\|A_+ - B_+\|}{\alpha}.$$

In the above for a PSD matrix $A \in \mathbb{R}^{n \times n}$ with an eigenvalue decomposition $A = U \Lambda U^T$ we use $A^{1/2}$ to denote the square root of the matrix given by $A = U \Lambda^{1/2} U^T$. We shall use this result with $A_+ = A + \frac{\alpha^2}{4} I_n$ and $B_+ = B + \frac{\alpha^2}{4} I_n$ to conclude that

$$\left\|A_+^{1/2} - B_+^{1/2}\right\| \le \frac{\|A_+ - B_+\|}{\alpha} = \frac{\|A - B\|}{\alpha}.$$

Furthermore, using the fact that the eigenvalues of $A_+$ and $B_+$ are just shifted versions of the eigenvalues of $A$ and $B$ by $\alpha^2/4$ we can conclude that

$$\left\|A_+^{1/2} - A^{1/2}\right\| \le \frac{\alpha}{2} \quad \text{and} \quad \left\|B_+^{1/2} - B^{1/2}\right\| \le \frac{\alpha}{2}.$$

Combining the latter two inequalities with the assumption that $\|A - B\| \le \alpha^2$ we conclude that

$$
\begin{aligned}
\left\|A^{1/2} - B^{1/2}\right\| &\le \left\|A_+^{1/2} - B_+^{1/2}\right\| + \left\|A_+^{1/2} - A^{1/2}\right\| + \left\|B_+^{1/2} - B^{1/2}\right\| \\
&\le \frac{\|A - B\|}{\alpha} + \frac{\alpha}{2} + \frac{\alpha}{2} \\
&\le 2\alpha.
\end{aligned}
\tag{6.4}
$$

Suppose $p \geq n$ and assume the matrices $A$ and $B$ have eigenvalue decompositions given by $A = U_A \Lambda_A U_A^T$ and $B = U_B \Lambda_B U_B^T$. Then, any $X \in \mathbb{R}^{n \times p}$ with $p \geq n$ has the form $X = U_A \Lambda_A^{1/2} V_A^T$ with $V_A \in \mathbb{R}^{p \times n}$ an orthonormal matrix. Now pick

$$Y = U_B \Lambda_B^{1/2} U_B^T U_A V_A^T.$$

Then clearly $YY^T = B$. Furthermore, we have

$$
\begin{aligned}
\|X - Y\| &= \left\| U_A \Lambda_A^{1/2} V_A^T - U_B \Lambda_B^{1/2} U_B^T U_A V_A^T \right\| \\
&= \left\| U_A \Lambda_A^{1/2} U_A^T U_A V_A^T - U_B \Lambda_B^{1/2} U_B^T U_A V_A^T \right\| \\
&= \left\| \left( U_A \Lambda_A^{1/2} U_A^T - U_B \Lambda_B^{1/2} U_B^T \right) U_A V_A^T \right\| \\
&= \left\| \left( A^{1/2} - B^{1/2} \right) U_A V_A^T \right\| \\
&= \left\| A^{1/2} - B^{1/2} \right\|.
\end{aligned}
$$

Combining the latter with (6.4) completes the proof. ∎

### 6.1.2 KEY LEMMAS FOR GENERAL NONLINEARITIES

Throughout this section we assume $J$ is the reference Jacobian per Definition 5.1 with eigenvalue decomposition $J = U \Lambda V^T = \sum_{s=1}^{Kn} \lambda_s u_s v_s^T$ with $\Lambda = \text{diag}(\lambda)$. We also define $a = U^T r_0 = U^T \widetilde{r}_0 \in \mathbb{R}^{nK}$ be the coefficients of the initial residual in the span of the column space of this reference Jacobian.

We shall first characterize the evolution of the linearized parameter $\widetilde{\theta}_\tau$ and residual $\widetilde{r}_\tau$ vectors from (5.11) in the following lemma.

**Lemma 6.5** *The linearized residual vector $\widetilde{r}_\tau$ can be written in the form*

$$\widetilde{r}_\tau = U \left( I - \eta \Lambda^2 \right)^\tau a = \sum_{s=1}^{nK} (1 - \eta \lambda_s^2)^\tau a_s u_s. \tag{6.5}$$

*Furthermore, assuming $\eta \leq 1/\lambda_1^2$ the linear updates $\widetilde{\theta}_\tau$ obey*

$$\| \widetilde{\theta}_\tau - \widetilde{\theta}_0 \|_{\ell_2}^2 \leq \sum_{s=1}^{r} \frac{a_s^2}{\lambda_s^2} + \tau^2 \eta^2 \sum_{s=r+1}^{nK} \lambda_s^2 a_s^2. \tag{6.6}$$

**Proof** Using the fact that $JJ^T = U \Lambda^2 U^T$ we have

$$\left( I - \eta J J^T \right)^\tau = U \left( I - \eta \Lambda^2 \right)^\tau U^T$$

Using the latter combined with (5.11) we thus have

$$
\begin{aligned}
\widetilde{r}_\tau &= (I - \eta J J^T)^\tau r_0, \\
&= U \left( I - \eta \Lambda^2 \right)^\tau U^T r_0, \\
&= U \left( I - \eta \Lambda^2 \right)^\tau a, \\
&= \sum_{s=1}^{nK} (1 - \eta \lambda_s^2)^\tau a_s u_s,
\end{aligned}
$$

completing the proof of (6.5).

We now turn our attention to proving (6.6) by tracking the representation of $\widetilde{\theta}_\tau$ in terms of the right singular vectors of $J$. To do this note that using (6.5) we have

$$J^T \widetilde{r}_t = V \Lambda U^T \widetilde{r}_t = V \Lambda \left( I - \eta \Lambda^2 \right)^t a.$$

Using the latter together with the gradient update on the linearized problem we have

$$\widetilde{\theta}_\tau - \widetilde{\theta}_0 = -\eta \left( \sum_{t=0}^{\tau-1} \nabla \mathcal{L}_{lin}(\widetilde{\theta}_t) \right) = -\eta \left( \sum_{t=0}^{\tau-1} J^T \widetilde{r}_t \right) = -\eta V \left( \sum_{t=0}^{\tau-1} \Lambda \left( I - \eta \Lambda^2 \right)^t \right) a.$$

Thus for any $s \in \{1, 2, \ldots, nK\}$

$$\boldsymbol{v}_s^T \left( \widetilde{\boldsymbol{\theta}}_\tau - \widetilde{\boldsymbol{\theta}}_0 \right) = -\eta \lambda_s \boldsymbol{a}_s \left( \sum_{t=0}^{\tau-1} \left( 1 - \eta \lambda_s^2 \right)^t \right) = -\eta \lambda_s \boldsymbol{a}_s \frac{1 - \left( 1 - \eta \lambda_s^2 \right)^\tau}{\eta \lambda_s^2} = -\boldsymbol{a}_s \frac{1 - \left( 1 - \eta \lambda_s^2 \right)^\tau}{\lambda_s}.$$

Noting that for $\eta \le 1/\lambda_1^2 \le 1/\lambda_s^2$ we have $1 - \eta \lambda_s^2 \ge 0$, the latter identity implies that

$$\left| \boldsymbol{v}_s^T \left( \widetilde{\boldsymbol{\theta}}_\tau - \widetilde{\boldsymbol{\theta}}_0 \right) \right| \le \frac{|\boldsymbol{a}_s|}{\lambda_s}. \tag{6.7}$$

Furthermore, using the fact that $1 - \eta \lambda_s^2 \le 1$ we have

$$\left| \boldsymbol{v}_s^T \left( \widetilde{\boldsymbol{\theta}}_\tau - \widetilde{\boldsymbol{\theta}}_0 \right) \right| = \eta \lambda_s |\boldsymbol{a}_s| \left( \sum_{t=0}^{\tau-1} \left( 1 - \eta \lambda_s^2 \right)^t \right) \le \eta \lambda_s |\boldsymbol{a}_s| \tau \tag{6.8}$$

Combining (6.7) for $1 \le s \le r$ and (6.8) for $s > r$ we have

$$\left\| \widetilde{\boldsymbol{\theta}}_\tau - \widetilde{\boldsymbol{\theta}}_0 \right\|_{\ell_2}^2 = \sum_{s=1}^{nK} \left| \boldsymbol{v}_s^T \left( \widetilde{\boldsymbol{\theta}}_\tau - \widetilde{\boldsymbol{\theta}}_0 \right) \right|^2 \le \sum_{s=1}^r \frac{a_s^2}{\lambda_s^2} + \tau^2 \eta^2 \sum_{s=r+1}^{nK} \lambda_s^2 a_s^2,$$

completing the proof of (6.6). ∎

For future use we also state a simple corollary of the above Lemma below.

**Corollary 6.6** *Consider the setting and assumptions of Lemma 6.5. Then, after $\tau$ iterations we have*

$$\| \widetilde{\boldsymbol{r}}_\tau \|_{\ell_2} \le \left( 1 - \eta \alpha^2 \right)^\tau \| \Pi_{\mathcal{I}}(\boldsymbol{r}_0) \|_{\ell_2} + \| \Pi_{\mathcal{N}}(\boldsymbol{r}_0) \|_{\ell_2}. \tag{6.9}$$

*Furthermore, after $T = \frac{\Gamma}{\eta \alpha^2}$ iterations we have*

$$\| \widetilde{\boldsymbol{r}}_T \|_{\ell_2} \le e^{-\Gamma} \| \Pi_{\mathcal{I}}(\boldsymbol{r}_0) \|_{\ell_2} + \| \Pi_{\mathcal{N}}(\boldsymbol{r}_0) \|_{\ell_2}. \tag{6.10}$$

*and*

$$\left\| \widetilde{\boldsymbol{\theta}}_T - \widetilde{\boldsymbol{\theta}}_0 \right\|_{\ell_2}^2 \le \sum_{s=1}^r \frac{a_s^2}{\lambda_s^2} + \Gamma^2 \sum_{s=r+1}^{nK} \frac{\lambda_s^2 a_s^2}{\alpha^4} = \frac{\mathcal{B}_{\alpha,\Gamma}^2}{\alpha^2}.$$

*with $\mathcal{B}_{\alpha,\Gamma}$ given by (6.1) per Definition 6.2.*

**Proof** To prove the first bound on the residual ((6.9))note that using (6.5) we have

$$\boldsymbol{U}_{\mathcal{I}}^T \widetilde{\boldsymbol{r}}_\tau = \left( \boldsymbol{I} - \eta \Lambda_{\mathcal{I}}^2 \right)^\tau \boldsymbol{U}_{\mathcal{I}}^T \widetilde{\boldsymbol{r}}_0 \quad \text{and} \quad \boldsymbol{U}_{\mathcal{N}}^T \widetilde{\boldsymbol{r}}_\tau = \left( \boldsymbol{I} - \eta \Lambda_{\mathcal{N}}^2 \right)^\tau \boldsymbol{U}_{\mathcal{N}}^T \widetilde{\boldsymbol{r}}_0$$

Thus, using the fact that for $s \le r$ we have $\lambda_s \ge \alpha$ we have $(1 - \eta \lambda_s^2)^\tau \le (1 - \eta \alpha^2)^\tau$ and for $s > r$ we have $(1 - \eta \lambda_s^2)^\tau \le 1$, we can conclude that

$$\left\| \boldsymbol{U}_{\mathcal{I}}^T \widetilde{\boldsymbol{r}}_\tau \right\|_{\ell_2} \le \left( 1 - \eta \alpha^2 \right)^\tau \left\| \boldsymbol{U}_{\mathcal{I}}^T \boldsymbol{r}_0 \right\|_{\ell_2} \quad \text{and} \quad \left\| \boldsymbol{U}_{\mathcal{N}}^T \widetilde{\boldsymbol{r}}_\tau \right\|_{\ell_2} \le \left\| \boldsymbol{U}_{\mathcal{N}}^T \boldsymbol{r}_0 \right\|_{\ell_2}.$$

Combining these with the triangular inequality we have

$$\| \widetilde{\boldsymbol{r}}_\tau \|_{\ell_2} = \left\| \begin{bmatrix} \boldsymbol{U}_{\mathcal{I}}^T \widetilde{\boldsymbol{r}}_\tau \\ \boldsymbol{U}_{\mathcal{N}}^T \widetilde{\boldsymbol{r}}_\tau \end{bmatrix} \right\|_{\ell_2} \le \left\| \boldsymbol{U}_{\mathcal{I}}^T \widetilde{\boldsymbol{r}}_\tau \right\|_{\ell_2} + \left\| \boldsymbol{U}_{\mathcal{N}}^T \widetilde{\boldsymbol{r}}_\tau \right\|_{\ell_2} \le \left( 1 - \eta \alpha^2 \right)^\tau \left\| \boldsymbol{U}_{\mathcal{I}}^T \boldsymbol{r}_0 \right\|_{\ell_2} + \left\| \boldsymbol{U}_{\mathcal{N}}^T \boldsymbol{r}_0 \right\|_{\ell_2},$$

concluding the proof of (6.9). The second bound on the residual simply follows from the fact that $(1 - \eta \alpha^2)^T \le e^{-\Gamma}$. The bound on $\left\| \widetilde{\boldsymbol{\theta}}_T - \widetilde{\boldsymbol{\theta}}_0 \right\|_{\ell_2}^2$ is trivially obtained by using $T^2 = \frac{\Gamma^2}{\eta^2 \alpha^4}$ in (6.6). ∎

The lemma above shows that with enough iterations, gradient descent on the linearized problem fits the residual over the information space and the residual is (in the worst case) unchanged over the nuisance subspace $\mathcal{N}$. Our hypothesis is that, when the model is generalizable the residual mostly lies on the information space $\mathcal{I}$ which contains the directions aligned with the top singular vectors. Hence, the smaller term $\|\Pi_{\mathcal{N}}(\boldsymbol{r}_0)\|_{\ell_2}$ over the nuisance space will not affect generalization significantly. To make this intuition precise however we need to connect the residual of the original problem to that of the linearized problem. The following lemma sheds light on the evolution of the original problem (5.6) by characterizing the evolution of the difference between the residuals of the original and linearized problems from one iteration to the next.

**Lemma 6.7 (Keeping track of perturbation - one step)** *Assume Assumptions 1 and 2 hold and $\boldsymbol{\theta}_\tau$ and $\boldsymbol{\theta}_{\tau+1}$ are within an $R$ neighborhood of $\boldsymbol{\theta}_0$, that is,*

$$\|\boldsymbol{\theta}_\tau - \boldsymbol{\theta}_0\|_{\ell_2} \leq R \quad and \quad \|\boldsymbol{\theta}_{\tau+1} - \boldsymbol{\theta}_0\|_{\ell_2} \leq R.$$

*Then with a learning rate obeying $\eta \leq 1/\beta^2$, the deviation in the residuals of the original and linearized problems $\boldsymbol{e}_{\tau+1} = \boldsymbol{r}_{\tau+1} - \widetilde{\boldsymbol{r}}_{\tau+1}$ obey*

$$\|\boldsymbol{e}_{\tau+1}\|_{\ell_2} \leq \eta(\varepsilon_0^2 + \varepsilon\beta)\|\widetilde{\boldsymbol{r}}_\tau\|_{\ell_2} + (1 + \eta\varepsilon^2)\|\boldsymbol{e}_\tau\|_{\ell_2}. \tag{6.11}$$

**Proof** For simplicity, denote $\boldsymbol{B}_1 = \mathcal{J}(\boldsymbol{\theta}_{\tau+1}, \boldsymbol{\theta}_\tau)$, $\boldsymbol{B}_2 = \mathcal{J}(\boldsymbol{\theta}_\tau)$, $\boldsymbol{A} = \mathcal{J}(\boldsymbol{\theta}_0)$ where

$$\mathcal{J}(\boldsymbol{b}, \boldsymbol{a}) = \int_0^1 \mathcal{J}(t\boldsymbol{b} + (1-t)\boldsymbol{a})dt.$$

We can write the predictions due to $\boldsymbol{\theta}_{\tau+1}$ as

$$f(\boldsymbol{\theta}_{\tau+1}) = f(\boldsymbol{\theta}_\tau - \eta\nabla\mathcal{L}(\boldsymbol{\theta}_\tau)) = f(\boldsymbol{\theta}_\tau) + \eta\mathcal{J}(\boldsymbol{\theta}_{\tau+1}, \boldsymbol{\theta}_\tau)\nabla\mathcal{L}(\boldsymbol{\theta}_\tau)$$
$$= f(\boldsymbol{\theta}_\tau) + \eta\mathcal{J}(\boldsymbol{\theta}_{\tau+1}, \boldsymbol{\theta}_\tau)\mathcal{J}^T(\boldsymbol{\theta}_\tau)(f(\boldsymbol{\theta}_\tau) - \boldsymbol{y}).$$

This implies that

$$\boldsymbol{r}_{\tau+1} = f(\boldsymbol{\theta}_{\tau+1}) - \boldsymbol{y} = (\boldsymbol{I} - \eta\boldsymbol{B}_1\boldsymbol{B}_2^T)\boldsymbol{r}_\tau.$$

Similarly, for linearized problem we have $\widetilde{\boldsymbol{r}}_{\tau+1} = (\boldsymbol{I} - \eta\boldsymbol{J}\boldsymbol{J}^T)\widetilde{\boldsymbol{r}}_\tau$. Thus,

$$\begin{aligned}
\|\boldsymbol{e}_{\tau+1}\|_{\ell_2} &= \left\|(\boldsymbol{I} - \eta\boldsymbol{B}_1\boldsymbol{B}_2^T)\boldsymbol{r}_\tau - (\boldsymbol{I} - \eta\boldsymbol{J}\boldsymbol{J}^T)\widetilde{\boldsymbol{r}}_\tau\right\|_{\ell_2} \\
&= \left\|(\boldsymbol{I} - \eta\boldsymbol{B}_1\boldsymbol{B}_2^T)\boldsymbol{e}_\tau - \eta(\boldsymbol{B}_1\boldsymbol{B}_2^T - \boldsymbol{J}\boldsymbol{J}^T)\widetilde{\boldsymbol{r}}_\tau\right\|_{\ell_2} \\
&\leq \left\|(\boldsymbol{I} - \eta\boldsymbol{B}_1\boldsymbol{B}_2^T)\boldsymbol{e}_\tau\right\|_{\ell_2} + \eta\left\|(\boldsymbol{B}_1\boldsymbol{B}_2^T - \boldsymbol{J}\boldsymbol{J}^T)\widetilde{\boldsymbol{r}}_\tau\right\|_{\ell_2} \\
&\leq \left\|(\boldsymbol{I} - \eta\boldsymbol{B}_1\boldsymbol{B}_2^T)\boldsymbol{e}_\tau\right\|_{\ell_2} + \eta\left\|(\boldsymbol{B}_1\boldsymbol{B}_2^T - \boldsymbol{J}\boldsymbol{J}^T)\right\|\|\widetilde{\boldsymbol{r}}_\tau\|_{\ell_2}. \tag{6.12}
\end{aligned}$$

We proceed by bounding each of these two terms. For the first term, we apply Lemma 6.3 with $\boldsymbol{A} = \boldsymbol{B}_1$ and $\boldsymbol{B} = \boldsymbol{B}_2$ and use $\|\boldsymbol{B}_1 - \boldsymbol{B}_2\| \leq \varepsilon$ to conclude that

$$\left\|(\boldsymbol{I} - \eta\boldsymbol{B}_1\boldsymbol{B}_2^T)\boldsymbol{e}_\tau\right\|_{\ell_2} \leq (1 + \eta\varepsilon^2)\|\boldsymbol{e}_\tau\|_{\ell_2}. \tag{6.13}$$

Next we turn our attention to bounding the second term. To this aim note that

$$\begin{aligned}
\|\boldsymbol{B}_1\boldsymbol{B}_2^T - \boldsymbol{J}\boldsymbol{J}^T\| &= \|\boldsymbol{B}_1\boldsymbol{B}_2^T - \boldsymbol{A}\boldsymbol{A}^T + \boldsymbol{A}\boldsymbol{A}^T - \boldsymbol{J}\boldsymbol{J}^T\| \\
&\leq \|\boldsymbol{B}_1\boldsymbol{B}_2^T - \boldsymbol{A}\boldsymbol{A}^T\| + \|\boldsymbol{A}\boldsymbol{A}^T - \boldsymbol{J}\boldsymbol{J}^T\| \\
&\leq \|(\boldsymbol{B}_1 - \boldsymbol{A})\boldsymbol{B}_2^T\| + \|\boldsymbol{A}(\boldsymbol{B}_2 - \boldsymbol{A})^T\| + \|\boldsymbol{A}\boldsymbol{A}^T - \boldsymbol{J}\boldsymbol{J}^T\| \\
&\leq \|\boldsymbol{B}_1 - \boldsymbol{A}\|\|\boldsymbol{B}_2\| + \|\boldsymbol{B}_2 - \boldsymbol{A}\|\|\boldsymbol{A}\| + \|\boldsymbol{A}\boldsymbol{A}^T - \boldsymbol{J}\boldsymbol{J}^T\| \\
&\leq \beta\frac{\varepsilon}{2} + \beta\frac{\varepsilon}{2} + \varepsilon_0^2 \\
&= \varepsilon_0^2 + \varepsilon\beta. \tag{6.14}
\end{aligned}$$

In the last inequality we use the fact that per Assumption 2 we have $\|\boldsymbol{B}_1 - \boldsymbol{A}\| \leq \varepsilon/2$ and $\|\boldsymbol{B}_2 - \boldsymbol{A}\| \leq \varepsilon/2$ as well as the fact that per Definition 5.1 $\|\boldsymbol{A}\boldsymbol{A}^T - \boldsymbol{J}\boldsymbol{J}^T\| \leq \varepsilon_0^2$. Plugging (6.13) and (6.14) in (6.12) completes the proof. ∎

Next we prove a result about the growth of sequences obeying certain assumptions. As we will see later on in the proofs this lemma allows us to control the growth of the perturbation between the original and linearized residuals ($e_\tau = \|\boldsymbol{e}_\tau\|_{\ell_2}$).

**Lemma 6.8 (Bounding residual perturbation growth for general nonlinearities)** *Consider positive scalars $\Gamma, \alpha, \varepsilon, \eta > 0$. Also assume $\eta \leq 1/\alpha^2$ and $\alpha \geq \sqrt{2\Gamma}\varepsilon$ and set $T = \frac{\Gamma}{\eta\alpha^2}$. Assume the scalar sequences $e_\tau$ (with $e_0 = 0$) and $\widetilde{r}_\tau$ obey the following identities*

$$\begin{aligned}
\widetilde{r}_\tau &\leq (1 - \eta\alpha^2)^\tau \rho_+ + \rho_-, \\
e_\tau &\leq (1 + \eta\varepsilon^2)e_{\tau-1} + \eta\Theta\widetilde{r}_{\tau-1}, \tag{6.15}
\end{aligned}$$

*for all $0 \leq \tau \leq T$ and non-negative values $\rho_-, \rho_+ \geq 0$. Then, for all $0 \leq \tau \leq T$,*

$$e_\tau \leq \Theta\Lambda \quad holds \ with \quad \Lambda = \frac{2(\Gamma\rho_- + \rho_+)}{\alpha^2}. \tag{6.16}$$

**Proof** We shall prove the result inductively. Suppose (6.16) holds for all $t \le \tau - 1$. Consequently, we have

$$
\begin{aligned}
e_{t+1} &\le (1 + \eta \varepsilon^2) e_t + \eta \Theta \widetilde{r}_t \\
&\le e_t + \eta \varepsilon^2 e_t + \eta \Theta \left( (1 - \eta \alpha^2)^t \rho_+ + \rho_- \right) \\
&\le e_t + \eta \Theta \left( \varepsilon^2 \Lambda + (1 - \eta \alpha^2)^t \rho_+ + \rho_- \right).
\end{aligned}
$$

Thus

$$
\frac{e_{t+1} - e_t}{\Theta} \le \eta \left( \varepsilon^2 \Lambda + (1 - \eta \alpha^2)^t \rho_+ + \rho_- \right). \tag{6.17}
$$

Summing up both sides of (6.17) for $0 \le t \le \tau - 1$ we conclude that

$$
\begin{aligned}
\frac{e_\tau}{\Theta} &= \sum_{t=0}^{\tau-1} \frac{e_{t+1} - e_t}{\Theta} \\
&\le \eta \tau \left( \varepsilon^2 \Lambda + \rho_- \right) + \eta \rho_+ \sum_{t=0}^{\tau-1} (1 - \eta \alpha^2)^t \\
&= \eta \tau \left( \varepsilon^2 \Lambda + \rho_- \right) + \eta \rho_+ \frac{1 - \left( 1 - \eta \alpha^2 \right)^\tau}{\eta \alpha^2} \\
&\le \eta \left( \tau \varepsilon^2 \Lambda + \frac{\rho_+}{\eta \alpha^2} + \tau \rho_- \right) \\
&= \eta \tau (\varepsilon^2 \Lambda + \rho_-) + \frac{\rho_+}{\alpha^2} \\
&\le \eta T (\varepsilon^2 \Lambda + \rho_-) + \frac{\rho_+}{\alpha^2} \\
&= \frac{\Gamma \varepsilon^2 \Lambda + \Gamma \rho_- + \rho_+}{\alpha^2} \\
&= \frac{\Gamma \varepsilon^2 \Lambda}{\alpha^2} + \frac{\Lambda}{2} \\
&\le \Lambda,
\end{aligned}
$$

where in the last inequality we used the fact that $\alpha^2 \ge 2\Gamma \varepsilon^2$. This completes the proof of the induction step and the proof of the lemma. ∎

### 6.1.3 COMPLETING THE PROOF OF THEOREM 5.3

With the key lemmas in place in this section we wish to complete the proof of Theorem 5.3. We will use induction to prove the result. Suppose the statement is true for some $\tau - 1 \le T - 1$. In particular, we assume the identities (5.14) and (5.15) hold for all $0 \le t \le \tau - 1$. We aim to prove these identities continue to hold for iteration $\tau$. We will prove this result in multiple steps.

**Step I: Next iterate obeys $\|\boldsymbol{\theta}_\tau - \boldsymbol{\theta}_0\|_{\ell_2} \le R$.**
We first argue that $\boldsymbol{\theta}_\tau$ lies in the domain of interest as dictated by (5.12), i.e. $\|\boldsymbol{\theta}_\tau - \boldsymbol{\theta}_0\|_{\ell_2} \le R$. To do this note that per the induction assumption (5.15) holds for iteration $\tau - 1$ and thus $\|\boldsymbol{\theta}_{\tau-1} - \boldsymbol{\theta}_0\|_{\ell_2} \le R/2$. As a result using the triangular inequality to show $\|\boldsymbol{\theta}_\tau - \boldsymbol{\theta}_0\|_{\ell_2} \le R$ holds it suffices to show

that $\|\boldsymbol{\theta}_\tau - \boldsymbol{\theta}_{\tau-1}\|_{\ell_2} \le R/2$ holds. To do this note that

$$
\begin{aligned}
\|\boldsymbol{\theta}_\tau - \boldsymbol{\theta}_{\tau-1}\|_{\ell_2} &= \eta \|\nabla \mathcal{L}(\boldsymbol{\theta}_{\tau-1})\|_{\ell_2} \\
&= \eta \left\| \mathcal{J}^T(\boldsymbol{\theta}_{\tau-1}) \boldsymbol{r}_{\tau-1} \right\|_{\ell_2} \\
&= \eta \left\| \overline{\mathcal{J}}^T(\boldsymbol{\theta}_{\tau-1}) \boldsymbol{r}_{\tau-1} \right\|_{\ell_2} \\
&\overset{(a)}{\le} \eta \| \overline{\mathcal{J}}^T(\boldsymbol{\theta}_{\tau-1}) \widetilde{\boldsymbol{r}}_{\tau-1}\|_{\ell_2} + \eta \| \overline{\mathcal{J}}^T(\boldsymbol{\theta}_{\tau-1})(\boldsymbol{r}_{\tau-1} - \widetilde{\boldsymbol{r}}_{\tau-1})\|_{\ell_2} \\
&\overset{(b)}{\le} \eta \left\| \boldsymbol{J}^T \widetilde{\boldsymbol{r}}_{\tau-1} \right\|_{\ell_2} + \eta \left\| \overline{\mathcal{J}}(\boldsymbol{\theta}_{\tau-1}) - \boldsymbol{J} \right\| \|\widetilde{\boldsymbol{r}}_{\tau-1}\|_{\ell_2} + \eta \left\| \overline{\mathcal{J}}(\boldsymbol{\theta}_{\tau-1}) \right\| \|\boldsymbol{r}_{\tau-1} - \widetilde{\boldsymbol{r}}_{\tau-1}\|_{\ell_2} \\
&\overset{(c)}{\le} \eta \left\| \boldsymbol{J}^T \widetilde{\boldsymbol{r}}_{\tau-1} \right\|_{\ell_2} + \frac{\varepsilon_0 + \varepsilon}{\beta^2} \|\widetilde{\boldsymbol{r}}_{\tau-1}\|_{\ell_2} + \frac{1}{\beta} \|\boldsymbol{r}_{\tau-1} - \widetilde{\boldsymbol{r}}_{\tau-1}\|_{\ell_2} \\
&\overset{(d)}{\le} \eta \|\boldsymbol{J}^T \widetilde{\boldsymbol{r}}_{\tau-1}\|_{\ell_2} + \frac{2\delta\alpha}{5\beta^2} \|\boldsymbol{r}_0\|_{\ell_2} + \frac{1}{\beta} \|\boldsymbol{r}_{\tau-1} - \widetilde{\boldsymbol{r}}_{\tau-1}\|_{\ell_2} \\
&\overset{(e)}{\le} \eta \|\boldsymbol{J}^T \widetilde{\boldsymbol{r}}_{\tau-1}\|_{\ell_2} + \frac{2\delta\alpha}{5\beta^2} \|\boldsymbol{r}_0\|_{\ell_2} + \frac{3\delta\alpha}{5\beta^2} \|\boldsymbol{r}_0\|_{\ell_2} \\
&= \eta \|\boldsymbol{J}^T \widetilde{\boldsymbol{r}}_{\tau-1}\|_{\ell_2} + \frac{\delta\alpha}{\beta^2} \|\boldsymbol{r}_0\|_{\ell_2} \\
&\overset{(f)}{\le} \eta \beta^2 \frac{\mathcal{B}_{\alpha,\Gamma}}{\alpha} + \frac{\delta\alpha}{\beta^2} \|\boldsymbol{r}_0\|_{\ell_2} \\
&\overset{(g)}{\le} \frac{\mathcal{B}_{\alpha,\Gamma}}{\alpha} + \frac{\delta\alpha}{\beta^2} \|\boldsymbol{r}_0\|_{\ell_2} \\
&\overset{(h)}{\le} \frac{\mathcal{B}_{\alpha,\Gamma}}{\alpha} + \frac{\delta\Gamma}{\alpha} \|\boldsymbol{r}_0\|_{\ell_2} \\
&= \frac{R}{2}.
\end{aligned}
$$

Here, (a) and (b) follow from a simple application of the triangular inequality, (c) from the fact that $\|\mathcal{J}(\boldsymbol{\theta}_{\tau-1}) - \boldsymbol{J}\| \le \|\mathcal{J}(\boldsymbol{\theta}_{\tau-1}) - \mathcal{J}(\boldsymbol{\theta}_0)\| + \|\mathcal{J}(\boldsymbol{\theta}_0) - \boldsymbol{J}\| \le \varepsilon + \varepsilon_0$, (d) from combining the bounds in (5.13), (e) from the induction hypothesis that postulates (5.14) holds for iteration $\tau - 1$, (f) from considering the SVD $\boldsymbol{J} = \boldsymbol{U}\boldsymbol{\Lambda}\boldsymbol{V}^T$ which implies that

$$
\begin{aligned}
\left\| \boldsymbol{J}^T \widetilde{\boldsymbol{r}}_{\tau-1} \right\|_{\ell_2}^2 &= \left\| \boldsymbol{J}^T \left( \boldsymbol{I} - \eta \boldsymbol{J}\boldsymbol{J}^T \right)^{\tau-1} \boldsymbol{r}_0 \right\|_{\ell_2}^2 = \left\| \boldsymbol{V}\boldsymbol{\Lambda} \left( \boldsymbol{I} - \eta\boldsymbol{\Lambda}^2 \right)^{\tau-1} \boldsymbol{U}^T \boldsymbol{r}_0 \right\|_{\ell_2}^2 \\
&= \left\| \boldsymbol{\Lambda} \left( \boldsymbol{I} - \eta\boldsymbol{\Lambda}^2 \right)^{\tau-1} \boldsymbol{U}^T \boldsymbol{r}_0 \right\|_{\ell_2}^2 \\
&= \sum_{s=1}^{nK} \lambda_s^2 (1 - \eta\lambda_s^2)^{2(\tau-1)} (\langle \boldsymbol{u}_s, \boldsymbol{r}_0 \rangle)^2 \\
&\le \sum_{s=1}^{nK} \lambda_s^2 (\langle \boldsymbol{u}_s, \boldsymbol{r}_0 \rangle)^2 \\
&= \sum_{s=1}^{r} \lambda_s^2 (\langle \boldsymbol{u}_s, \boldsymbol{r}_0 \rangle)^2 + \sum_{s=r+1}^{nK} \lambda_s^2 (\langle \boldsymbol{u}_s, \boldsymbol{r}_0 \rangle)^2 \\
&\le \beta^4 \sum_{s=1}^{r} \frac{1}{\lambda_s^2} (\langle \boldsymbol{u}_s, \boldsymbol{r}_0 \rangle)^2 + \sum_{s=r+1}^{nK} \lambda_s^2 (\langle \boldsymbol{u}_s, \boldsymbol{r}_0 \rangle)^2 \\
&\le \beta^4 \left( \sum_{s=1}^{r} \frac{1}{\lambda_s^2} (\langle \boldsymbol{u}_s, \boldsymbol{r}_0 \rangle)^2 + \Gamma^2 \sum_{s=r+1}^{nK} \frac{\lambda_s^2}{\alpha^4} (\langle \boldsymbol{u}_s, \boldsymbol{r}_0 \rangle)^2 \right) \\
&= \beta^4 \left( \frac{\mathcal{B}_{\alpha,\Gamma}}{\alpha} \right)^2
\end{aligned}
$$

(g) from the fact that $\eta \le \frac{1}{\beta^2}$, and (h) from the fact that $\alpha \le \beta$ and $\Gamma \ge 1$.

**Step II: Original and linearized residuals are close (first part of (5.14)).**
In this step we wish to show that the first part of (5.14) holds for iteration $\tau$. Since we established in

the previous step that $\|\boldsymbol{\theta}_\tau - \boldsymbol{\theta}_0\|_{\ell_2} \le R$ the assumption of Lemma 6.7 holds for iterations $\tau - 1$ and $\tau$. Hence, using Lemma 6.7 equation (6.11) we conclude that

$$\|\boldsymbol{e}_\tau\|_{\ell_2} \le \eta(\varepsilon_0^2 + \varepsilon\beta)\|\widetilde{\boldsymbol{r}}_{\tau-1}\|_{\ell_2} + (1 + \eta\varepsilon^2)\|\boldsymbol{e}_{\tau-1}\|_{\ell_2}.$$

This combined with the induction assumption implies that

$$\|\boldsymbol{e}_t\|_{\ell_2} \le \eta(\varepsilon_0^2 + \varepsilon\beta)\|\widetilde{\boldsymbol{r}}_{t-1}\|_{\ell_2} + (1 + \eta\varepsilon^2)\|\boldsymbol{e}_{t-1}\|_{\ell_2}, \tag{6.18}$$

holds for all $t \le \tau \le T$. Furthermore, using Lemma 6.5 equation (6.9) for all $t \le \tau \le T$ we have

$$\|\widetilde{\boldsymbol{r}}_t\|_{\ell_2} \le \left(1 - \eta\alpha^2\right)^t \|\Pi_{\mathcal{I}}(\boldsymbol{r}_0)\|_{\ell_2} + \|\Pi_{\mathcal{N}}(\boldsymbol{r}_0)\|_{\ell_2}, \tag{6.19}$$

To proceed, we shall apply Lemma 6.8 with the following variable substitutions

$$\Theta := \varepsilon_0^2 + \varepsilon\beta, \quad \rho_+ = \|\Pi_{\mathcal{I}}(\boldsymbol{r}_0)\|_{\ell_2}, \quad \rho_- = \|\Pi_{\mathcal{N}}(\boldsymbol{r}_0)\|_{\ell_2}, \quad e_\tau := \|\boldsymbol{e}_\tau\|_{\ell_2}, \quad \widetilde{r}_\tau := \|\widetilde{\boldsymbol{r}}_\tau\|_{\ell_2}. \tag{6.20}$$

We note that Lemma 6.8 is applicable since (i) $\eta \le 1/\beta^2 \le 1/\alpha^2$, (ii) based on (5.13) we have $\frac{\alpha}{\varepsilon} \ge \frac{5\Gamma}{\delta}\frac{\beta^2}{\alpha^2} \ge \sqrt{2\Gamma}$, (iii) $\tau$ obeys $\tau \le T = \frac{\Gamma}{\eta\alpha^2}$, and (iv) (6.15) holds based on (6.18) and (6.19). Thus using Lemma 6.8 we can conclude that

$$\begin{aligned}
\|\boldsymbol{e}_\tau\|_{\ell_2} &\le 2(\varepsilon_0^2 + \varepsilon\beta)\frac{(\|\Pi_{\mathcal{I}}(\boldsymbol{r}_0)\|_{\ell_2} + \Gamma\|\Pi_{\mathcal{N}}(\boldsymbol{r}_0)\|_{\ell_2})}{\alpha^2} \\
&\le \frac{2\Gamma(\varepsilon_0^2 + \varepsilon\beta)\|\boldsymbol{r}_0\|_{\ell_2}}{\alpha^2} \tag{6.21} \\
&\le \left(\frac{2}{25} + \frac{2}{5}\right)\frac{\delta\alpha}{\beta}\|\boldsymbol{r}_0\|_{\ell_2} \le \frac{3}{5}\frac{\delta\alpha}{\beta}\|\boldsymbol{r}_0\|_{\ell_2}, \tag{6.22}
\end{aligned}$$

where in the last inequality we used (5.13). This completes the first part of (5.14) via induction.

**Step III: Original and linearized parameters are close (second part of (5.14)).**
In this step we wish to show that the second part of (5.14) holds for iteration $\tau$. To do this we begin by noting that by the fact that $\boldsymbol{J}$ is a reference Jacobian we have $\|\overline{\mathcal{J}}(\boldsymbol{\theta}_0) - \boldsymbol{J}\| \le \varepsilon_0$ where $\overline{\mathcal{J}}$ augments $\mathcal{J}(\boldsymbol{\theta}_0)$ by padding zero columns to match size of $\boldsymbol{J}$. Also by Assumption 2 we have $\|\mathcal{J}(\boldsymbol{\theta}) - \mathcal{J}(\boldsymbol{\theta}_0)\| \le \frac{\varepsilon}{2}$. Combining the latter two via the triangular inequality we conclude that

$$\|\overline{\mathcal{J}}(\boldsymbol{\theta}_\tau) - \boldsymbol{J}\| \le \varepsilon_0 + \varepsilon. \tag{6.23}$$

Let $\overline{\boldsymbol{\theta}}$ and $\nabla\bar{\mathcal{L}}(\boldsymbol{\theta})$ be vectors augmented by zero padding $\boldsymbol{\theta}, \nabla\mathcal{L}(\boldsymbol{\theta})$ so that they have dimension $\max(Kn, p)$. Now, we track the difference between $\overline{\boldsymbol{\theta}}$ and linearized $\tilde{\boldsymbol{\theta}}$ as follows

$$\begin{aligned}
\frac{\|\overline{\boldsymbol{\theta}}_\tau - \widetilde{\boldsymbol{\theta}}_\tau\|_{\ell_2}}{\eta} &= \left\|\sum_{t=0}^{\tau-1} \nabla\bar{\mathcal{L}}(\boldsymbol{\theta}_t) - \nabla\mathcal{L}_{lin}(\widetilde{\boldsymbol{\theta}}_t)\right\|_{\ell_2} \\
&= \left\|\sum_{t=0}^{\tau-1} \overline{\mathcal{J}}(\boldsymbol{\theta}_t)^T \boldsymbol{r}_t - \boldsymbol{J}^T\widetilde{\boldsymbol{r}}_t\right\|_{\ell_2} \\
&\le \sum_{t=0}^{\tau-1} \|\overline{\mathcal{J}}(\boldsymbol{\theta}_t)^T \boldsymbol{r}_t - \boldsymbol{J}^T\widetilde{\boldsymbol{r}}_t\|_{\ell_2} \\
&\le \sum_{t=0}^{\tau-1} \|(\overline{\mathcal{J}}(\boldsymbol{\theta}_t) - \boldsymbol{J})^T\widetilde{\boldsymbol{r}}_t\|_{\ell_2} + \|\overline{\mathcal{J}}(\boldsymbol{\theta}_t)^T(\boldsymbol{r}_t - \widetilde{\boldsymbol{r}}_t)\|_{\ell_2} \\
&= \sum_{t=0}^{\tau-1} \|(\overline{\mathcal{J}}(\boldsymbol{\theta}_t) - \boldsymbol{J})^T\widetilde{\boldsymbol{r}}_t\|_{\ell_2} + \|\overline{\mathcal{J}}(\boldsymbol{\theta}_t)^T\boldsymbol{e}_t\|_{\ell_2} \\
&\le \sum_{t=0}^{\tau-1} (\varepsilon + \varepsilon_0)\|\widetilde{\boldsymbol{r}}_t\|_{\ell_2} + \beta\|\boldsymbol{e}_t\|_{\ell_2}. \tag{6.24}
\end{aligned}$$

In the last inequality we used the fact that $\|\overline{\mathcal{J}}(\boldsymbol{\theta}_t) - \boldsymbol{J}\| \le \varepsilon + \varepsilon_0$ and $\|\boldsymbol{J}\| \le \beta$. We proceed by bounding each of the two terms in (6.24) above. For the first term we use the fact that $\|\widetilde{\boldsymbol{r}}_\tau\|_{\ell_2} \le \|\boldsymbol{r}_0\|_{\ell_2}$ to conclude

$$\sum_{t=0}^{\tau-1} \|\widetilde{\boldsymbol{r}}_t\|_{\ell_2} \le \tau\|\boldsymbol{r}_0\|_{\ell_2} \le T\|\boldsymbol{r}_0\|_{\ell_2} = \frac{\Gamma\|\boldsymbol{r}_0\|_{\ell_2}}{\eta\alpha^2}. \tag{6.25}$$

To bound the second term in (6.24) we use (6.21) together with $\tau \le T \le \frac{\Gamma}{\eta\alpha^2}$ to conclude that

$$\sum_{t=0}^{\tau-1} \|\boldsymbol{e}_t\|_{\ell_2} \le \tau \frac{2(\varepsilon\beta + \varepsilon_0^2)}{\alpha^2}\Gamma\|\boldsymbol{r}_0\|_{\ell_2} \le \frac{2\Gamma^2(\varepsilon\beta + \varepsilon_0^2)}{\eta\alpha^4}\|\boldsymbol{r}_0\|_{\ell_2}. \tag{6.26}$$

Combining (6.25) and (6.26) in (6.24), we conclude that

$$
\begin{aligned}
\|\overline{\boldsymbol{\theta}}_\tau - \widetilde{\boldsymbol{\theta}}_\tau\|_{\ell_2} &\le \left( \frac{2\Gamma(\varepsilon\beta^2 + \varepsilon_0^2\beta)}{\alpha^3} + \frac{\varepsilon + \varepsilon_0}{\alpha} \right)\frac{\Gamma}{\alpha}\|\boldsymbol{r}_0\|_{\ell_2} \\
&= \left( \varepsilon\frac{2\Gamma\beta^2}{\alpha^3} + \varepsilon_0^2\frac{2\Gamma\beta}{\alpha^3} + \frac{\varepsilon + \varepsilon_0}{\alpha} \right)\frac{\Gamma}{\alpha}\|\boldsymbol{r}_0\|_{\ell_2} \\
&\overset{(a)}{\le} \left( \frac{2}{5}\delta + \varepsilon_0^2\frac{2\Gamma\beta}{\alpha^3} + \frac{\varepsilon + \varepsilon_0}{\alpha} \right)\frac{\Gamma}{\alpha}\|\boldsymbol{r}_0\|_{\ell_2} \\
&\overset{(b)}{\le} \left( \frac{2}{5}\delta + \frac{2}{25}\delta + \frac{\varepsilon + \varepsilon_0}{\alpha} \right)\frac{\Gamma}{\alpha}\|\boldsymbol{r}_0\|_{\ell_2} \\
&\overset{(c)}{\le} \left( \frac{2}{5}\delta + \frac{2}{25}\delta + \frac{1}{5}\delta + \frac{\varepsilon_0}{\alpha} \right)\frac{\Gamma}{\alpha}\|\boldsymbol{r}_0\|_{\ell_2} \\
&\overset{(d)}{\le} \left( \frac{2}{5}\delta + \frac{2}{25}\delta + \frac{1}{5}\delta + \frac{1}{5}\delta \right)\frac{\Gamma}{\alpha}\|\boldsymbol{r}_0\|_{\ell_2} \\
&= \frac{22}{25}\frac{\delta}{\alpha}\Gamma\|\boldsymbol{r}_0\|_{\ell_2}.
\end{aligned}
$$

Here, (a) follows from $\varepsilon \le \frac{\delta\alpha^3}{5\Gamma\beta^2}$ per Assumption (5.13), (b) from $\varepsilon_0 \le \frac{1}{5}\sqrt{\frac{\delta\alpha^3}{\Gamma\beta}}$ per Assumption (5.13), (c) from $\varepsilon \le \frac{\delta\alpha^3}{5\Gamma\beta^2} \le \frac{\delta\alpha}{5\Gamma} \le \frac{\delta\alpha}{5}$ per Assumption (5.13), and (d) from $\varepsilon_0 \le \frac{\delta\alpha}{5}$ per Assumption (5.13). Thus,

$$\|\overline{\boldsymbol{\theta}}_\tau - \widetilde{\boldsymbol{\theta}}_\tau\|_{\ell_2} \le \frac{\delta}{\alpha}\Gamma\|\boldsymbol{r}_0\|_{\ell_2}.$$

Combining the latter with the fact that $\|\widetilde{\boldsymbol{\theta}}_\tau - \overline{\boldsymbol{\theta}}_0\|_{\ell_2} \le \frac{\mathcal{B}_{\alpha,\Gamma}}{\alpha}$ (which follows from Lemma 6.5 equation (6.6)) we conclude that

$$\|\boldsymbol{\theta}_\tau - \boldsymbol{\theta}_0\|_{\ell_2} = \|\overline{\boldsymbol{\theta}}_\tau - \overline{\boldsymbol{\theta}}_0\|_{\ell_2} \le \|\widetilde{\boldsymbol{\theta}}_\tau - \overline{\boldsymbol{\theta}}_0\|_{\ell_2} + \|\overline{\boldsymbol{\theta}}_\tau - \widetilde{\boldsymbol{\theta}}_\tau\|_{\ell_2} \le \frac{\mathcal{B}_{\alpha,\Gamma}}{\alpha} + \frac{\delta}{\alpha}\Gamma\|\boldsymbol{r}_0\|_{\ell_2} \le \|\boldsymbol{J}_\mathcal{I}^\dagger \boldsymbol{r}_0\|_{\ell_2} + \frac{\Gamma}{\alpha}\|\Pi_\mathcal{N}(\boldsymbol{r}_0)\|_{\ell_2} + \frac{\delta}{\alpha}\Gamma\|\boldsymbol{r}_0\|_{\ell_2}$$

The completes the proof of the bound (5.15).

**Step V: Bound on residual with early stopping.**
In this step we wish to prove (5.16). To this aim note that

$$
\begin{aligned}
\|\boldsymbol{r}_T\|_{\ell_2} &\overset{(a)}{\le} \|\widetilde{\boldsymbol{r}}_T\|_{\ell_2} + \|\widetilde{\boldsymbol{r}}_T - \boldsymbol{r}_T\|_{\ell_2} \\
&\overset{(b)}{\le} \|\widetilde{\boldsymbol{r}}_T\|_{\ell_2} + \frac{\delta\alpha}{\beta}\|\boldsymbol{r}_0\|_{\ell_2} \\
&\overset{(c)}{\le} e^{-\Gamma}\|\Pi_\mathcal{I}(\boldsymbol{r}_0)\|_{\ell_2} + \|\Pi_\mathcal{N}(\boldsymbol{r}_0)\|_{\ell_2} + \frac{\delta\alpha}{\beta}\|\boldsymbol{r}_0\|_{\ell_2}
\end{aligned}
$$

where (a) follows from the triangular inequality, (b) from the conclusion of Step II (first part of (5.14)), and (c) from Corollary 6.6 equation (6.10). This completes the proof of (5.16).

## 6.2 Key lemmas and identities for neural networks

In this section we prove some key lemmas and identities regarding the Jacobian of one-hidden layer networks as well as the size of the initial residual that when combined with Theorem 5.3 allows us to prove theorems involving neural networks. We begin with some preliminary identities and calculations in Section 6.2.1. Next, in Section 6.2.2 we prove a few key properties of the Jacobian mapping of a one-hidden layer neural network. Section 6.2.3 focuses on a few further properties of the Jacobian at a random initialization. Finally, in Section 6.2.4 we provide bounds on the initial misfit.

For two matrices

$$\boldsymbol{A} = \begin{bmatrix} \boldsymbol{A}_1 \\ \boldsymbol{A}_2 \\ \vdots \\ \boldsymbol{A}_p \end{bmatrix} \in \mathbb{R}^{p \times m} \quad \text{and} \quad \boldsymbol{B} = \begin{bmatrix} \boldsymbol{B}_1 \\ \boldsymbol{B}_2 \\ \vdots \\ \boldsymbol{B}_p \end{bmatrix} \in \mathbb{R}^{p \times n},$$

we define their Khatri-Rao product as $\boldsymbol{A} * \boldsymbol{B} = [\boldsymbol{A}_1 \otimes \boldsymbol{B}_1, \ldots, \boldsymbol{A}_p \otimes \boldsymbol{B}_p] \in \mathbb{R}^{p \times mn}$, where $\otimes$ denotes the Kronecker product.

### 6.2.1 PRELIMINARY IDENTITIES AND CALCULATIONS

We begin by discussing a few notations. Throughout we use $\boldsymbol{w}_\ell$ and $\boldsymbol{v}_\ell$ to denote the $\ell$th row of input and output weight matrices $\boldsymbol{W}$ and $\boldsymbol{V}$. Given a matrix $\boldsymbol{M}$ we use $\|\boldsymbol{M}\|_{2,\infty}$ to denote the largest Euclidean norm of the rows of $\boldsymbol{M}$. We begin by noting that for a one-hidden layer neural network of the form $\boldsymbol{x} \mapsto \boldsymbol{V}\phi(\boldsymbol{W}\boldsymbol{x})$, the Jacobian matrix with respect to $\text{vect}(\boldsymbol{W}) \in \mathbb{R}^{kd}$ takes the form

$$\mathcal{J}(\boldsymbol{W}) = \begin{bmatrix} \mathcal{J}_1(\boldsymbol{W}) \\ \vdots \\ \mathcal{J}_K(\boldsymbol{W}) \end{bmatrix} \in \mathbb{R}^{Kn \times kd} \tag{6.27}$$

where $\mathcal{J}_\ell(\boldsymbol{W})$ is the Jacobian matrix associated with the $\ell$th class. In particular, $\mathcal{J}_\ell(\boldsymbol{W})$ is given by

$$\mathcal{J}_\ell(\boldsymbol{W}) = [\mathcal{J}_\ell(\boldsymbol{w}_1) \quad \ldots \quad \mathcal{J}_\ell(\boldsymbol{w}_k)] \in \mathbb{R}^{n \times kd} \quad \text{with} \quad \mathcal{J}_\ell(\boldsymbol{w}_s) := \boldsymbol{V}_{\ell,s} \text{diag}(\phi'(\boldsymbol{X}\boldsymbol{w}_s))\boldsymbol{X}.$$

Alternatively using Khatri-Rao products this can be rewritten in the more compact form

$$\mathcal{J}_\ell(\boldsymbol{W}) = (\phi'(\boldsymbol{X}\boldsymbol{W}^T)\text{diag}(\boldsymbol{v}_\ell)) * \boldsymbol{X}. \tag{6.28}$$

An alternative characterization of the Jacobian is via its matrix representation. Given a vector $\boldsymbol{u} \in \mathbb{R}^{Kn}$ let us partition it into $K$ size $n$ subvectors so that $\boldsymbol{u} = [\boldsymbol{u}_1^T \ \ldots \ \boldsymbol{u}_K^T]^T$. We have

$$\text{mat}(\mathcal{J}^T(\boldsymbol{W})\boldsymbol{u}) = \sum_{\ell=1}^{K} \text{diag}(\boldsymbol{v}_\ell)\phi'(\boldsymbol{W}\boldsymbol{X}^T)\text{diag}(\boldsymbol{u}_\ell)\boldsymbol{X}. \tag{6.29}$$

### 6.2.2 FUNDAMENTAL PROPERTIES OF THE JACOBIAN OF THE NEURAL NETWORK

In this section we prove a few key properties of the Jacobian mapping of a one-hidden layer neural network.

**Lemma 6.9 (Properties of Single Output Neural Net Jacobian)** *Let $K = 1$ so that $\boldsymbol{V}^T = \boldsymbol{v} \in \mathbb{R}^n$. Suppose $\phi$ is an activation obeying $|\phi'(z)| \le B$ for all $z$. Then, for any $\boldsymbol{W} \in \mathbb{R}^{k \times d}$ and any unit length vector $\boldsymbol{u}$, we have*

$$\|\mathcal{J}(\boldsymbol{W})\| \le B\sqrt{k}\|\boldsymbol{v}\|_{\ell_\infty}\|\boldsymbol{X}\|$$

*and*

$$\|\text{mat}(\mathcal{J}^T(\boldsymbol{W})\boldsymbol{u})\|_{2,\infty} \le B\|\boldsymbol{v}\|_{\ell_\infty}\|\boldsymbol{X}\| \tag{6.30}$$

*Furthermore, suppose $\phi$ is twice differentiable and $|\phi''(z)| \le B$ for all $z$. Also assume all data points have unit Euclidean norm ($\|\boldsymbol{x}_i\|_{\ell_2} = 1$). Then the Jacobian mapping is Lipschitz with respect to spectral norm i.e. for all $\widetilde{\boldsymbol{W}}, \boldsymbol{W} \in \mathbb{R}^{k \times d}$ we have*

$$\|\mathcal{J}(\widetilde{\boldsymbol{W}}) - \mathcal{J}(\boldsymbol{W})\| \le B\|\boldsymbol{v}\|_{\ell_\infty}\|\boldsymbol{X}\|\|\widetilde{\boldsymbol{W}} - \boldsymbol{W}\|_F.$$

**Proof** The result on spectral norm and Lipschitzness of $\mathcal{J}(\boldsymbol{W})$ have been proven in Oymak & Soltanolkotabi (2019). To show the row-wise bound (6.30), we use (6.29) to conclude that

$$\begin{aligned}
\|\text{mat}(\mathcal{J}^T(\boldsymbol{W})\boldsymbol{u})\|_{2,\infty} &= \|\text{diag}(\boldsymbol{v})\phi'(\boldsymbol{W}\boldsymbol{X}^T)\text{diag}(\boldsymbol{u})\boldsymbol{X}\|_{2,\infty} \\
&\le \|\boldsymbol{v}\|_{\ell_\infty} \max_{1 \le \ell \le k} \|\phi'(\boldsymbol{w}_\ell^T\boldsymbol{X}^T)\text{diag}(\boldsymbol{u})\boldsymbol{X}\|_{\ell_2} \\
&\le \|\boldsymbol{v}\|_{\ell_\infty}\|\boldsymbol{X}\| \max_{1 \le \ell \le k} \|\phi'(\boldsymbol{w}_\ell^T\boldsymbol{X}^T)\text{diag}(\boldsymbol{u})\|_{\ell_2} \\
&\le B\|\boldsymbol{v}\|_{\ell_\infty}\|\boldsymbol{X}\|\|\boldsymbol{u}\|_{\ell_2} \\
&= B\|\boldsymbol{v}\|_{\ell_\infty}\|\boldsymbol{X}\|.
\end{aligned}$$

∎

Next we extend the lemma above to the multi-class setting.

**Lemma 6.10 (Properties of Multiclass Neural Net Jacobian)** *Suppose $\phi$ is an activation obeying $|\phi'(z)| \leq B$ for all $z$. Then, for any $\boldsymbol{W} \in \mathbb{R}^{k \times d}$ and any unit length vector $\boldsymbol{u}$, we have*

$$\|\mathcal{J}(\boldsymbol{W})\| \leq B\sqrt{Kk} \|\boldsymbol{V}\|_{\ell_\infty} \|\boldsymbol{X}\| \tag{6.31}$$

*and*

$$\|mat\left(\mathcal{J}^T(\boldsymbol{W})\boldsymbol{u}\right)\|_{2,\infty} \leq B\sqrt{K}\|\boldsymbol{V}\|_{\ell_\infty}\|\boldsymbol{X}\|. \tag{6.32}$$

*Furthermore, suppose $\phi$ is twice differentiable and $|\phi''(z)| \leq B$ for all $z$. Also assume all data points have unit Euclidean norm ($\|\boldsymbol{x}_i\|_{\ell_2} = 1$). Then the Jacobian mapping is Lipschitz with respect to spectral norm i.e. for all $\widetilde{\boldsymbol{W}}, \boldsymbol{W} \in \mathbb{R}^{k \times d}$ we have*

$$\left\|\mathcal{J}(\widetilde{\boldsymbol{W}}) - \mathcal{J}(\boldsymbol{W})\right\| \leq B\sqrt{K}\|\boldsymbol{V}\|_{\ell_\infty}\|\boldsymbol{X}\|\left\|\widetilde{\boldsymbol{W}} - \boldsymbol{W}\right\|_F.$$

**Proof** The proof will follow from Lemma 6.9. First, given $\boldsymbol{A} = \begin{bmatrix} \boldsymbol{A}_1^T & \dots & \boldsymbol{A}_K^T \end{bmatrix}^T$ and $\boldsymbol{B} = \begin{bmatrix} \boldsymbol{B}_1^T & \dots & \boldsymbol{B}_K^T \end{bmatrix}^T$, observe that

$$\|\boldsymbol{A}\| \leq \sqrt{K} \sup_{1 \leq \ell \leq K} \|\boldsymbol{A}_\ell\| \quad \text{and} \quad \|\boldsymbol{A} - \boldsymbol{B}\| \leq \sqrt{K} \sup_{1 \leq \ell \leq K} \|\boldsymbol{A}_\ell - \boldsymbol{B}_\ell\|.$$

These two identities applied to the components $\mathcal{J}_\ell(\boldsymbol{W})$ and $\mathcal{J}_\ell(\widetilde{\boldsymbol{W}}) - \mathcal{J}_\ell(\boldsymbol{W})$ completes the proof of the bound on the spectral norm and the perturbation. To prove the bound in (6.32) we use the identity (6.29) to conclude that

$$
\begin{aligned}
\|\text{mat}\left(\mathcal{J}^T(\boldsymbol{W})\boldsymbol{u}\right)\|_{2,\infty} &= \|\sum_{\ell=1}^{K} \text{diag}(\boldsymbol{v}_\ell)\phi'\left(\boldsymbol{W}\boldsymbol{X}^T\right)\text{diag}(\boldsymbol{u}_\ell)\boldsymbol{X}\|_{2,\infty} \\
&\leq \sum_{\ell=1}^{K} \|\text{diag}(\boldsymbol{v}_\ell)\phi'\left(\boldsymbol{W}\boldsymbol{X}^T\right)\text{diag}(\boldsymbol{u}_\ell)\boldsymbol{X}\|_{2,\infty} \\
&\leq \sum_{\ell=1}^{K} B\|\boldsymbol{V}\|_{\ell_\infty}\|\boldsymbol{X}\|\|\boldsymbol{u}_\ell\|_{\ell_2} \\
&= B\|\boldsymbol{V}\|_{\ell_\infty}\|\boldsymbol{X}\|\left(\sum_{\ell=1}^{K}\|\boldsymbol{u}_\ell\|_{\ell_2}\right) \\
&\leq B\|\boldsymbol{V}\|_{\ell_\infty}\|\boldsymbol{X}\|\sqrt{K}\left(\sum_{\ell=1}^{K}\|\boldsymbol{u}_\ell\|_{\ell_2}^2\right)^{1/2} \\
&= B\|\boldsymbol{V}\|_{\ell_\infty}\|\boldsymbol{X}\|\sqrt{K},
\end{aligned}
$$

where the penultimate inequality follows from Cauchy Schwarz, completing the proof. ∎

### 6.2.3 PROPERTIES OF THE JACOBIAN AT RANDOM INITIALIZATION

In this section we prove a few lemmas characterizing the properties of the Jacobian at the random initialization.

**Lemma 6.11 (Multiclass covariance)** *Given input and output layer weights $\boldsymbol{V}$ and $\boldsymbol{W}$, consider the Jacobian described in (6.27). Given an $Kn \times Kn$ matrix $\boldsymbol{M}$, for $1 \leq \ell, \widetilde{\ell} \leq K$, let $\boldsymbol{M}[\ell, \widetilde{\ell}]$ denote the $(\ell, \widetilde{\ell})$th submatrix. For $\boldsymbol{C}(\boldsymbol{W}) = \mathcal{J}(\boldsymbol{W})\mathcal{J}(\boldsymbol{W})^T$ we have*

$$\boldsymbol{C}(\boldsymbol{W})[\ell, \widetilde{\ell}] = \sum_{s=1}^{k}(\boldsymbol{X}\boldsymbol{X}^T) \odot (\boldsymbol{V}_{\ell,s}\boldsymbol{V}_{\widetilde{\ell},s}\phi'(\boldsymbol{X}\boldsymbol{w}_s)\phi'(\boldsymbol{X}\boldsymbol{w}_s)^T).$$

*Suppose $\boldsymbol{W} \overset{i.i.d.}{\sim} \mathcal{N}(0,1)$ and $\boldsymbol{V}$ has i.i.d. zero-mean entries with $\nu^2$ variance. Then $\mathbb{E}[\boldsymbol{C}(\boldsymbol{W})]$ is a block diagonal matrix given by the Kronecker product*

$$\mathbb{E}[\boldsymbol{C}(\boldsymbol{W})] = k\nu^2 \boldsymbol{\Sigma}(\boldsymbol{X}).$$

*where $\boldsymbol{\Sigma}(\boldsymbol{X})$ is equal to $\boldsymbol{I}_K \otimes [(\boldsymbol{X}\boldsymbol{X}^T) \odot \mathbb{E}[\phi'(\boldsymbol{X}\boldsymbol{w}_s)\phi'(\boldsymbol{X}\boldsymbol{w}_s)^T]]$.*

**Proof** The $(\ell, \widetilde{\ell})$th submatrix of $\boldsymbol{C}(\boldsymbol{W})$ is given by

$$
\begin{aligned}
\boldsymbol{C}(\boldsymbol{W})[\ell, \widetilde{\ell}] &= ((\operatorname{diag}(\boldsymbol{v}_\ell)\phi'(\boldsymbol{W}\boldsymbol{X}^T)) * \boldsymbol{X}^T)((\operatorname{diag}(\boldsymbol{v}_{\widetilde{\ell}})\phi'(\boldsymbol{W}\boldsymbol{X}^T)) * \boldsymbol{X}^T)^T \\
&= \sum_{s=1}^{k} \mathcal{J}_\ell(\boldsymbol{w}_s)\mathcal{J}_{\widetilde{\ell}}(\boldsymbol{w}_s)^T \\
&= \sum_{s=1}^{k} \boldsymbol{V}_{\ell,s}\boldsymbol{V}_{\widetilde{\ell},s}(\operatorname{diag}(\phi'(\boldsymbol{X}\boldsymbol{w}_s))\boldsymbol{X})(\operatorname{diag}(\phi'(\boldsymbol{X}\boldsymbol{w}_s))\boldsymbol{X})^T \\
&= \sum_{s=1}^{k} \boldsymbol{V}_{\ell,s}\boldsymbol{V}_{\widetilde{\ell},s}(\boldsymbol{X}\boldsymbol{X}^T) \odot (\phi'(\boldsymbol{X}\boldsymbol{w}_s)\phi'(\boldsymbol{X}\boldsymbol{w}_s)^T) \\
&= \sum_{s=1}^{k} (\boldsymbol{X}\boldsymbol{X}^T) \odot (\boldsymbol{V}_{\ell,s}\boldsymbol{V}_{\widetilde{\ell},s}\phi'(\boldsymbol{X}\boldsymbol{w}_s)\phi'(\boldsymbol{X}\boldsymbol{w}_s)^T). \quad (6.33)
\end{aligned}
$$

Setting $\boldsymbol{W} \overset{\text{i.i.d.}}{\sim} \mathcal{N}(0,1)$ and $\boldsymbol{V}$ with i.i.d. zero-mean and $\nu^2$-variance entries, we conclude that

$$
\begin{aligned}
\mathbb{E}[\boldsymbol{C}(\boldsymbol{W})[\ell, \widetilde{\ell}]] &= \sum_{s=1}^{k} (\boldsymbol{X}\boldsymbol{X}^T) \odot (\mathbb{E}[\boldsymbol{V}_{\ell,s}\boldsymbol{V}_{\widetilde{\ell},s}] \, \mathbb{E}[\phi'(\boldsymbol{X}\boldsymbol{w}_s)\phi'(\boldsymbol{X}\boldsymbol{w}_s)^T]) \\
&= \sum_{s=1}^{k} \nu^2\delta(\ell - \widetilde{\ell})[(\boldsymbol{X}\boldsymbol{X}^T) \odot \mathbb{E}[\phi'(\boldsymbol{X}\boldsymbol{w}_s)\phi'(\boldsymbol{X}\boldsymbol{w}_s)^T]] \\
&= k\delta(\ell - \widetilde{\ell})\nu^2\tilde{\boldsymbol{\Sigma}}(\boldsymbol{X}),
\end{aligned}
$$

where $\delta(x)$ is the discrete $\delta$ function which is 0 for $x \neq 0$ and 1 for $x = 0$ and $\tilde{\boldsymbol{\Sigma}}(\boldsymbol{X})$ is single output kernel matrix which concludes the proof. ∎

Next we state a useful lemma from Schur (1911) which allows us to bound the eigenvalues of the Hadamard product of the two PSD matrices.

**Lemma 6.12 (Schur (1911))** *Let $\boldsymbol{A}, \boldsymbol{B} \in \mathbb{R}^{n \times n}$ be two Positive Semi-Definite (PSD) matrices. Then,*

$$
\lambda_{\min}(\boldsymbol{A} \odot \boldsymbol{B}) \geq \left(\min_i \boldsymbol{B}_{ii}\right)\lambda_{\min}(\boldsymbol{A}),
$$

$$
\lambda_{\max}(\boldsymbol{A} \odot \boldsymbol{B}) \leq \left(\max_i \boldsymbol{B}_{ii}\right)\lambda_{\max}(\boldsymbol{A}).
$$

Next we state a lemma regarding concentration of the Jacobian matrix at initialization.

**Lemma 6.13 (Concentration of the Jacobian at initialization)** *Consider a one-hidden layer neural network model of the form $\boldsymbol{x} \mapsto \boldsymbol{V}\phi(\boldsymbol{W}\boldsymbol{x})$ where the activation $\phi$ obeys $|\phi(0)| \leq B$ and $|\phi'(z)| \leq B$ for all $z$. Also assume we have $n \geq K$ data points $\boldsymbol{x}_1, \boldsymbol{x}_2, \ldots, \boldsymbol{x}_n \in \mathbb{R}^d$ with unit euclidean norm ($\|\boldsymbol{x}_i\|_{\ell_2} = 1$) aggregated as the rows of a matrix $\boldsymbol{X} \in \mathbb{R}^{n \times d}$. Furthermore, suppose $\boldsymbol{V}$ has i.i.d. $\nu$-scaled Rademacher entries (i.e. $\pm\nu$ equally-likely). Then, the Jacobian matrix at a random point $\boldsymbol{W}_0 \in \mathbb{R}^{k \times d}$ with i.i.d. $\mathcal{N}(0,1)$ entries obeys*

$$
\|\mathcal{J}(\boldsymbol{W}_0)\mathcal{J}(\boldsymbol{W}_0)^T - \mathbb{E}[\mathcal{J}(\boldsymbol{W}_0)\mathcal{J}(\boldsymbol{W}_0)^T]\| \leq 30K\sqrt{k}\nu^2 B^2\|\boldsymbol{X}\|^2\log(n).
$$

*with probability at least $1 - 1/n^{100}$. In particular, as long as*

$$
k \geq \frac{1000K^2 B^4\|\boldsymbol{X}\|^4\log(n)}{\delta^2},
$$

*with the same probability, we have that*

$$
\left\|\frac{1}{k\nu^2}\mathcal{J}(\boldsymbol{W}_0)\mathcal{J}(\boldsymbol{W}_0)^T - \boldsymbol{\Sigma}(\boldsymbol{X})\right\| \leq \delta.
$$

**Proof** Define $\boldsymbol{C} = \mathcal{J}(\boldsymbol{W}_0)\mathcal{J}(\boldsymbol{W}_0)^T$. We begin by showing that the diagonal blocks of $\boldsymbol{C}$ are concentrated. To do this first for $1 \leq s \leq k$ define the random matrices

$$
\boldsymbol{A}_s = \left(\phi'(\boldsymbol{X}\boldsymbol{w}_s)\phi'(\boldsymbol{X}\boldsymbol{w}_s)^T\right) \odot \left(\boldsymbol{X}\boldsymbol{X}^T\right).
$$

Now consider $n \times n$ diagonal blocks of $\boldsymbol{C}$ (denoted by $\boldsymbol{C}[\ell, \ell]$) and note that we have

$$\boldsymbol{C}[\ell, \ell] = \left(\phi'\left(\boldsymbol{X}\boldsymbol{W}^T\right) \operatorname{diag}(\boldsymbol{v}_\ell)\operatorname{diag}(\boldsymbol{v}_\ell)\phi'\left(\boldsymbol{W}\boldsymbol{X}^T\right)\right) \odot \left(\boldsymbol{X}\boldsymbol{X}^T\right)$$

$$= \sum_{s=1}^{k} \boldsymbol{V}_{\ell,s}^2 \boldsymbol{A}_s$$

$$= \nu^2 \sum_{s=1}^{k} \boldsymbol{A}_s.$$

Furthermore, using Lemma 6.12

$$\|\boldsymbol{A}_s\| \le \left(\max_i \left(\phi'(\boldsymbol{x}_i^T \boldsymbol{w}_s)\right)^2\right)\|\boldsymbol{X}\|^2 \le B^2 \|\boldsymbol{X}\|^2.$$

Also, using Jensen's inequality

$$\|\mathbb{E}[\boldsymbol{A}_s]\| \le \mathbb{E}\|\boldsymbol{A}_s\| \le B^2 \|\boldsymbol{X}\|^2.$$

Combining the latter two identities via the triangular inequality we conclude that

$$\left\|(\boldsymbol{A}_s - \mathbb{E}[\boldsymbol{A}_s])^2\right\| = \|\boldsymbol{A}_s - \mathbb{E}[\boldsymbol{A}_s]\|^2 \le (\|\boldsymbol{A}_s\| + \|\mathbb{E}[\boldsymbol{A}_s]\|)^2 \le \left(2B^2\|\boldsymbol{X}\|^2\right)^2. \qquad (6.34)$$

To proceed, we will bound the weighted sum

$$\boldsymbol{S} = \sum_{s=1}^{k} \nu^2 (\boldsymbol{A}_s - \mathbb{E}[\boldsymbol{A}_s])$$

in spectral norm. To this aim we utilize the Matrix Hoeffding inequality which states that

$$\mathbb{P}(\|\boldsymbol{S}\| \ge t) \le 2n e^{-\frac{t^2}{2\Delta^2}},$$

where $\Delta^2$ is an upper bound on $\left\|\sum_{s=1}^{k} \nu^4 (\boldsymbol{A}_s - \mathbb{E}[\boldsymbol{A}_s])^2\right\|$. Using (6.34) we can pick $\Delta^2 = \sum_{s=1}^{k}(2\nu^2 B^2\|\boldsymbol{X}\|^2)^2 = 4k\nu^4 B^4\|\boldsymbol{X}\|^4$. Setting $t = 30\sqrt{k}\nu^2 B^2\|\boldsymbol{X}\|^2\sqrt{\log(n)}$, we conclude that

$$\mathbb{P}\left\{\|\boldsymbol{C}[\ell, \ell] - \mathbb{E}[\boldsymbol{C}[\ell, \ell]]\| \ge t\right\} = \mathbb{P}(\|\boldsymbol{S}\| \ge t) \le n^{-102}$$

concluding the proof of concentration of the diagonal blocks of $\boldsymbol{C}$.

For the off-diagonal blocks $\boldsymbol{C}[\ell, \widetilde{\ell}]$ using (6.33) from the proof of Lemma 6.11 we have that

$$\boldsymbol{C}[\ell, \widetilde{\ell}] = \sum_{s=1}^{k} \boldsymbol{V}_{\ell,s} \boldsymbol{V}_{\widetilde{\ell},s} \boldsymbol{A}_s.$$

Note that by construction $\{\boldsymbol{V}_{\ell,s}\boldsymbol{V}_{\widetilde{\ell},s}\}_{s=1}^{k}$ are i.i.d. $\pm\nu^2$ Rademacher variables and thus $\boldsymbol{C}[\ell, \widetilde{\ell}]$ is sum of zero-mean i.i.d. matrices and we are again in the position to apply Hoeffding's inequality. To this aim note that

$$\left\|\sum_{s=1}^{k} \boldsymbol{V}_{\ell,s}^2 \boldsymbol{V}_{\widetilde{\ell},s}^2 \boldsymbol{A}_s^2\right\| = \nu^4 \left\|\sum_{s=1}^{k} \boldsymbol{A}_s^2\right\| \le \nu^4 \sum_{s=1}^{k} \|\boldsymbol{A}_s\|^2 \le \nu^4 k B^4\|\boldsymbol{X}\|^4,$$

so that we can take $\Delta^2 = \nu^4 k B^4\|\boldsymbol{X}\|^4$ and again conclude that for $t = 30\sqrt{k}\nu^2 B^2\|\boldsymbol{X}\|^2\log(n)$ we have

$$\mathbb{P}\left\{\|\boldsymbol{C}[\ell, \widetilde{\ell}]\| \ge t\right\} \le n^{-102}$$

Using the fact that $\mathbb{E}[\boldsymbol{C}[\ell, \widetilde{\ell}]] = 0$ and $K \le n$, combined with a union bound over all sub-matrices $1 \le \ell, \widetilde{\ell} \le K$ we conclude that

$$\mathbb{P}\left\{\|\boldsymbol{C}[\ell, \widetilde{\ell}] - \mathbb{E}\left[\boldsymbol{C}[\ell, \widetilde{\ell}]\right]\| \ge t\right\} \le K^2 n^{-102} \le n^{-100}.$$

All that remains is to combine the concentration results for the sub-matrices to arrive at the complete bound. In mathematical terms we need to bound $\mathbf{D} := \|\boldsymbol{C} - \mathbb{E}[\boldsymbol{C}]\|$. To this aim define $\boldsymbol{D}[\ell, :]$ to denote the $\ell$th block row of $\boldsymbol{D}$. Standard bounds on spectral norm in terms of sub-matrices allow us to conclude that

$$\|\boldsymbol{D}[\ell, :]\| \le \sqrt{K} \sup_{1 \le \widetilde{\ell} \le K} \|\boldsymbol{D}[\ell, \widetilde{\ell}]\| \le \sqrt{K}t \quad \Rightarrow$$

$$\|\boldsymbol{D}\| \le \sqrt{K} \sup_{1 \le \ell \le K} \|\boldsymbol{D}[\ell, :]\| \le \sqrt{K}\sqrt{K}t = Kt = 30K\sqrt{k}\nu^2 B^2\|\boldsymbol{X}\|^2\log(n),$$

concluding the proof. The result in terms of $\delta$ is obtained by using the population covariance Lemma 6.11. ∎

### 6.2.4 UPPER BOUND ON INITIAL RESIDUAL

In this section we prove a lemma concerning the size of the initial misfit. The proof of this lemma (stated below) follows from a similar argument in the proof of (Oymak & Soltanolkotabi, 2019, Lemma 6.12).

**Lemma 6.14 (Upper bound on initial residual)** *Consider a one-hidden layer neural network model of the form* $x \mapsto V\phi(Wx)$ *where the activation* $\phi$ *has bounded derivatives obeying* $|\phi(0)|, |\phi'(z)| \le B$. *Also assume we have* $n$ *data points* $x_1, x_2, \ldots, x_n \in \mathbb{R}^d$ *with unit euclidean norm* $(\|x_i\|_{\ell_2} = 1)$ *aggregated as rows of a matrix* $X \in \mathbb{R}^{n \times d}$ *and the corresponding labels given by* $y \in \mathbb{R}^{Kn}$. *Furthermore, assume the entries of* $V$ *are i.i.d. Rademacher variables scaled by* $\frac{\nu \|y\|_{\ell_2}}{50B\sqrt{K \log(2K)kn}}$ *and the entries of* $W \in \mathbb{R}^{k \times d}$ *are i.i.d.* $\mathcal{N}(0,1)$. *Then,*

$$\left\| V\phi(WX^T) \right\|_F \le \nu \|y\|_{\ell_2},$$

*holds with probability at least* $1 - (2K)^{-100}$.

**Proof** We begin the proof by noting that

$$\left\| V\phi(WX^T) \right\|_F^2 = \sum_{\ell=1}^K \left\| v_\ell^T \phi(WX^T) \right\|_{\ell_2}^2.$$

We will show that for any row $v$ of $V$, with probability at least $1 - (2K)^{-101}$,

$$\left\| v_\ell^T \phi(WX^T) \right\|_{\ell_2} \le \frac{\nu}{\sqrt{K}} \|y\|_{\ell_2}. \tag{6.35}$$

so that a simple union bound can conclude the proof. Therefore, all that remains is to show (6.35) holds. To prove the latter, note that for any two matrices $\widetilde{W}, W \in \mathbb{R}^{k \times d}$ we have

$$
\begin{aligned}
\left| \left\| \phi(X\widetilde{W}^T)v \right\|_{\ell_2} - \left\| \phi(XW^T)v \right\|_{\ell_2} \right| &\le \left\| \phi(X\widetilde{W}^T)v - \phi(XW^T)v \right\|_{\ell_2} \\
&\le \left\| \phi(X\widetilde{W}^T) - \phi(XW^T) \right\| \|v\|_{\ell_2} \\
&\le \left\| \phi(X\widetilde{W}^T) - \phi(XW^T) \right\|_F \|v\|_{\ell_2} \\
&\overset{(a)}{=} \left\| \left( \phi'\left( S \odot X\widetilde{W}^T + (1_{k \times n} - S) \odot XW^T \right) \right) \odot \left( X(\widetilde{W} - W)^T \right) \right\|_F \|v\|_{\ell_2} \\
&\le B \left\| X(\widetilde{W} - W)^T \right\|_F \|v\|_{\ell_2} \\
&\le B \|X\| \|v\|_{\ell_2} \left\| \widetilde{W} - W \right\|_F,
\end{aligned}
$$

where in (a) we used the mean value theorem with $S$ a matrix with entries obeying $0 \le S_{i,j} \le 1$ and $1_{k \times n}$ the matrix of all ones. Thus, $\left\| \phi(XW^T)v \right\|_{\ell_2}$ is a $B \|X\| \|v\|_{\ell_2}$-Lipschitz function of $W$. Thus, fixing $v$, for a matrix $W$ with i.i.d. Gaussian entries

$$\left\| \phi(XW^T)v \right\|_{\ell_2} \le \mathbb{E}\left[ \left\| \phi(XW^T)v \right\|_{\ell_2} \right] + t, \tag{6.36}$$

holds with probability at least $1 - e^{-\frac{t^2}{2B^2 \|v\|_{\ell_2}^2 \|X\|^2}}$. Next given $g \sim \mathcal{N}(0,1)$, we have

$$|\mathbb{E}[\phi(g)]| \le |\mathbb{E}[\phi(0)]| + |\mathbb{E}[\phi(g) - \phi(0)]| \le B + B\,\mathbb{E}[|g|] \le 2B \quad \text{and} \quad \text{Var}(\phi(g)) \le B^2. \tag{6.37}$$

where the latter follows from Poincare inequality (e.g. see (Ledoux, 2001, p. 49)). Furthermore, since $v$ has i.i.d. Rademacher entries, applying Bernstein bound, event

$$E_v := \{ |1^T v|^2 \le 250 \log K \|v\|_{\ell_2}^2 \} \tag{6.38}$$

holds with probability $1 - (2K)^{-102}$. Conditioned on $E_{\boldsymbol{v}}$, we now upper bound the expectation via

$$\mathbb{E}\big[\left\|\phi\big(\boldsymbol{X}\boldsymbol{W}^T\big)\boldsymbol{v}\right\|_{\ell_2}\big] \overset{(a)}{\leq} \sqrt{\mathbb{E}\big[\left\|\phi\big(\boldsymbol{X}\boldsymbol{W}^T\big)\boldsymbol{v}\right\|_{\ell_2}^2\big]}$$

$$= \sqrt{\sum_{i=1}^n \mathbb{E}\big[\big(\boldsymbol{v}^T\phi(\boldsymbol{W}\boldsymbol{x}_i)\big)^2\big]}$$

$$\overset{(b)}{=} \sqrt{n}\sqrt{\mathbb{E}_{\boldsymbol{g}\sim\mathcal{N}(\boldsymbol{0},\boldsymbol{I}_k)}\big[\big(\boldsymbol{v}^T\phi(\boldsymbol{g})\big)^2\big]}$$

$$\overset{(c)}{=} \sqrt{n}\sqrt{\left\|\boldsymbol{v}\right\|_{\ell_2}^2 \mathbb{E}_{g\sim\mathcal{N}(0,1)}\big[\big(\phi(g) - \mathbb{E}[\phi(g)]\big)^2\big] + (\boldsymbol{1}^T\boldsymbol{v})^2(\mathbb{E}_{g\sim\mathcal{N}(0,1)}[\phi(g)])^2}$$

$$\overset{(d)}{\leq} \sqrt{n}\left\|\boldsymbol{v}\right\|_{\ell_2}\sqrt{250 \times 4B^2\log(2K) + B^2}$$

$$\leq 32\sqrt{n\log(2K)}B\left\|\boldsymbol{v}\right\|_{\ell_2}.$$

Here, (a) follows from Jensen's inequality, (b) from linearity of expectation and the fact that for $\boldsymbol{x}_i$ with unit Euclidean norm $\boldsymbol{W}\boldsymbol{x}_i \sim \mathcal{N}(\boldsymbol{0}, \boldsymbol{I}_k)$, (c) from simple algebraic manipulations, (d) from the inequalities (6.38) and (6.37). Thus using $t = 18\sqrt{n\log(2K)}B\left\|\boldsymbol{v}\right\|_{\ell_2}$ in (6.36), conditioned on $E_{\boldsymbol{v}}$ we conclude that

$$\left\|\phi\big(\boldsymbol{X}\boldsymbol{W}^T\big)\boldsymbol{v}\right\|_{\ell_2} \leq 50\sqrt{n\log(2K)}B\left\|\boldsymbol{v}\right\|_{\ell_2} = 50\sqrt{n\log(2K)}B\sqrt{k}\frac{\nu\left\|\boldsymbol{y}\right\|_{\ell_2}}{50B\sqrt{K\log(2K)kn}} = \frac{\nu\left\|\boldsymbol{y}\right\|_{\ell_2}}{\sqrt{K}},$$
(6.39)

holds with probability at least $1 - \exp(-102\log(2K)\frac{n}{\left\|\boldsymbol{X}\right\|^2}) \geq 1 - (2K)^{-102}$ where we used $n \geq \left\|\boldsymbol{X}\right\|^2$. Using a union bound over $E_{\boldsymbol{v}}$ and the conditional concentration over $\boldsymbol{W}$, the overall probability of success in (6.39) is at least $1 - (2K)^{-101}$ concluding the proof of (6.35) and the Lemma. ∎

## 6.3 RADEMACHER COMPLEXITY AND GENERALIZATION BOUNDS

In this section we state and prove some Rademacher complexity results that will be used in our generalization bounds. We begin with some basic notation regarding Rademacher complexity. Let $\mathcal{F}$ be a function class. Suppose $f \in \mathcal{F}$ maps $\mathbb{R}^d$ to $\mathbb{R}^K$. Let $\{\boldsymbol{\varepsilon}_i\}_{i=1}^n$ be i.i.d. vectors in $\mathbb{R}^K$ with i.i.d. Rademacher variables. Given i.i.d. samples $\mathcal{S} = \{(\boldsymbol{x}_i, \boldsymbol{y}_i)\}_{i=1}^n \sim \mathcal{D}$, we define the empirical Rademacher complexity to be

$$\mathcal{R}_{\mathcal{S}}(\mathcal{F}) = \frac{1}{n}\mathbb{E}\left[\sup_{f\in\mathcal{F}}\sum_{i=1}^n \boldsymbol{\varepsilon}_i^T f(\boldsymbol{x}_i)\right].$$

We begin by stating a vector contraction inequality by Maurer Maurer (2016). This is obtained by setting $h_i(f(\boldsymbol{x}_i)) = h(\boldsymbol{y}_i, f(\boldsymbol{x}_i))$ in Corollary 4 of Maurer (2016).

**Lemma 6.15** *Let $f(\cdot) : \mathbb{R}^d \to \mathbb{R}^K$ and let $\ell : \mathbb{R}^K \times \mathbb{R}^K \to \mathbb{R}$ be a 1 Lipschitz loss function with respect to second variable. Let $\{\varepsilon_i\}_{i=1}^n$ be i.i.d. Rademacher variables. Given i.i.d. samples $\{(\boldsymbol{x}_i, \boldsymbol{y}_i)\}_{i=1}^n$, define*

$$\mathcal{R}_{\mathcal{S}}(\ell, \mathcal{F}) = \mathbb{E}\left[\sup_{f\in\mathcal{F}}\sum_{i=1}^n \varepsilon_i\ell(\boldsymbol{y}_i, f(\boldsymbol{x}_i))\right].$$

*We have that*

$$\mathcal{R}_{\mathcal{S}}(\ell, \mathcal{F}) \leq \sqrt{2}\mathcal{R}_{\mathcal{S}}(\mathcal{F}).$$

Combining the above result with standard generalization bounds based on Rademacher complexity Bartlett & Mendelson (2002) allows us to prove the following result.

**Lemma 6.16** *Let $\ell(\cdot, \cdot) : \mathbb{R}^K \times \mathbb{R}^K \to [0, 1]$ be a 1 Lipschitz loss function. Given i.i.d. samples $\{(\boldsymbol{x}_i, \boldsymbol{y}_i)\}_{i=1}^n$, consider the empirical loss*

$$\mathcal{L}(f, \ell) = \frac{1}{n}\sum_{i=1}^n \ell(\boldsymbol{y}_i, f(\boldsymbol{x}_i)).$$

*With probability $1 - \delta$ over the samples, for all $f \in \mathcal{F}$, we have that*

$$\mathbb{E}[\mathcal{L}(f, \ell)] \leq \mathcal{L}(f, \ell) + 2\sqrt{2}\mathcal{R}_{\mathcal{S}}(\mathcal{F}) + \sqrt{\frac{5\log(2/\delta)}{n}}$$

**Proof** Based on Bartlett & Mendelson (2002),

$$\mathbb{E}[\mathcal{L}(f,\ell)] \le \mathcal{L}(f,\ell) + 2\mathcal{R}_{\mathcal{S}}(\ell,\mathcal{F}) + \sqrt{\frac{5\log(2/\delta)}{n}}$$

holds with $1 - \delta$ probability. Combining the latter with Lemma 6.15 completes the proof. ∎

**Lemma 6.17** *Consider a neural network model of the form $\boldsymbol{x} \mapsto f(\boldsymbol{x}; \boldsymbol{V}, \boldsymbol{W}) = \boldsymbol{V}\phi(\boldsymbol{W}\boldsymbol{x})$ with $\boldsymbol{W} \in \mathbb{R}^{k \times d}$ and $\boldsymbol{V} \in \mathbb{R}^{K \times k}$ denoting the input and output weight matrices. Suppose $\boldsymbol{V}_0 \in \mathbb{R}^{K \times k}$ is a matrix obeying $\|\boldsymbol{V}_0\|_{\ell_\infty} \le \nu/\sqrt{kK}$. Also let $\boldsymbol{W}_0 \in \mathbb{R}^{k \times d}$ be a reference input weight matrix. Furthermore, we define the neural network function space parameterized by the weights as follows*

$$\mathcal{F}_{\mathcal{V},\mathcal{W}} = \left\{ f(\boldsymbol{x}; \boldsymbol{V}, \boldsymbol{W}) \quad such\ that \quad \boldsymbol{V} \in \mathcal{V} \quad and \quad \boldsymbol{W} \in \mathcal{W} \right\} \quad with \quad \mathcal{V} = \left\{ \boldsymbol{V} : \|\boldsymbol{V} - \boldsymbol{V}_0\|_F \le \frac{\nu M_{\mathcal{V}}}{\sqrt{Kk}} \right\}$$

$$and \quad \mathcal{W} = \left\{ \boldsymbol{W} : \|\boldsymbol{W} - \boldsymbol{W}_0\|_F \le M_{\mathcal{W}} \quad and \quad \|\boldsymbol{W} - \boldsymbol{W}_0\|_{2,\infty} \le \frac{R}{\sqrt{k}} \right\}. \tag{6.40}$$

*Additionally, assume the training data $\{(\boldsymbol{x}_i, \boldsymbol{y}_i)\}_{i=1}^n$ are generated i.i.d. with the input data points of unit Euclidean norm (i.e. $\|\boldsymbol{x}_i\|_{\ell_2} = 1$). Also, define the average energy at $\boldsymbol{W}_0$ as*

$$E = \left( \frac{1}{kn} \sum_{i=1}^n \|\phi(\boldsymbol{W}_0\boldsymbol{x}_i)\|_{\ell_2}^2 \right)^{1/2}.$$

*Also let $\{\boldsymbol{\xi}_i\}_{i=1}^n \in \mathbb{R}^K$ be i.i.d. vectors with i.i.d. Rademacher entries and define the empirical Rademacher complexity*

$$\mathcal{R}_{\mathcal{S}}(\mathcal{F}_{\mathcal{V},\mathcal{W}}) := \frac{1}{n} \mathbb{E}\left[ \sup_{f \in \mathcal{F}_{\mathcal{V},\mathcal{W}}} \sum_{i=1}^n \boldsymbol{\xi}_i^T f(\boldsymbol{x}_i) \right].$$

*Then,*

$$\mathcal{R}_{\mathcal{S}}(\mathcal{F}_{\mathcal{V},\mathcal{W}}) \le \nu B \left( \frac{M_{\mathcal{W}} + E M_{\mathcal{V}}}{\sqrt{n}} + \frac{R^2 + M_{\mathcal{W}} M_{\mathcal{V}}}{\sqrt{k}} \right). \tag{6.41}$$

**Proof** We shall use $\boldsymbol{w}_\ell$ to denote the rows of $\boldsymbol{W}$ (same for $\boldsymbol{W}_0, \boldsymbol{V}, \boldsymbol{V}_0$). We will approximate $\phi(\langle \boldsymbol{w}_\ell, \boldsymbol{x}_i \rangle)$ by its linear approximation $\phi(\langle \boldsymbol{w}_\ell^0, \boldsymbol{x}_i \rangle) + \phi'(\langle \boldsymbol{w}_\ell^0, \boldsymbol{x}_i \rangle)(\langle \boldsymbol{w}_\ell - \boldsymbol{w}_\ell^0, \boldsymbol{x}_i \rangle)$ via the second order Taylor's mean value theorem. We thus have

$$\mathcal{R}_{\mathcal{S}}(\mathcal{F}_{\mathcal{V},\mathcal{W}}) \le \frac{1}{n} \mathbb{E}\left[ \sum_{i=1}^n \boldsymbol{\xi}_i^T \boldsymbol{V}_0 \phi(\boldsymbol{W}_0 \boldsymbol{x}_i) \right]$$

$$+ \underbrace{\frac{1}{n} \mathbb{E}\left[ \sup_{\boldsymbol{W} \in \mathcal{W}} \sum_{i=1}^n \boldsymbol{\xi}_i^T \boldsymbol{V}_0 \mathrm{diag}(\phi'(\boldsymbol{W}_0 \boldsymbol{x}_i))(\boldsymbol{W} - \boldsymbol{W}_0) \boldsymbol{x}_i \right]}_{\mathcal{R}_1}$$

$$+ \underbrace{\frac{1}{2n} \mathbb{E}_{\xi_{i,j} \overset{i.i.d.}{\sim} \pm 1}\left[ \sup_{\boldsymbol{W} \in \mathcal{W}} \sum_{i=1}^n \sum_{\ell=1}^k \sum_{j=1}^K \xi_{i,j}^T \boldsymbol{V}_{j,\ell}^0 \phi''\left((1 - t_{i\ell})\langle \boldsymbol{w}_\ell^0, \boldsymbol{x}_i \rangle + t_{i\ell}\langle \boldsymbol{w}_\ell, \boldsymbol{x}_i \rangle\right)\left(\langle \boldsymbol{w}_\ell - \boldsymbol{w}_\ell^0, \boldsymbol{x}_i \rangle\right)^2 \right]}_{\mathcal{R}_2}$$

$$+ \underbrace{\frac{1}{n} \mathbb{E}\left[ \sup_{\boldsymbol{V} \in \mathcal{V}, \boldsymbol{W} \in \mathcal{W}} \sum_{i=1}^n \boldsymbol{\xi}_i^T (\boldsymbol{V} - \boldsymbol{V}_0)(\phi(\boldsymbol{W}\boldsymbol{x}_i) - \phi(\boldsymbol{W}_0\boldsymbol{x}_i)) \right]}_{\mathcal{R}_3}$$

$$+ \underbrace{\frac{1}{n} \mathbb{E}\left[ \sup_{\boldsymbol{V} \in \mathcal{V}} \sum_{i=1}^n \boldsymbol{\xi}_i^T (\boldsymbol{V} - \boldsymbol{V}_0)\phi(\boldsymbol{W}_0\boldsymbol{x}_i) \right]}_{\mathcal{R}_4}$$

We proceed by bounding each of these four terms. For the first term note that

$$
\begin{aligned}
\mathcal{R}_1 &\leq \frac{1}{n} \mathbb{E}\left[\sup_{\|\boldsymbol{W}-\boldsymbol{W}_0\|_F \leq M_{\mathcal{W}}} \sum_{i=1}^{n} \boldsymbol{\xi}_i^T \boldsymbol{V}_0 \mathrm{diag}\left(\phi'\left(\boldsymbol{W}_0 \boldsymbol{x}_i\right)\right)\left(\boldsymbol{W}-\boldsymbol{W}_0\right)\boldsymbol{x}_i\right] \\
&\leq \frac{1}{n} \mathbb{E}\left[\sup_{\|\boldsymbol{W}-\boldsymbol{W}_0\|_F \leq M_{\mathcal{W}}} \left\langle \sum_{i=1}^{n} \mathrm{diag}\left(\phi'\left(\boldsymbol{W}_0 \boldsymbol{x}_i\right)\right) \boldsymbol{V}_0^T \boldsymbol{\xi}_i \boldsymbol{x}_i^T, \boldsymbol{W}-\boldsymbol{W}_0\right\rangle\right] \\
&\leq \frac{M_{\mathcal{W}}}{n} \mathbb{E}\left[\left\|\left(\sum_{i=1}^{n} \mathrm{diag}\left(\boldsymbol{V}_0^T \boldsymbol{\xi}_i\right)\phi'\left(\boldsymbol{W}_0 \boldsymbol{x}_i\right)\boldsymbol{x}_i^T\right)\right\|_F\right] \\
&\leq \frac{M_{\mathcal{W}}}{n} \mathbb{E}\left[\left\|\left(\sum_{i=1}^{n} \mathrm{diag}\left(\boldsymbol{V}_0^T \boldsymbol{\xi}_i\right)\phi'\left(\boldsymbol{W}_0 \boldsymbol{x}_i\right)\boldsymbol{x}_i^T\right)\right\|_F^2\right]^{1/2} \\
&= \frac{M_{\mathcal{W}}}{n} \left[\sum_{i=1}^{n} \mathbb{E}\left\|\mathrm{diag}\left(\boldsymbol{V}_0^T \boldsymbol{\xi}_i\right)\phi'\left(\boldsymbol{W}_0 \boldsymbol{x}_i\right)\boldsymbol{x}_i^T\right\|_F^2\right]^{1/2} \\
&= \frac{M_{\mathcal{W}}}{n} \left[\sum_{i=1}^{n} \mathbb{E}\left\|\mathrm{diag}\left(\boldsymbol{V}_0^T \boldsymbol{\xi}_i\right)\phi'\left(\boldsymbol{W}_0 \boldsymbol{x}_i\right)\right\|_{\ell_2}^2\right]^{1/2} \\
&\leq \frac{B M_{\mathcal{W}}}{n} \left[\sum_{i=1}^{n} \mathbb{E}\left\|\boldsymbol{V}_0^T \boldsymbol{\xi}_i\right\|_{\ell_2}^2\right]^{1/2} \\
&\leq \frac{B M_{\mathcal{W}}}{n} \|\boldsymbol{V}_0\|_F \\
&\leq \frac{B M_{\mathcal{W}} \nu}{\sqrt{n}},
\end{aligned}
$$

where in the last inequality we used the fact that $\|\boldsymbol{V}_0\|_F \leq \nu$. For the second term note that

$$
\begin{aligned}
\mathcal{R}_2 &\leq \frac{1}{2n} \mathbb{E}\left[\sup_{\|\boldsymbol{W}-\boldsymbol{W}_0\|_{2,\infty} \leq R} \sum_{i=1}^{n}\sum_{\ell=1}^{k}\sum_{j=1}^{K} \xi_{i,j} \boldsymbol{v}_{0,j,\ell} \phi''\left(\left(1-t_{i\ell}\right)\langle \boldsymbol{w}_\ell^0, \boldsymbol{x}_i\rangle + t_{i\ell}\langle \boldsymbol{w}_\ell, \boldsymbol{x}_i\rangle\right)\left(\langle \boldsymbol{w}_\ell - \boldsymbol{w}_\ell^0, \boldsymbol{x}_i\rangle\right)^2\right] \\
&\leq \frac{1}{2n}\sum_{\ell=1}^{k} \mathbb{E}\left[\sup_{\|\boldsymbol{w}_\ell - \boldsymbol{w}_\ell^0\|_{\ell_2} \leq R} \sum_{i=1}^{n}\left|\sum_{j=1}^{K} \xi_{i,j}\boldsymbol{v}_{0,j,\ell}\right|\left|\phi''\left(\left(1-t_{i\ell}\right)\langle \boldsymbol{w}_\ell^0, \boldsymbol{x}_i\rangle + t_{i\ell}\langle \boldsymbol{w}_\ell, \boldsymbol{x}_i\rangle\right)\right|\left(\langle \boldsymbol{w}_\ell - \boldsymbol{w}_\ell^0, \boldsymbol{x}_i\rangle\right)^2\right] \\
&\leq \frac{1}{2kn}\sum_{\ell=1}^{k}\sum_{i=1}^{n} \mathbb{E}\left[\left|\sum_{j=1}^{K} \xi_{i,j}\boldsymbol{v}_{0,j,\ell}\right|\right]R^2 B \\
&\leq \frac{B R^2}{2k}\|\boldsymbol{V}_0^T\|_{2,1} \\
&\leq \frac{B R^2 \nu}{2\sqrt{k}}.
\end{aligned}
$$

In the above we used $\|M\|_{2,1}$ for a matrix $M$ to denote the sum of the Euclidean norm of the rows of $M$. We also used the fact that $\|V_0^T\|_{2,1} \le \nu\sqrt{k}$. To bound the third term note that

$$
\begin{aligned}
\mathcal{R}_3 &= \frac{1}{n}\,\mathbb{E}\left[\sup_{V \in \mathcal{V}, W \in \mathcal{W}} \sum_{i=1}^{n} \boldsymbol{\xi}_i^T (V - V_0)(\phi(Wx_i) - \phi(W_0 x_i))\right] \\
&\le \frac{1}{n}\,\mathbb{E}\left[\sup_{V \in \mathcal{V}, W \in \mathcal{W}} \sum_{i=1}^{n} \|V - V_0\|_F \|\boldsymbol{\xi}_i\|_{\ell_2} \|(\phi(Wx_i) - \phi(W_0 x_i))\|_{\ell_2}\right] \\
&\le \frac{\nu M_{\mathcal{V}}}{n\sqrt{kK}}\,\mathbb{E}\left[\sup_{W \in \mathcal{W}} \sum_{i=1}^{n} \|\boldsymbol{\xi}_i\|_{\ell_2} \|\phi(Wx_i) - \phi(W_0 x_i)\|_{\ell_2}\right] \\
&\le \frac{\nu M_{\mathcal{V}}}{n\sqrt{k}} \cdot \sup_{W \in \mathcal{W}} \sum_{i=1}^{n} \|\phi(Wx_i) - \phi(W_0 x_i)\|_{\ell_2} \\
&\le \frac{\nu B M_{\mathcal{V}}}{n\sqrt{k}} \cdot \sup_{W \in \mathcal{W}} \sum_{i=1}^{n} \|(W - W_0)x_i\|_{\ell_2} \\
&\le \frac{\nu B M_{\mathcal{V}}}{n\sqrt{k}} \cdot \sup_{W \in \mathcal{W}} \sum_{i=1}^{n} \|(W - W_0)\|_F \\
&= \frac{\nu B M_{\mathcal{V}}}{\sqrt{k}} \cdot \sup_{W \in \mathcal{W}} \|(W - W_0)\|_F \\
&= \frac{\nu B M_{\mathcal{V}} M_{\mathcal{W}}}{\sqrt{k}}.
\end{aligned}
$$

Finally, to bound the fourth term note that we have

$$
\begin{aligned}
\mathcal{R}_4 &= \frac{1}{n}\,\mathbb{E}\left[\sup_{V \in \mathcal{V}} \sum_{i=1}^{n} \boldsymbol{\xi}_i^T (V - V_0)\phi(W_0 x_i)\right] \\
&= \frac{1}{n}\,\mathbb{E}\left[\sup_{\|V - V_0\|_F \le \frac{\nu M_{\mathcal{V}}}{\sqrt{kK}}} \left\langle \sum_{i=1}^{n} \boldsymbol{\xi}_i \phi(W_0 x_i)^T, (V - V_0)\right\rangle\right] \\
&= \frac{\nu M_{\mathcal{V}}}{n\sqrt{kK}}\,\mathbb{E}\left[\left\|\sum_{i=1}^{n} \phi(W_0 x_i)\boldsymbol{\xi}_i^T\right\|_F\right] \\
&\le \frac{\nu M_{\mathcal{V}}}{n\sqrt{kK}}\left(\mathbb{E}\left[\left\|\sum_{i=1}^{n} \phi(W_0 x_i)\boldsymbol{\xi}_i^T\right\|_F^2\right]\right)^{1/2} \\
&= \frac{\nu M_{\mathcal{V}}}{n\sqrt{kK}}\left(\sum_{i=1}^{n} \mathbb{E}\left[\|\phi(W_0 x_i)\boldsymbol{\xi}_i^T\|_F^2\right]\right)^{1/2} \\
&= \frac{\nu M_{\mathcal{V}}}{\sqrt{n}}\left(\frac{1}{kn}\sum_{i=1}^{n} \mathbb{E}\left[\|\phi(W_0 x_i)\|_F^2\right]\right)^{1/2} \\
&= \frac{\nu E M_{\mathcal{V}}}{\sqrt{n}} \\
&\le \frac{\nu B E M_{\mathcal{V}}}{\sqrt{n}}.
\end{aligned}
$$

Combining these four bounds we conclude that

$$
\mathcal{R}_{\mathcal{S}}(\mathcal{F}_{\mathcal{V},\mathcal{W}}) \le \nu B\left(\frac{M_{\mathcal{W}}}{\sqrt{n}} + \frac{R^2}{\sqrt{k}} + \frac{M_{\mathcal{V}} M_{\mathcal{W}}}{\sqrt{k}} + \frac{E M_{\mathcal{V}}}{\sqrt{n}}\right),
$$

concluding the proof of (6.41). ∎

Next we state a crucial lemma that connects the test error measured by any Lipschitz loss to that of the quadratic loss on the training data.

**Lemma 6.18** *Consider a one-hidden layer neural network with input to output mapping of the form* $x \in \mathbb{R}^d \mapsto f(x; V, W) = V\phi(Wx) \in \mathbb{R}^K$ *with* $W \in \mathbb{R}^{k \times d}$ *denoting the input-to-hidden weights and*

$V \in \mathbb{R}^{k \times K}$ *the hidden-to-output weights. Suppose* $V_0 \in \mathbb{R}^{K \times k}$ *is a matrix obeying* $\|V_0\|_{\ell_\infty} \le \nu/\sqrt{kK}$. *Also let* $W_0 \in \mathbb{R}^{k \times d}$ *be a reference input weight matrix. Also define the empirical losses*

$$\mathcal{L}(V, W) = \frac{1}{n} \sum_{i=1}^{n} \|y_i - f(x_i; V, W)\|_{\ell_2}^2,$$

*and*

$$\mathcal{L}(f, \ell) = \frac{1}{n} \sum_{i=1}^{n} \ell(f(x_i; V, W), y_i),$$

*with* $\ell : \mathbb{R}^K \times \mathbb{R}^K \to [0, 1]$ *a one Lipschitz loss function obeying* $\ell(y, y) = 0$. *Additionally, assume the training data* $\{(x_i, y_i)\}_{i=1}^n$ *are generated i.i.d. according to a distribution* $\mathcal{D}$ *with the input data points of unit Euclidean norm (i.e.* $\|x_i\|_{\ell_2} = 1$). *Also, define the average energy at* $W_0$ *as*

$$E = \left( \frac{1}{kn} \sum_{i=1}^{n} \|\phi(W_0 x_i)\|_{\ell_2}^2 \right)^{1/2}.$$

*Then for all* $f$ *in the function class* $\mathcal{F}_{\mathcal{W}}$ *given by* (6.40)

$$\mathbb{E}[\mathcal{L}(f, \ell)] \le \sqrt{\mathcal{L}(V, W)} + 2\sqrt{2}\nu B \left( \frac{M_{\mathcal{W}}}{\sqrt{n}} + \frac{R^2}{\sqrt{k}} + \frac{\nu B M_{\mathcal{V}} M_{\mathcal{W}}}{\sqrt{k}} + \frac{E M_{\mathcal{V}}}{\sqrt{n}} \right) + \sqrt{\frac{5 \log(2/\delta)}{n}},$$
$$(6.42)$$

*holds with probability at least* $1 - \delta$. *Furthermore, Suppose labels are one-hot encoded and thus unit Euclidian norm. Given a sample* $(x, y) \in \mathbb{R}^d \times \mathbb{R}^K$ *generated according to the distribution* $\mathcal{D}$, *define the population classification error*

$$Err_{\mathcal{D}}(W) = \mathbb{P}(\arg\max_{1 \le \ell \le K} y_i \ne \arg\max_{1 \le \ell \le K} f_i(x, W)).$$

*Then, we also have*

$$Err_{\mathcal{D}}(W) \le 2 \left[ \sqrt{\mathcal{L}(V, W)} + 2\sqrt{2}\nu B \left( \frac{M_{\mathcal{W}}}{\sqrt{n}} + \frac{R^2}{\sqrt{k}} + \frac{\nu B M_{\mathcal{V}} M_{\mathcal{W}}}{\sqrt{k}} + \frac{E M_{\mathcal{V}}}{\sqrt{n}} \right) + \sqrt{\frac{5 \log(2/\delta)}{n}} \right].$$
$$(6.43)$$

**Proof** To begin first note that any 1-Lipschitz $\ell$ with $\ell(y, y) = 0$ obeys $\ell(y, \hat{y}) \le \|y - \hat{y}\|_{\ell_2}$. Thus, we have

$$\mathcal{L}(f, \ell) \le \frac{1}{n} \sum_{i=1}^{n} \|y_i - f(x_i; V, W)\|_{\ell_2} \le \sqrt{\mathcal{L}(V, W)},$$

where the last inequality follows from Cauchy-Schwarz. Consequently, applying Lemmas 6.16 and 6.17 we conclude that

$$\mathbb{E}[\mathcal{L}(f, \ell)] \le \mathcal{L}(f, \ell) + 2\sqrt{2} \cdot \mathcal{R}_{\mathcal{S}}(\mathcal{F}) + \sqrt{\frac{5 \log(2/\delta)}{n}},$$

$$\le \sqrt{\mathcal{L}(W)} + 2\sqrt{2}\nu B \left( \frac{M_{\mathcal{W}}}{\sqrt{n}} + \frac{R^2}{\sqrt{k}} + \frac{\nu B M_{\mathcal{V}} M_{\mathcal{W}}}{\sqrt{k}} + \frac{E M_{\mathcal{V}}}{\sqrt{n}} \right) + \sqrt{\frac{5 \log(2/\delta)}{n}},$$

which yields the first statement.

To prove the second statement on classification accuracy, we pick the $\ell$ function as follows

$$\ell(y, \hat{y}) = \min(1, \|y - \hat{y}\|_{\ell_2}).$$

Note that, given a sample $(x, y) \in \mathbb{R}^d \times \mathbb{R}^K$ with one-hot encoded labels, if

$$\arg\max_{1 \le \ell \le K} y_\ell \ne \arg\max_{1 \le \ell \le K} f_\ell(x; V, W),$$

this implies

$$\ell(y, f(x; V, W)) \ge 0.5.$$

Combining the latter with Markov inequality we arrive at

$$\text{Err}_{\mathcal{D}}(W) \le 2 \mathbb{E}_{(x,y) \sim \mathcal{D}}[\ell(y, f(x; V, W))] = 2 \mathbb{E}[\mathcal{L}(\ell, W)].$$

Now since $\ell$ is 1 Lipschitz and bounded, it obeys (6.42), which combined with the above identity yields (6.43), completing the proof. ∎

### 6.4 Proofs for neural nets with arbitrary initialization (Proof of Theorem 2.3)

In this section we prove Theorem 2.3. We first discuss a preliminary optimization result in Section 6.4.1. Next, in Section 6.4.2 we build upon this result to prove our main optimization result. Finally, in Section 6.4.3 we use these optimization results to prove our main generalization result, completing the proof of Theorem 2.3.

### 6.4.1 Preliminary Optimization Result

**Lemma 6.19 (Deterministic convergence guarantee)** *Consider a one-hidden layer neural net of the form* $x \mapsto f(x; W) := V\phi(Wx)$ *with input weights* $W \in \mathbb{R}^{k \times d}$ *and output weights* $V \in \mathbb{R}^{K \times k}$ *and an activation* $\phi$ *obeying* $|\phi(0)| \le B$, $|\phi'(z)| \le B$, *and* $|\phi''(z)| \le B$ *for all* $z$. *Also assume* $V$ *is fixed with all entries bounded by* $\|V\|_{\ell_\infty} \le \frac{\nu}{\sqrt{kK}}$ *and we train over* $W$ *based on the loss*

$$\mathcal{L}(W) = \frac{1}{2} \sum_{i=1}^{n} \|f(x_i; W) - y_i\|_{\ell_2}^2.$$

*Also, consider a point* $W_0 \in \mathbb{R}^{k \times d}$ *with* $J$ *an* $(\epsilon_0, \nu B\|X\|)$ *reference Jacobian associated with* $\mathcal{J}(W_0)$ *per Definition 5.1. Furthermore, define the information* $\mathcal{I}$ *and nuisance* $\mathcal{N}$ *subspaces and the truncated Jacobian* $J_\mathcal{I}$ *associated with the reference Jacobian* $J$ *based on a cut-off spectrum value of* $\alpha$ *per Definition 5.2. Let the initial residual vector be* $r_0 = y - f(W_0) \in \mathbb{R}^{nK}$. *Furthermore, assume*

$$\varepsilon_0 \le \frac{\alpha}{5} \min\left(\delta, \sqrt{\frac{\delta\alpha}{\Gamma\nu B\|X\|}}\right) \tag{6.44}$$

*and*

$$k \ge 400 \frac{\nu^6 B^6 \|X\|^6 \Gamma^2}{\delta^2 \alpha^8} \left(\mathcal{B}_{\alpha,\Gamma} + \delta\Gamma\|r_0\|_{\ell_2}\right)^2, \tag{6.45}$$

*with* $0 \le \delta \le 1$ *and* $\Gamma \ge 1$. *We run gradient descent iterations of the form* $W_{\tau+1} = W_\tau - \eta\nabla\mathcal{L}(W_\tau)$ *starting from* $W_0$ *with step size* $\eta$ *obeying* $\eta \le \frac{1}{\nu^2 B^2\|X\|^2}$. *Then for all iterates* $\tau$ *obeying* $0 \le \tau \le T :=$ $\frac{\Gamma}{\eta\alpha^2}$

$$\|W_\tau - W_0\|_F \le \frac{\mathcal{B}_{\alpha,\Gamma}}{\alpha} + \delta\frac{\Gamma}{\alpha}\|r_0\|_{\ell_2}. \tag{6.46}$$

$$\|W_\tau - W_0\|_{2,\infty} \le \frac{2\nu B\Gamma\|X\|}{\sqrt{k}\alpha^2}\|r_0\|_{\ell_2}. \tag{6.47}$$

*Furthermore, after* $\tau = T$ *iteration we have*

$$\|r_T\|_{\ell_2} \le e^{-\Gamma}\|\Pi_\mathcal{I}(r_0)\|_{\ell_2} + \|\Pi_\mathcal{N}(r_0)\|_{\ell_2} + \frac{\delta\alpha}{\nu B\|X\|}\|r_0\|_{\ell_2}. \tag{6.48}$$

**Proof** To prove this lemma we wish to apply Theorem 5.3. We thus need to ensure that the assumptions of this theorem are satisfied. To do this note that by Lemma 6.10 Assumption 1 holds with $\beta = \nu B\|X\|$. Furthermore, we pick $\varepsilon = \frac{\delta\alpha^3}{5\Gamma\nu^2 B^2\|X\|^2} = \frac{\delta\alpha^3}{5\Gamma\beta^2}$ which together with (6.44) guarantees (5.13) holds. We now turn our attention to verifying Assumption 2. To this aim note that for all $W \in \mathbb{R}^{k \times d}$ obeying

$$\|W - W_0\|_F \le R := 2\left(\frac{\mathcal{B}_{\alpha,\Gamma}}{\alpha} + \delta\frac{\Gamma}{\alpha}\|r_0\|_{\ell_2}\right)$$

as long as (6.45) holds by Lemma 6.10 we have

$$
\begin{aligned}
\|\mathcal{J}(\boldsymbol{W}) - \mathcal{J}(\boldsymbol{W}_0)\| &\leq B\sqrt{K}\,\|\boldsymbol{V}\|_{\ell_\infty}\,\|\boldsymbol{X}\|\,R \\
&\leq \frac{\nu}{\sqrt{k}}B\,\|\boldsymbol{X}\|\,R \\
&= \frac{\delta\alpha^3}{10\Gamma\nu^2 B^2\,\|\boldsymbol{X}\|^2}\frac{\frac{20\Gamma\nu^3 B^3\|\boldsymbol{X}\|^3}{\delta\alpha^4}\left(\mathcal{B}_{\alpha,\Gamma} + \delta\Gamma\|\boldsymbol{r}_0\|_{\ell_2}\right)}{\sqrt{k}} \\
&\leq \frac{\delta\alpha^3}{10\Gamma\nu^2 B^2\,\|\boldsymbol{X}\|^2} \\
&= \frac{\varepsilon}{2}.
\end{aligned}
$$

Thus, Assumption 2 holds with $\|\boldsymbol{W} - \boldsymbol{W}_0\|_F \leq R := 2\left(\frac{\mathcal{B}_{\alpha,\Gamma}}{\alpha} + \delta\frac{\Gamma}{\alpha}\|\boldsymbol{r}_0\|_{\ell_2}\right)$. Now that we have verified that the assumptions of Theorem 5.3 hold so do its conclusions and thus (6.46) and (6.48) hold.

We now turn our attention to proving the row-wise bound (6.47). To this aim let $\boldsymbol{w}_\ell^{(\tau)}$ denote the $\ell$th row of $\boldsymbol{W}_\tau$. Also note that

$$
\nabla\mathcal{L}(\boldsymbol{w}_\ell) = \ell\text{th row of } \mathrm{mat}(\mathcal{J}(\boldsymbol{W})^T\boldsymbol{r}_\tau).
$$

Hence, using Lemma 6.10 equation (6.32) we conclude that

$$
\left\|\nabla\mathcal{L}(\boldsymbol{w}_\ell^{(\tau)})\right\|_{\ell_2} \leq B\sqrt{K}\,\|\boldsymbol{V}\|_{\ell_\infty}\,\|\boldsymbol{X}\|\,\|\boldsymbol{r}_\tau\|_{\ell_2} \leq \frac{\nu B\,\|\boldsymbol{X}\|}{\sqrt{k}}\|\boldsymbol{r}_\tau\|_{\ell_2}.
$$

Consequently, for any row $1 \leq \ell \leq k$, we have

$$
\left\|\boldsymbol{w}_\ell^{(\tau)} - \boldsymbol{w}_\ell^{(0)}\right\|_{\ell_2} \leq \eta\frac{\nu B\,\|\boldsymbol{X}\|}{\sqrt{k}}\sum_{t=0}^{\tau-1}\|\boldsymbol{r}_t\|_{\ell_2}. \tag{6.49}
$$

To bound the right-hand side we use the triangular inequality combined with (6.25) and (6.26) to conclude that

$$
\begin{aligned}
\eta\sum_{t=0}^{\tau-1}\|\boldsymbol{r}_\tau\|_{\ell_2} &\leq \eta\sum_{t=0}^{\tau-1}\|\widetilde{\boldsymbol{r}}_\tau\|_{\ell_2} + \eta\sum_{t=0}^{\tau-1}\|\boldsymbol{r}_\tau - \widetilde{\boldsymbol{r}}_\tau\|_{\ell_2} \\
&\leq \frac{\Gamma}{\alpha^2}\|\boldsymbol{r}_0\|_{\ell_2} + \frac{2\Gamma^2(\varepsilon\beta + \varepsilon_0^2)}{\alpha^4}\|\boldsymbol{r}_0\|_{\ell_2} \\
&= \frac{2\Gamma(\varepsilon_0^2 + \varepsilon\beta) + \alpha^2}{\alpha^4}\Gamma\|\boldsymbol{r}_0\|_{\ell_2} \\
&\leq 2\frac{\Gamma}{\alpha^2}\|\boldsymbol{r}_0\|_{\ell_2}, \tag{6.50}
\end{aligned}
$$

where in the last inequality we used the fact that $\varepsilon_0^2 \leq \frac{\alpha^2}{25\Gamma}$ per (6.44) and $\epsilon\beta = \frac{\delta\alpha^3}{5\Gamma\beta} \leq \frac{\alpha^2}{5\Gamma}$ per our choice of $\epsilon$. Combining (6.49) and (6.50), we obtain

$$
\left\|\boldsymbol{w}_\ell^{(\tau)} - \boldsymbol{w}_\ell^{(0)}\right\|_{\ell_2} \leq \frac{2\nu B\,\|\boldsymbol{X}\|\,\Gamma}{\sqrt{k}\alpha^2}\|\boldsymbol{r}_0\|_{\ell_2},
$$

completing the proof of (6.47) and the theorem. ∎

### 6.4.2 MAIN OPTIMIZATION RESULT

**Lemma 6.20 (Deterministic optimization guarantee)** *Consider the setting and assumptions of Lemma 6.19. Also assume $\|\Pi_{\mathcal{I}}(\boldsymbol{r}_0)\|_{\ell_2} \geq c\|\boldsymbol{r}_0\|_{\ell_2}$ for a constant $c > 0$ if $\varepsilon_0 > 0$. Furthermore, assume*

$$
\varepsilon_0^2 \leq \frac{\alpha^2}{25}\min\left(c\frac{\mathcal{B}_{\alpha,\Gamma}\alpha}{\nu B\Gamma^2\|\boldsymbol{r}_0\|_{\ell_2}\,\|\boldsymbol{X}\|}, \frac{\zeta^2\nu^2 B^2\,\|\boldsymbol{X}\|^2}{\alpha^2}, \frac{\zeta}{\Gamma}\right), \tag{6.51}
$$

*and*

$$k \geq 1600 \left( \frac{\alpha}{\zeta \nu B \|X\|} + \frac{\Gamma \|r_0\|_{\ell_2}}{\mathcal{B}_{\alpha,\Gamma}} \right)^2 \frac{\nu^6 B^6 \|X\|^6 \Gamma^2 \mathcal{B}_{\alpha,\Gamma}^2}{\alpha^8}, \tag{6.52}$$

*and $\Gamma \geq 1$. We run gradient descent iterations of the form $W_{\tau+1} = W_\tau - \eta \nabla \mathcal{L}(W_\tau)$ starting from $W_0$ with step size $\eta$ obeying $\eta \leq \frac{1}{\nu^2 B^2 \|X\|^2}$. Then for all iterates $\tau$ obeying $0 \leq \tau \leq T := \frac{\Gamma}{\eta \alpha^2}$*

$$\|W_\tau - W_0\|_F \leq \frac{2\mathcal{B}_{\alpha,\Gamma}}{\alpha}. \tag{6.53}$$

$$\|W_\tau - W_0\|_{2,\infty} \leq \frac{2\nu B \Gamma \|X\|}{\sqrt{k}\alpha^2} \|r_0\|_{\ell_2}. \tag{6.54}$$

*Furthermore, after $\tau = T$ iteration we have*

$$\|f(W_T) - y\|_{\ell_2} \leq e^{-\Gamma} \|\Pi_{\mathcal{I}}(r_0)\|_{\ell_2} + \|\Pi_{\mathcal{N}}(r_0)\|_{\ell_2} + \zeta \|r_0\|_{\ell_2}. \tag{6.55}$$

**Proof** To prove this lemma we aim to substitute

$$\delta = \min \left( \frac{\zeta \nu B \|X\|}{\alpha}, \frac{\mathcal{B}_{\alpha,\Gamma}}{\Gamma \|r_0\|_{\ell_2}} \right) \leq 1, \tag{6.56}$$

in Theorem 6.19. To do this we need to verify the assumptions of Theorem 6.19. To this aim note that the choice of $\delta$ from (6.56) combined with (6.52) ensures that

$$\begin{aligned}
k &\geq 1600 \left( \frac{\alpha}{\zeta \nu B \|X\|} + \frac{\Gamma \|r_0\|_{\ell_2}}{\mathcal{B}_{\alpha,\Gamma}} \right)^2 \frac{\nu^6 B^6 \|X\|^6 \Gamma^2 \mathcal{B}_{\alpha,\Gamma}^2}{\alpha^8} \\
&\geq \max \left( \frac{\alpha}{\zeta \nu B \|X\|}, \frac{\Gamma \|r_0\|_{\ell_2}}{\mathcal{B}_{\alpha,\Gamma}} \right)^2 \frac{1600 \nu^6 B^6 \|X\|^6 \Gamma^2 \mathcal{B}_{\alpha,\Gamma}^2}{\alpha^8} \\
&\geq \frac{1}{\min \left( \frac{\zeta \nu B \|X\|}{\alpha}, \frac{\mathcal{B}_{\alpha,\Gamma}}{\Gamma \|r_0\|_{\ell_2}} \right)^2} \frac{1600 \nu^6 B^6 \|X\|^6 \Gamma^2 \mathcal{B}_{\alpha,\Gamma}^2}{\alpha^8} \\
&= \frac{1600 \Gamma^2 \nu^6 B^6 \|X\|^6 \mathcal{B}_{\alpha,\Gamma}^2}{\delta^2 \alpha^8} \\
&= 400 \frac{\Gamma^2 \nu^6 B^6 \|X\|^6 (\mathcal{B}_{\alpha,\Gamma} + \mathcal{B}_{\alpha,\Gamma})^2}{\delta^2 \alpha^8} \\
&\geq 400 \frac{\Gamma^2 \nu^6 B^6 \|X\|^6 (\mathcal{B}_{\alpha,\Gamma} + \delta \Gamma \|r_0\|_{\ell_2})^2}{\delta^2 \alpha^8},
\end{aligned}$$

so that (6.45) holds. We thus turn our attention to proving (6.44). If $\varepsilon_0 = 0$, the statement already holds. Otherwise, note that based on Lemma 6.2 equation (6.3) we have

$$\mathcal{B}_{\alpha,\Gamma} \geq \frac{\alpha \|\Pi_{\mathcal{I}}(r_0)\|_{\ell_2}}{\lambda_1} \geq \frac{\alpha \|\Pi_{\mathcal{I}}(r_0)\|_{\ell_2}}{\nu B \|X\|} \geq \frac{c\alpha \|r_0\|_{\ell_2}}{\nu B \|X\|} \quad \Rightarrow \quad \frac{\mathcal{B}_{\alpha,\Gamma}}{c \|r_0\|_{\ell_2}} \geq \frac{\alpha}{\nu B \|X\|}. \tag{6.57}$$

Recall that $\alpha = \frac{\nu \alpha_0}{\sqrt{K}}$, which implies that

- If $\delta = \frac{\zeta \nu B \|X\|}{\alpha}$: For (6.44) to hold it suffices to have $\varepsilon_0 \leq \frac{\alpha}{5} \min \left( \frac{\zeta \nu B \|X\|}{\alpha}, \sqrt{\frac{\zeta}{\Gamma}} \right)$.

- If $\delta = \frac{\mathcal{B}_{\alpha,\Gamma}}{\Gamma\|r_0\|_{\ell_2}}$: For (6.44) to hold it suffices to have $\varepsilon_0 \leq \frac{\alpha}{5}\sqrt{c\frac{\mathcal{B}_{\alpha,\Gamma}\alpha}{\nu B\Gamma^2\|r_0\|_{\ell_2}\|X\|}}$ as based on

  (6.57) we have $\sqrt{\frac{c\alpha}{\beta\Gamma}} = \sqrt{\frac{c\alpha}{\nu B\Gamma\|X\|}} \leq \sqrt{\delta}$

$$\varepsilon_0 \leq \frac{\alpha}{5}\sqrt{\frac{c\mathcal{B}_{\alpha,\Gamma}\alpha}{\nu B\Gamma^2\|r_0\|_{\ell_2}\|X\|}}$$
$$= \frac{\alpha}{5}\sqrt{\delta}\sqrt{\frac{c\alpha}{\Gamma\beta}}$$
$$= \frac{\alpha}{5}\sqrt{\delta}\cdot\min\left(\sqrt{\delta},\sqrt{\frac{c\alpha}{\Gamma\beta}}\right)$$
$$= \frac{\alpha}{5}\min\left(\delta,\sqrt{\frac{\delta\alpha}{\Gamma\beta}}\right)$$

Combining the latter two cases as long as

$$\varepsilon_0^2 \leq \frac{\alpha^2}{25}\min\left(c\frac{\mathcal{B}_{\alpha,\Gamma}\alpha}{\nu B\Gamma^2\|r_0\|_{\ell_2}\|X\|}, \frac{\zeta^2\nu^2 B^2\|X\|^2}{\alpha^2}, \frac{\zeta}{\Gamma}\right) \quad\Leftrightarrow\quad (6.51),$$

then (6.44) holds. As a result when (6.51) and (6.52) hold with $\delta = \min\left(\frac{\zeta\nu B\|X\|}{\alpha}, \frac{\mathcal{B}_{\alpha,\Gamma}}{\Gamma\|r_0\|_{\ell_2}}\right)$ both assumptions of Theorem 6.19 also hold and so do its conclusions. In particular, (6.53) follows from (6.46) by noting that based on our choice of $\delta$ we have $\delta\frac{\Gamma}{\alpha}\|r_0\|_{\ell_2} \leq \frac{\mathcal{B}_{\alpha,\Gamma}}{\alpha}$, (6.54) follows immediately from (6.47), and (6.55) follows from (6.48) by noting that based on our choice of $\delta$ we have $\frac{\delta\alpha}{\nu B\|X\|} \leq \zeta$. ∎

### 6.4.3 MAIN GENERALIZATION RESULT (COMPLETING THE PROOF OF THEOREM 2.3)

We state the following rigorous and stronger version of Theorem 2.3. The differences (besides constant terms) are the use of $\|X\|$ rather than $\sqrt{n}$ (we always have $\|X\| \leq \sqrt{n}$) and use of the tighter bound $\|J_\mathcal{I}^\dagger r_0\|_{\ell_2}$ rather than $\|\Pi_\mathcal{I}(r_0)\|_{\ell_2}/\alpha$.

**Theorem 6.21** *Let $\zeta, \Gamma, \bar{\alpha}$ be scalars obeying $\zeta \leq 1/2$, $\Gamma \geq 1$, and $\bar{\alpha} \geq 0$ which determine the overall precision, cut-off and learning duration, respectively.[4] Consider a training data set $\{(x_i, y_i)\}_{i=1}^n \in \mathbb{R}^d \times \mathbb{R}^K$ generated i.i.d. according to a distribution $\mathcal{D}$ where the input samples have unit Euclidean norm. Also consider a neural net with $k$ hidden nodes as described in (1.1) parameterized by $W$ where the activation function $\phi$ obeys $|\phi'(z)|, |\phi''(z)| \leq B$. Let $W_0$ be an arbitrary initial weight matrix. Also assume the output matrix has bounded entries obeying $\|V\|_{\ell_\infty} \leq \frac{\nu}{\sqrt{kK}}$. Furthermore, set $J := \mathcal{J}(W_0)$ and define the information $\mathcal{I}$ and nuisance $\mathcal{N}$ subspaces and the truncated Jacobian $J_\mathcal{I}$ associated with the reference/initial Jacobian $J$ based on a cut-off spectrum value $\alpha = \nu B\bar{\alpha}\sqrt[4]{n}\sqrt{\|X\|}$. Also define the initial residual $r_0 = f(W_0) - y \in \mathbb{R}^{nK}$ and pick $C_r > 0$ so that $\frac{\|r_0\|_{\ell_2}}{\sqrt{n}} \leq C_r$. Suppose number of hidden nodes $k$ obeys*

$$k \gtrsim \frac{C_r^2\Gamma^4}{\bar{\alpha}^8\nu^2\zeta^2}, \tag{6.58}$$

*with $\Gamma \geq 1$ and tolerance level $\zeta$. Run gradient descent updates (1.5) with learning rate $\eta \leq \frac{1}{\nu^2 B^2\|X\|^2}$. Then, after $T = \frac{\Gamma}{\eta\alpha^2}$ iterations, training loss obeys*

$$\|f(W_T) - y\|_{\ell_2} \leq \|\Pi_\mathcal{N}(r_0)\|_{\ell_2} + C_r\left(e^{-\Gamma} + \zeta\right)\sqrt{n}.$$

*and with probability at least $1 - \delta$, the generalization error obeys*

$$Err_\mathcal{D}(W_T) \leq \underbrace{\frac{2\|\Pi_\mathcal{N}(r_0)\|_{\ell_2}}{\sqrt{n}}}_{bias\ term} + \underbrace{\frac{12\nu B}{\sqrt{n}}\left(\|J_\mathcal{I}^\dagger r_0\|_{\ell_2} + \frac{\Gamma}{\alpha}\|\Pi_\mathcal{N}(r_0)\|_{\ell_2}\right)}_{variance\ term} + 5\sqrt{\frac{\log(2/\delta)}{n}} + 2C_r(e^{-\Gamma} + \zeta).$$

---

[4]Note that this theorem and its conclusions hold for any choice of these parameters in the specified range.

Theorem 6.21 immediately follows from Theorem 6.22 below by upper bounding $\mathcal{D}_{\alpha,\Gamma}$ (see Definition 6.1) using Lemma 6.2 equation (6.2).

**Theorem 6.22** *Consider a training data set $\{(\boldsymbol{x}_i, \boldsymbol{y}_i)\}_{i=1}^{n} \in \mathbb{R}^d \times \mathbb{R}^K$ generated i.i.d. according to a distribution $\mathcal{D}$ where the input samples have unit Euclidean norm. Also consider a neural net with $k$ hidden nodes as described in (1.1) parameterized by $\boldsymbol{W}$ where the activation function $\phi$ obeys $|\phi'(z)|, |\phi''(z)| \le B$. Let $\boldsymbol{W}_0$ be an arbitrary initial weight matrix. Also assume the output matrix has bounded entries obeying $\|\boldsymbol{V}\|_{\ell_\infty} \le \frac{\nu}{\sqrt{kK}}$. Furthermore, set $\boldsymbol{J} := \mathcal{J}(\boldsymbol{W}_0)$ and define the information $\mathcal{I}$ and nuisance $\mathcal{N}$ subspaces and the truncated Jacobian $\boldsymbol{J}_{\mathcal{I}}$ associated with the reference/initial Jacobian $\boldsymbol{J}$ based on a cut-off spectrum value $\alpha = \nu B \bar{\alpha} \sqrt[4]{n} \sqrt{\|\boldsymbol{X}\|}$. Also define the initial residual $\boldsymbol{r}_0 = f(\boldsymbol{W}_0) - \boldsymbol{y} \in \mathbb{R}^{nK}$ and pick $C_r > 0$ so that $\frac{\|\boldsymbol{r}_0\|_{\ell_2}}{\sqrt{n}} \le C_r$. Also assume, the number of hidden nodes $k$ obeys*

$$k \ge 25600 \frac{C_r^2 \Gamma^4}{\bar{\alpha}^8 \nu^2 B^2 \zeta^2}, \tag{6.59}$$

*with $\Gamma \ge 1$ and tolerance level $\zeta \le 2$. Run gradient descent updates (1.5) with learning rate $\eta \le \frac{1}{\nu^2 B^2 \|\boldsymbol{X}\|^2}$. Then, after $T = \frac{\Gamma}{\eta \alpha^2}$ iterations, with probability at least $1 - \delta$, the generalization error obeys*

$$Err_{\mathcal{D}}(\boldsymbol{W}_T) \le \underbrace{\frac{2\|\Pi_{\mathcal{N}}(\boldsymbol{r}_0)\|_{\ell_2}}{\sqrt{n}}}_{bias\ term} + \underbrace{\frac{12\nu B \mathcal{D}_{\alpha,\Gamma}}{\sqrt{n}}}_{variance\ term} + 5\sqrt{\frac{\log(2/\delta)}{n}} + 2C_r(e^{-\Gamma} + \zeta), \tag{6.60}$$

*where $\mathcal{D}_{\alpha,\Gamma}$ is the early stopping distance as in Def. (6.1).*

**Proof** First, note that using $\alpha \le \beta = \nu B \|\boldsymbol{X}\|$, $\frac{\mathcal{B}_{\alpha,\Gamma}}{\Gamma} \le \|\boldsymbol{r}_0\|_{\ell_2}$ per (6.2), and $\|\boldsymbol{r}_0\|_{\ell_2} \le C_r \sqrt{n}$ we have

$$\frac{\alpha}{\nu B \|\boldsymbol{X}\|} \le 1 \le \frac{\Gamma \|\boldsymbol{r}_0\|_{\ell_2}}{\mathcal{B}_{\alpha,\Gamma}} \le C_r \frac{\Gamma \sqrt{n}}{\mathcal{B}_{\alpha,\Gamma}}.$$

This together with $\zeta \le 2$ implies that

$$\frac{\alpha}{\frac{\zeta}{2}\nu B \|\boldsymbol{X}\|} + \frac{\Gamma \|\boldsymbol{r}_0\|_{\ell_2}}{\mathcal{B}_{\alpha,\Gamma}} \le \frac{\alpha}{\frac{\zeta}{2}\nu B \|\boldsymbol{X}\|} + \frac{\Gamma \|\boldsymbol{r}_0\|_{\ell_2}}{\frac{\zeta}{2}\mathcal{B}_{\alpha,\Gamma}}$$

$$\le 2\frac{\Gamma \|\boldsymbol{r}_0\|_{\ell_2}}{\frac{\zeta}{2}\mathcal{B}_{\alpha,\Gamma}}$$

$$\le 2\frac{C_r}{\frac{\zeta}{2}}\frac{\Gamma}{\mathcal{B}_{\alpha,\Gamma}}\sqrt{n}. \tag{6.61}$$

Thus when

$$k \ge 25600 \frac{C_r^2 \Gamma^4}{\bar{\alpha}^8 \nu^2 B^2 \zeta^2}$$

$$= 6400 \left(\frac{C_r \Gamma}{\frac{\zeta}{2}}\right)^2 \frac{n^2 \Gamma^2 \nu^6 B^6 \|\boldsymbol{X}\|^4}{\alpha^8}$$

$$\overset{\sqrt{n} \ge \|\boldsymbol{X}\|}{\ge} 6400 \left(\frac{C_r \Gamma}{\frac{\zeta}{2}}\right)^2 \frac{n \nu^6 B^6 \|\boldsymbol{X}\|^6}{\alpha^8}$$

$$= 1600 \left(2\frac{C_r}{\frac{\zeta}{2}}\frac{\Gamma}{\mathcal{B}_{\alpha,\Gamma}}\sqrt{n}\right)^2 \frac{\mathcal{B}_{\alpha,\Gamma}^2 \nu^6 B^6 \|\boldsymbol{X}\|^6}{\alpha^8}$$

$$\overset{(6.61)}{\ge} 1600 \left(\frac{\alpha}{\frac{\zeta}{2}\nu B \|\boldsymbol{X}\|} + \frac{\Gamma \|\boldsymbol{r}_0\|_{\ell_2}}{\mathcal{B}_{\alpha,\Gamma}}\right)^2 \frac{\mathcal{B}_{\alpha,\Gamma}^2 \nu^6 B^6 \|\boldsymbol{X}\|^6}{\alpha^8}$$

Thus, (6.52) holds. Also (6.51) trivially holds for $\varepsilon_0$. Thus applying Theorem 6.20 with $\varepsilon_0 = 0$ the following three conclusions hold

$$\|\boldsymbol{W}_\tau - \boldsymbol{W}_0\|_F \le \frac{2\mathcal{B}_{\alpha,\Gamma}}{\alpha} = 2\mathcal{D}_{\alpha,\Gamma}. \tag{6.62}$$

and

$$\|\boldsymbol{W}_\tau - \boldsymbol{W}_0\|_{2,\infty} \le \frac{2\nu B\Gamma \|\boldsymbol{X}\|}{\sqrt{k}\alpha^2} \|\boldsymbol{r}_0\|_{\ell_2}$$
$$\overset{\|\boldsymbol{r}_0\|_{\ell_2} \le C_r\sqrt{n}}{\le} \frac{2\sqrt{n}C_r\nu B\Gamma \|\boldsymbol{X}\|}{\sqrt{k}\alpha^2}. \tag{6.63}$$

and

$$\|f(\boldsymbol{W}_T) - \boldsymbol{y}\|_{\ell_2} \le e^{-\Gamma}\|\Pi_{\mathcal{I}}(\boldsymbol{r}_0)\|_{\ell_2} + \|\Pi_{\mathcal{N}}(\boldsymbol{r}_0)\|_{\ell_2} + \frac{\zeta}{2}\|\boldsymbol{r}_0\|_{\ell_2}$$
$$\le \|\Pi_{\mathcal{N}}(\boldsymbol{r}_0)\|_{\ell_2} + \left(e^{-\Gamma} + \frac{\zeta}{2}\right)\|\boldsymbol{r}_0\|_{\ell_2}$$
$$\overset{\|\boldsymbol{r}_0\|_{\ell_2} \le C_r\sqrt{n}}{\le} \|\Pi_{\mathcal{N}}(\boldsymbol{r}_0)\|_{\ell_2} + C_r\left(e^{-\Gamma} + \frac{\zeta}{2}\right)\sqrt{n}. \tag{6.64}$$

Furthermore, using the assumption that $\|\boldsymbol{V}\|_{\ell_\infty} \le \frac{\nu}{\sqrt{kK}}$ Lemma 6.18 applies and hence equation (6.43) with $\boldsymbol{W} = \boldsymbol{W}_T$, $\sqrt{\mathcal{L}(\boldsymbol{W}_T)} = \frac{\|f(\boldsymbol{W}_T)-\boldsymbol{y}\|_{\ell_2}}{\sqrt{n}}$, $M_{\mathcal{W}} = 2\mathcal{D}_{\alpha,\Gamma}$, $M_{\mathcal{V}} = 0$, and $R = \frac{2\sqrt{n}C_r\nu B\Gamma\|\boldsymbol{X}\|}{\alpha^2}$ implies that

$$\text{Err}_{\mathcal{D}}(\boldsymbol{W}_T) \le 2\left[\frac{\|f(\boldsymbol{W}_T) - \boldsymbol{y}\|_{\ell_2}}{\sqrt{n}} + 3\nu B\left(\frac{2\mathcal{D}_{\alpha,\Gamma}}{\sqrt{n}} + \frac{R^2}{\sqrt{k}}\right) + \sqrt{\frac{5\log(2/\delta)}{n}}\right] \tag{6.65}$$

Also note that using (6.59) we have

$$\frac{3\nu B R^2}{\sqrt{k}} \le \frac{12 C_r^2 \Gamma^2 \nu^3 B^3 n \|\boldsymbol{X}\|^2}{\sqrt{k}\alpha^4} \le C_r \frac{\zeta}{2} \tag{6.66}$$

Plugging (6.64) and (6.66) into (6.65) completes the proof. ∎

## 6.5 Proofs for neural network with random initialization (proof of Theorem 2.2)

In this section we prove Theorem 2.2. We first discuss and prove an optimization result in Section 6.5.1. Next, in Section 6.5.2 we build upon this result to complete the proof of Theorem 2.2.

### 6.5.1 Optimization result

**Theorem 6.23 (Optimization guarantee for random initialization)** *Consider a training data set $\{(\boldsymbol{x}_i, \boldsymbol{y}_i)\}_{i=1}^n \in \mathbb{R}^d \times \mathbb{R}^K$ generated i.i.d. according to a distribution $\mathcal{D}$ where the input samples have unit Euclidean norm and the concatenated label vector obeys $\|\boldsymbol{y}\|_{\ell_2} = \sqrt{n}$ (e.g. one-hot encoding). Consider a neural net with $k$ hidden layers as described in (1.1) parameterized by $\boldsymbol{W}$ where the activation function $\phi$ obeys $|\phi'(z)|, |\phi''(z)| \le B$. Let $\boldsymbol{W}_0$ be the initial weight matrix with i.i.d. $\mathcal{N}(0,1)$ entries. Fix a precision level $\zeta$ and set*

$$\nu = \frac{\zeta}{50B\sqrt{\log(2K)}}. \tag{6.67}$$

*Also assume the output layer $\boldsymbol{V}$ has i.i.d. Rademacher entries scaled by $\frac{\nu}{\sqrt{kK}}$. Furthermore, set $\boldsymbol{J} := \Sigma(\boldsymbol{X})^{1/2}$ and define the information $\mathcal{I}$ and nuisance $\mathcal{N}$ spaces and the truncated Jacobian $\boldsymbol{J}_{\mathcal{I}}$ associated with the reference Jacobian $\boldsymbol{J}$ based on a cut-off spectrum value of $\alpha_0 = \bar{\alpha}\sqrt[4]{n}\sqrt{K\|\boldsymbol{X}\|}B \le B\sqrt{K}\|\boldsymbol{X}\|$ per Definition 1.1 so as to ensure $\|\Pi_{\mathcal{I}}(\boldsymbol{y})\|_{\ell_2} \ge c\|\boldsymbol{y}\|_{\ell_2}$ for some constant c. Assume*

$$k \geq 12 \times 10^7 \frac{\Gamma^4 K^4 B^8 \|\boldsymbol{X}\|^6 n \log(n)}{c^4 \zeta^4 \alpha_0^8} \tag{6.68}$$

with $\Gamma \geq 1$ and $\zeta \leq \frac{c}{2}$. We run gradient descent iterations of the form (1.5) with a learning rate $\eta \leq \frac{1}{\nu^2 B^2 \|\boldsymbol{X}\|^2}$. Then, after $T = \frac{\Gamma K}{\eta \nu^2 \alpha_0^2}$ iterations, the following identities

$$\|f(\boldsymbol{W}_{\tau_0}) - \boldsymbol{y}\|_{\ell_2} \leq \|\Pi_{\mathcal{N}}(\boldsymbol{y})\|_{\ell_2} + e^{-\Gamma} \|\Pi_{\mathcal{I}}(\boldsymbol{y})\|_{\ell_2} + 4\zeta\sqrt{n}, \tag{6.69}$$

$$\|\boldsymbol{W}_{\tau} - \boldsymbol{W}_0\|_{\ell_2} \leq \frac{2\sqrt{K}(\mathcal{B}_{\alpha_0,\Gamma}(\boldsymbol{y}) + \Gamma\zeta\sqrt{n})}{\nu\alpha_0}, \tag{6.70}$$

$$\|\boldsymbol{W}_{\tau} - \boldsymbol{W}_0\|_{2,\infty} \leq \frac{4\Gamma B K \|\boldsymbol{X}\|}{\nu\alpha_0^2} \sqrt{\frac{n}{k}}, \tag{6.71}$$

hold with probability at least $1 - (2K)^{-100}$.

**Proof** To prove this result we wish to apply Theorem 6.20. To do this we need to verify the assumptions of this theorem. To start with, using Lemma 6.14 with probability at least $1 - (2K)^{-100}$, the initial prediction vector $f(\boldsymbol{W}_0)$ obeys

$$\|f(\boldsymbol{W}_0)\|_{\ell_2} \leq \zeta \|\boldsymbol{y}\|_{\ell_2} = \zeta\sqrt{n} \leq \frac{\sqrt{n}}{2}. \tag{6.72}$$

Hence the initial residual obeys $\|\boldsymbol{r}_0\|_{\ell_2} \leq 2\sqrt{n}$. Furthermore, using $\zeta \leq c/2$

$$\|\boldsymbol{r}_0 + \boldsymbol{y}\|_{\ell_2} \leq \zeta \|\boldsymbol{y}\|_{\ell_2} \implies \|\Pi_{\mathcal{I}}(\boldsymbol{r}_0 + \boldsymbol{y})\|_{\ell_2} \leq \zeta \|\boldsymbol{y}\|_{\ell_2}. \tag{6.73}$$

Thus,

$$\begin{aligned}
\|\Pi_{\mathcal{I}}(\boldsymbol{r}_0)\|_{\ell_2} &\geq \|\Pi_{\mathcal{I}}(\boldsymbol{y})\|_{\ell_2} - \|\Pi_{\mathcal{I}}(\boldsymbol{r}_0 + \boldsymbol{y})\|_{\ell_2} \\
&\geq \|\Pi_{\mathcal{I}}(\boldsymbol{y})\|_{\ell_2} - \zeta \|\boldsymbol{y}\|_{\ell_2} \\
&\geq (c - \zeta) \|\boldsymbol{y}\|_{\ell_2} \\
&\geq \frac{c}{2} \|\boldsymbol{y}\|_{\ell_2} \tag{6.74} \\
&\geq \frac{c}{4} \|\boldsymbol{r}_0\|_{\ell_2}. \tag{6.75}
\end{aligned}$$

Thus the assumption on the ratio of information to total energy of residual holds and we can replace $c$ with $\frac{c}{4}$ in Theorem 6.20. Furthermore, since $\mathcal{B}_{\alpha_0,\Gamma}(\cdot)$ is $\Gamma$-Lipschitz function of its input vector in $\ell_2$ norm hence we also have

$$\mathcal{B}_{\alpha_0,\Gamma}(\boldsymbol{r}_0) \leq \mathcal{B}_{\alpha_0,\Gamma}(\boldsymbol{y}) + \Gamma\|\boldsymbol{r}_0 + \boldsymbol{y}\|_{\ell_2} \leq \mathcal{B}_{\alpha_0,\Gamma}(\boldsymbol{y}) + \Gamma\zeta\|\boldsymbol{y}\|_{\ell_2}. \tag{6.76}$$

Next we wish to show that (6.51) holds. In particular we will show that there exists an $\varepsilon_0$-reference Jacobian $\boldsymbol{J}$ for $\mathcal{J}(\boldsymbol{W}_0)$ satisfying $\boldsymbol{J}\boldsymbol{J}^T = \mathbb{E}[\mathcal{J}(\boldsymbol{W}_0)\mathcal{J}(\boldsymbol{W}_0)^T]$. Note that, such a $\boldsymbol{J}$ will have exactly same information/nuisance spaces as the square-root of the multiclass kernel matrix i.e. $(\mathbb{E}[\mathcal{J}(\boldsymbol{W}_0)\mathcal{J}(\boldsymbol{W}_0)^T])^{\frac{1}{2}}$ since these subspaces are governed by the left eigenvectors. Applying Lemmas 6.13 (with a scaling of the Jacobian by $1/\sqrt{kK}$ due to the different scaling of $\boldsymbol{V}$), we find that if

$$k \geq \frac{1000 K^2 B^4 \|\boldsymbol{X}\|^4 \log(n)}{\delta^2} \tag{6.77}$$

then,

$$\left\| \mathcal{J}(\boldsymbol{W}_0)\mathcal{J}(\boldsymbol{W}_0)^T - \mathbb{E}[\mathcal{J}(\boldsymbol{W}_0)\mathcal{J}(\boldsymbol{W}_0)^T] \right\| \leq \frac{\delta\nu^2}{K}. \tag{6.78}$$

Let $\overline{\mathcal{J}}(\boldsymbol{W})$ be obtained by adding $\max(Kn - p, 0)$ zero columns to $\mathcal{J}(\boldsymbol{W})$. Then, using (6.78) and Lemma 6.4, there exists $\boldsymbol{J}$ satisfying $\boldsymbol{J}\boldsymbol{J}^T = \mathbb{E}[\mathcal{J}(\boldsymbol{W}_0)\mathcal{J}(\boldsymbol{W}_0)^T]$ and

$$\left\| \overline{\mathcal{J}}(\boldsymbol{W}_0) - \boldsymbol{J} \right\| \leq 2\sqrt{\frac{\delta\nu^2}{K}}.$$

Therefore, $\boldsymbol{J}$ is an $\varepsilon_0^2 = 4\frac{\delta\nu^2}{K}$ reference Jacobian. Now set

$$\Theta = \min\left(c\frac{\mathcal{B}_{\alpha_0,\Gamma}\alpha_0}{B\Gamma^2\|\boldsymbol{X}\|\sqrt{nK}}, \left(\frac{\zeta B\sqrt{K}\|\boldsymbol{X}\|}{\alpha_0}\right)^2, \frac{\zeta}{\Gamma}\right)$$

and note that using $\alpha = \frac{\nu}{\sqrt{K}}\alpha_0$ and $\|\boldsymbol{r}_0\|_{\ell_2} \le 2\sqrt{n}$

$$\Theta = \min\left(c\frac{\mathcal{B}_{\alpha_0,\Gamma}\alpha_0}{B\Gamma^2\|\boldsymbol{X}\|\sqrt{nK}}, \left(\frac{\zeta B\sqrt{K}\|\boldsymbol{X}\|}{\alpha_0}\right)^2, \frac{\zeta}{\Gamma}\right)$$

$$= \min\left(c\frac{\mathcal{B}_{\alpha_0,\Gamma}\alpha}{\nu B\Gamma^2\|\boldsymbol{X}\|\sqrt{n}}, \left(\frac{\nu\zeta B\|\boldsymbol{X}\|}{\alpha}\right)^2, \frac{\zeta}{\Gamma}\right)$$

$$\le 2\cdot\min\left(c\frac{\mathcal{B}_{\alpha,\Gamma}\alpha}{\nu B\Gamma^2\|\boldsymbol{r}_0\|_{\ell_2}\|\boldsymbol{X}\|}, \frac{\zeta^2\nu^2 B^2\|\boldsymbol{X}\|^2}{\alpha^2}, \frac{\zeta}{\Gamma}\right) \tag{6.79}$$

To continue further, note that $\mathcal{B}_{\alpha_0,\Gamma}$ calculated with respect to $\boldsymbol{\Sigma}(\boldsymbol{X})^{1/2}$ with cutoff $\alpha_0$ is exactly same as $\mathcal{B}_{\alpha,\Gamma}$ calculated with respect to $\boldsymbol{J}$ with cutoff $\alpha = \frac{\nu\alpha_0}{\sqrt{K}}$ which is a square-root of $\mathbb{E}[\mathcal{J}(\boldsymbol{W}_0)\mathcal{J}(\boldsymbol{W}_0)^T]$. Thus, using (6.79) to ensure (6.51) holds it suffices to show

$$\varepsilon_0^2 = 4\frac{\delta\nu^2}{K} \le \frac{\alpha^2}{25}\frac{\Theta}{2} = \frac{\nu^2\alpha_0^2}{50K}\Theta.$$

Hence, to ensure (6.51) holds we need to ensure that $\delta$ obeys

$$\delta \le \frac{\alpha_0^2}{200}\Theta.$$

Thus using $\delta = \frac{\alpha_0^2}{200}\Theta$ to ensure (6.51) we need to make sure $k$ is sufficiently large so that (6.77) holds with this value of $\delta$. Thus it suffices to have

$$k \ge 12\times 10^7 \frac{\Gamma^4 K^4 B^8\|\boldsymbol{X}\|^6 n\log(n)}{c^4\zeta^4\alpha_0^8} \tag{6.80}$$

$$\ge 12\times 10^7 \frac{\Gamma^4 K^4 B^8\|\boldsymbol{X}\|^8\log(n)}{c^4\zeta^4\alpha_0^8}$$

$$\ge 4\times 10^7\cdot\left(\frac{4K^2 B^4\Gamma^4\|\boldsymbol{X}\|^4}{c^4\alpha_0^4} + \frac{1}{\zeta^4} + \frac{\Gamma^2}{\zeta^2}\right)\frac{K^2 B^4\|\boldsymbol{X}\|^4\log(n)}{\alpha_0^4}$$

$$\ge 4\times 10^7\cdot\left(\frac{4KB^4\Gamma^4\|\boldsymbol{X}\|^4}{c^4\alpha_0^4} + \frac{1}{\zeta^4} + \frac{\Gamma^2}{\zeta^2}\right)\frac{K^2 B^4\|\boldsymbol{X}\|^4\log(n)}{\alpha_0^4}$$

$$\ge 4\times 10^7\cdot\left(\frac{4KB^4\Gamma^4\|\boldsymbol{X}\|^4}{c^4\alpha_0^4} + \frac{\alpha_0^4}{\zeta^4 B^4 K^2\|\boldsymbol{X}\|^4} + \frac{\Gamma^2}{\zeta^2}\right)\frac{K^2 B^4\|\boldsymbol{X}\|^4\log(n)}{\alpha_0^4}$$

$$\stackrel{(a)}{\ge} 4\times 10^7\cdot\left(\frac{nKB^2\Gamma^4\|\boldsymbol{X}\|^2}{c^2\mathcal{B}_{\alpha_0,\Gamma}^2\alpha_0^2} + \frac{\alpha_0^4}{\zeta^4 B^4 K^2\|\boldsymbol{X}\|^4} + \frac{\Gamma^2}{\zeta^2}\right)\frac{K^2 B^4\|\boldsymbol{X}\|^4\log(n)}{\alpha_0^4}$$

$$\ge 4\times 10^7\cdot\max\left(\frac{nKB^2\Gamma^4\|\boldsymbol{X}\|^2}{c^2\mathcal{B}_{\alpha_0,\Gamma}^2\alpha_0^2}, \frac{\alpha_0^4}{\zeta^4 B^4 K^2\|\boldsymbol{X}\|^4}, \frac{\Gamma^2}{\zeta^2}\right)\frac{K^2 B^4\|\boldsymbol{X}\|^4\log(n)}{\alpha_0^4}$$

$$= 4\times 10^7\frac{K^2 B^4\|\boldsymbol{X}\|^4\log(n)}{\alpha_0^4\cdot\min\left(\left(\frac{c\mathcal{B}_{\alpha_0,\Gamma}\alpha_0}{B\Gamma^2\|\boldsymbol{X}\|\sqrt{nK}}\right)^2, \left(\frac{\zeta B\sqrt{K}\|\boldsymbol{X}\|}{\alpha_0}\right)^4, \frac{\zeta^2}{\Gamma^2}\right)}$$

$$= \frac{1000 K^2 B^4\|\boldsymbol{X}\|^4\log(n)}{\delta^2} \tag{6.81}$$

Here, (a) follows from the fact that $\|\boldsymbol{\Sigma}(\boldsymbol{X})^{1/2}\| := \lambda_1 \le B\|\boldsymbol{X}\|$, equation (6.3), and $\|\Pi_{\mathcal{I}}(\boldsymbol{r}_0)\|_{\ell_2} \ge \frac{c}{2}\|\boldsymbol{y}\|_{\ell_2} = \frac{c}{2}\sqrt{n}$ which combined imply

$$\mathcal{B}_{\alpha_0,\Gamma} \ge \frac{\alpha}{\lambda_1}\|\Pi_{\mathcal{I}}(\boldsymbol{r}_0)\|_{\ell_2} \ge \frac{\alpha}{B\|\boldsymbol{X}\|}\|\Pi_{\mathcal{I}}(\boldsymbol{r}_0)\|_{\ell_2} \ge \frac{\alpha_0 c}{2}\frac{\|\boldsymbol{y}\|_{\ell_2}}{B\|\boldsymbol{X}\|} = \frac{\alpha_0 c}{2}\frac{\sqrt{n}}{B\|\boldsymbol{X}\|}. \tag{6.82}$$

To be able to apply Theorem 6.20 we must also ensure (6.52) holds. Therefore, it suffices to have

$$k \geq 64 \times 10^6 \frac{K^4 B^8 \|\boldsymbol{X}\|^6 \Gamma^4 n \log(n)}{\zeta^4 \alpha_0^8} \tag{6.83}$$

$$\overset{(a)}{\geq} 25600 \frac{K^4 B^6 \|\boldsymbol{X}\|^6 \Gamma^4 n}{\zeta^2 \nu^2 \alpha_0^8} \tag{6.84}$$

$$k \geq 12800 \left( \frac{\alpha_0^2}{\zeta^2 B^2 K \|\boldsymbol{X}\|^2} + 1 \right) \frac{K^4 B^6 \|\boldsymbol{X}\|^6 \Gamma^4 n}{\nu^2 \alpha_0^8}$$

$$\overset{(b)}{\geq} 3200 \left( \frac{\alpha_0^2}{\zeta^2 B^2 K \|\boldsymbol{X}\|^2} + \frac{4\Gamma^2 n}{\mathcal{B}_{\alpha,\Gamma}^2} \right) \frac{K^4 B^6 \|\boldsymbol{X}\|^6 \Gamma^2 \mathcal{B}_{\alpha,\Gamma}^2}{\nu^2 \alpha_0^8}$$

$$\geq 3200 \left( \frac{\alpha_0^2}{\zeta^2 B^2 K \|\boldsymbol{X}\|^2} + \frac{\Gamma^2 \|\boldsymbol{r}_0\|_{\ell_2}^2}{\mathcal{B}_{\alpha,\Gamma}^2} \right) \frac{K^4 B^6 \|\boldsymbol{X}\|^6 \Gamma^2 \mathcal{B}_{\alpha,\Gamma}^2}{\nu^2 \alpha_0^8}$$

$$\geq 1600 \left( \frac{\alpha_0}{\zeta \sqrt{K} B \|\boldsymbol{X}\|} + \frac{\Gamma \|\boldsymbol{r}_0\|_{\ell_2}}{\mathcal{B}_{\alpha,\Gamma}} \right)^2 \frac{K^4 B^6 \|\boldsymbol{X}\|^6 \Gamma^2 \mathcal{B}_{\alpha,\Gamma}^2}{\nu^2 \alpha_0^8}$$

$$= 1600 \left( \frac{\alpha}{\zeta \nu B \|\boldsymbol{X}\|} + \frac{\Gamma \|\boldsymbol{r}_0\|_{\ell_2}}{\mathcal{B}_{\alpha,\Gamma}} \right)^2 \frac{\nu^6 B^6 \|\boldsymbol{X}\|^6 \Gamma^2 \mathcal{B}_{\alpha,\Gamma}^2}{\alpha^8} \tag{6.85}$$

Here, (a) follows from the fact that $n \geq K$ and the relationship between $\zeta$ and $\nu$ per (6.88) and (b) follows from the fact that per equation (6.2) we have

$$\mathcal{B}_{\alpha,\Gamma} \leq \Gamma \|\boldsymbol{r}_0\|_{\ell_2} \leq 2\Gamma \sqrt{n}$$

Note that (6.80) and (6.84) are implied by

$$k \geq 12 \times 10^7 \frac{\Gamma^4 K^4 B^8 \|\boldsymbol{X}\|^6 n \log(n)}{c^4 \zeta^4 \alpha_0^8}, \tag{6.86}$$

which is the same as (6.68). What remains is stating the optimization bounds in terms of the labels $\boldsymbol{y}$. This follows by substituting (6.72), (6.76), and the fact that $\|\boldsymbol{r}_0\|_{\ell_2} \leq 2\sqrt{n}$ into (6.55), (6.53), and (6.54), respectively. ∎

### 6.5.2 Generalization result (completing the proof of Theorem 2.2)

The theorem below is the formal statement and a more general version of Theorem 2.2. The differences (besides constant terms) are the use of $\|\boldsymbol{X}\|$ rather than $\sqrt{n}$ (we always have $\|\boldsymbol{X}\| \leq \sqrt{n}$) and use of the tighter bound $\left\| \boldsymbol{J}_{\mathcal{I}}^\dagger \boldsymbol{y} \right\|_{\ell_2}$ rather than $\|\Pi_{\mathcal{I}}(\boldsymbol{y})\|_{\ell_2} / \alpha_0$.

**Theorem 6.24** *Let* $\zeta, \Gamma, \bar{\alpha}$ *be scalars obeying* $\zeta \leq 1/2$, $\Gamma \geq 1$, *and* $\bar{\alpha} \geq 0$ *which determine the overall precision, cut-off and learning duration, respectively.*[5] *Consider a training data set* $\{(\boldsymbol{x}_i, \boldsymbol{y}_i)\}_{i=1}^n \in \mathbb{R}^d \times \mathbb{R}^K$ *generated i.i.d. according to a distribution* $\mathcal{D}$ *where the input samples have unit Euclidean norm and the concatenated label vector obeys* $\|\boldsymbol{y}\|_{\ell_2} = \sqrt{n}$ *(e.g. one-hot encoding). Consider a neural net with* $k$ *hidden nodes as described in* (1.1) *parameterized by* $\boldsymbol{W}$ *where the activation function* $\phi$ *obeys* $|\phi'(z)|, |\phi''(z)| \leq B$. *Let* $\boldsymbol{W}_0$ *be the initial weight matrix with i.i.d.* $\mathcal{N}(0,1)$ *entries. Fix a precision level* $\zeta$ *and set* $\nu = \zeta / (50 B \sqrt{\log(2K)})$. *Also assume the output layer* $\boldsymbol{V}$ *has i.i.d. Rademacher entries scaled by* $\frac{\nu}{\sqrt{kK}}$. *Furthermore, set* $\boldsymbol{J} := (\boldsymbol{\Sigma}(\boldsymbol{X}))^{1/2}$ *and define the information* $\mathcal{I}$ *and nuisance* $\mathcal{N}$ *spaces and the truncated Jacobian* $\boldsymbol{J}_{\mathcal{I}}$ *associated with the Jacobian* $\boldsymbol{J}$ *based on a cut-off spectrum value of* $\alpha_0 = \bar{\alpha} \sqrt[4]{n} \sqrt{K} \|\boldsymbol{X}\| B$ *per Definition 1.1. Assume*

$$k \gtrsim \frac{\Gamma^4 \log n}{\zeta^4 \bar{\alpha}^8} \tag{6.87}$$

---

[5]Note that this theorem and its conclusions hold for any choice of these parameters in the specified range.

with $\Gamma \geq 1$. We run gradient descent iterations of the form (1.5) with a learning rate $\eta \leq \frac{1}{\nu^2 B^2 \|\boldsymbol{X}\|^2}$. Then, after $T = \frac{\Gamma K}{\eta \nu^2 \alpha_0^2}$ iterations, training loss obeys (6.69) and classification error $Err_{\mathcal{D}}(\boldsymbol{W}_T)$ is upper bounded by

$$\underbrace{\frac{2\|\Pi_{\mathcal{N}}(\boldsymbol{y})\|_{\ell_2}}{\sqrt{n}}}_{\text{bias term}} + \underbrace{\frac{12B\sqrt{K}}{\sqrt{n}}\left(\|\boldsymbol{J}_{\mathcal{I}}^{\dagger}\boldsymbol{y}\|_{\ell_2} + \frac{\Gamma}{\alpha_0}\|\Pi_{\mathcal{N}}(\boldsymbol{y})\|_{\ell_2}\right)}_{\text{variance term}} + 12\left(1 + \frac{\Gamma}{\bar{\alpha}\sqrt[4]{n}\|\boldsymbol{X}\|^2}\right)\zeta + 5\sqrt{\frac{\log(2/\delta)}{n}} + 2e^{-\Gamma},$$

holds with probability at least $1 - (2K)^{-100} - \delta$.

The theorem below is a restatement of Theorem 6.24 after substituting the upper bound on the early stopping distance $\mathcal{D}_{\alpha_0, \Gamma}$ of Def. (6.1).

**Theorem 6.25 (Neural Net – Generalization)** *Consider a training data set $\{(\boldsymbol{x}_i, \boldsymbol{y}_i)\}_{i=1}^n \in \mathbb{R}^d \times \mathbb{R}^K$ generated i.i.d. according to a distribution $\mathcal{D}$ where the input samples have unit Euclidean norm and the concatenated label vector obeys $\|\boldsymbol{y}\|_{\ell_2} = \sqrt{n}$ (e.g. one-hot encoding). Consider a neural net with $k$ hidden nodes as described in (1.1) parameterized by $\boldsymbol{W}$ where the activation function $\phi$ obeys $|\phi'(z)|, |\phi''(z)| \leq B$. Let $\boldsymbol{W}_0$ be the initial weight matrix with i.i.d. $\mathcal{N}(0,1)$ entries. Fix a precision level $\zeta \leq \frac{c}{2}$ and set*

$$\nu = \frac{\zeta}{50B\sqrt{\log(2K)}}. \tag{6.88}$$

*Also assume the output layer $\boldsymbol{V}$ has i.i.d. Rademacher entries scaled by $\frac{\nu}{\sqrt{kK}}$. Furthermore, set $\boldsymbol{J} := \boldsymbol{\Sigma}(\boldsymbol{X})^{1/2}$ and define the information $\mathcal{I}$ and nuisance $\mathcal{N}$ spaces and the truncated Jacobian $\boldsymbol{J}_{\mathcal{I}}$ associated with the Jacobian $\boldsymbol{J}$ based on a cut-off spectrum value of $\alpha_0 = \bar{\alpha}\sqrt[4]{n}\sqrt{K}\|\boldsymbol{X}\|B \leq B\|\boldsymbol{X}\|$ per Definition 1.1 chosen to ensure $\|\Pi_{\mathcal{I}}(\boldsymbol{y})\|_{\ell_2} \geq c\|\boldsymbol{y}\|_{\ell_2}$ for some constant $c > 0$. Assume*

$$k \geq 12 \times 10^7 \frac{\Gamma^4 K^4 B^8 \|\boldsymbol{X}\|^4 n^2 \log(n)}{c^4 \zeta^4 \alpha_0^8} \tag{6.89}$$

*with $\Gamma \geq 1$. We run gradient descent iterations of the form (1.5) with a learning rate $\eta \leq \frac{1}{\nu^2 B^2 \|\boldsymbol{X}\|^2}$. Then, after $T = \frac{\Gamma K}{\eta \nu^2 \alpha_0^2}$ iterations, classification error $Err_{\mathcal{D}}(\boldsymbol{W}_T)$ is upper bounded by*

$$Err_{\mathcal{D}}(\boldsymbol{W}_T) \leq 2\frac{\|\Pi_{\mathcal{N}}(\boldsymbol{y})\|_{\ell_2} + e^{-\Gamma}\|\Pi_{\mathcal{I}}(\boldsymbol{y})\|_{\ell_2}}{\sqrt{n}} + \frac{12B\sqrt{K}}{\sqrt{n}}\mathcal{D}_{\alpha_0, \Gamma} + 12\left(1 + \frac{\Gamma}{\bar{\alpha}\sqrt[4]{n}\|\boldsymbol{X}\|^2}\right)\zeta + 10\sqrt{\frac{\log(2/\delta)}{n}},$$

*holds with probability at least $1 - (2K)^{-100} - \delta$.*

**Proof** Under the stated assumptions, Theorem 6.23 holds with probability $1 - (2K)^{-100}$. The proof will condition on outcomes of the Theorem 6.23. Specifically, we shall apply (6.43) of Lemma 6.18 with $M_{\mathcal{W}}$ and $R$ dictated by Theorem 6.23 where the output layer $\boldsymbol{V}$ is fixed. Observe that $\|\boldsymbol{V}\|_F = \sqrt{Kk}\|\boldsymbol{V}\|_{\ell_\infty} = \nu$, we have

$$Err_{\mathcal{D}}(\boldsymbol{W}_T) \leq 2\left[\frac{\|f(\boldsymbol{W}_T) - \boldsymbol{y}\|_{\ell_2}}{\sqrt{n}} + 3\nu B\left(\frac{M_{\mathcal{W}}}{\sqrt{n}} + \frac{R^2}{\sqrt{k}}\right) + \sqrt{\frac{5\log(2/\delta)}{n}}\right]. \tag{6.90}$$

Theorem 6.23 yields

$$\frac{\|f(\boldsymbol{W}_T) - \boldsymbol{y}\|_{\ell_2}}{\sqrt{n}} \leq \frac{\|\Pi_{\mathcal{N}}(\boldsymbol{y})\|_{\ell_2} + e^{-\Gamma}\|\Pi_{\mathcal{I}}(\boldsymbol{y})\|_{\ell_2}}{\sqrt{n}} + 4\zeta. \tag{6.91}$$

Using (6.70) for $M_{\mathcal{W}}$

$$\frac{\nu B M_{\mathcal{W}}}{\sqrt{n}} \leq \frac{2B\sqrt{K}\mathcal{D}_{\alpha_0, \Gamma}(\boldsymbol{y})}{\sqrt{n}} + \frac{2B\sqrt{K}\Gamma\zeta}{\alpha_0}. \tag{6.92}$$

Using (6.71) on row bound $R$ and lower bound on $k$

$$3\nu B \frac{R^2}{\sqrt{k}} = \frac{48n\Gamma^2 B^3 K^2 \|X\|^2}{\nu \alpha_0^4 \sqrt{k}}$$

$$\leq \frac{c^2 \zeta^2}{230\nu B \log(n)}$$

$$\leq \zeta. \tag{6.93}$$

Plugging in (6.91), (6.92), and (6.93) into (6.90) concludes the proof. ∎

## A  THE JACOBIAN OF THE MIXTURE MODEL IS LOW-RANK (PROOFS FOR SECTION 2.3)

The following theorem considers a simple noiseless mixture model and proves that its Jacobian is low-rank and the concatenated multiclass label vectors lie on a rank $K^2C$ information space associated with this Jacobian.

**Theorem A.1** *Consider a data set of size $n$ consisting of input/label pairs $\{(x_i, y_i)\}_{i=1}^n \in \mathbb{R}^d \times \mathbb{R}^K$ generated according to the Gaussian mixture model of Definition 2.4 with $K$ classes each consisting of $C$ clusters with the cluster centers given by $\{\mu_{\ell,\widetilde{\ell}}\}_{(\ell,\widetilde{\ell})=(1,1)}^{(K,C)}$ and $\sigma = 0$. Let $\Sigma(X)$ be the multiclass neural tangent kernel matrix associated with input matrix $X = [x_1 \ \ldots \ x_n]^T$ with the standard deviation of the output layer set to $\nu = \frac{1}{\sqrt{k}}$. Also define the information space $\mathcal{I}$ to be the range space of $\Sigma(X)$. Also let $M = [\mu_{1,1} \ \ldots \ \mu_{K,C}]^T$ be the matrix obtained by aggregating all the cluster centers as rows and let $g$ be a Gaussian random vector with distribution $\mathcal{N}(0, I_d)$. Define the neural tangent kernel matrix associated with the cluster centers as*

$$\widetilde{\Sigma}(M) = (MM^T) \odot \mathbb{E}_{g\sim\mathcal{N}(0,I_d)}[\phi'(Mg)\phi'(Mg)^T] \in \mathbb{R}^{KC \times KC},$$

*and assume that $\widetilde{\Sigma}(M)$ is full rank. Then, the following properties hold with probability $1 - KC \exp(-\frac{n}{8KC})$*

- $\mathcal{I}$ *is a $K^2C$ dimensional subspace.*

- *The concatenated label vector $y = \begin{bmatrix} y_1^T & y_2^T & \ldots & y_n^T \end{bmatrix}^T$ lies on $\mathcal{I}$.*

- *The nonzero eigenvalues (top $K^2C$ eigenvalues) of $\Sigma(X)$ are between $\frac{n}{2KC}s_{\min}(\Sigma(X))$ and $\frac{2n}{KC}\|\Sigma(X)\|$. Hence the eigenvalues of the information space grow with $\frac{n}{KC}$.*

**Proof**  First, we establish that each cluster has around the same size. Applying Chernoff bound and a union bound, we find that with probability $1 - KC \exp(-\frac{n}{8KC})$

$$0.5\tilde{n} \leq \widetilde{n}_{\ell,\tilde{\ell}} \leq 2\tilde{n}.$$

Note that based on Lemma 6.11, the multiclass covariance is given by

$$\Sigma(X) = k\nu^2 I_K \otimes \widetilde{\Sigma}(X).$$

where $\widetilde{\Sigma}(X) = (XX^T) \odot \mathbb{E}_{g\overset{\text{i.i.d.}}{\sim}\mathcal{N}(0,1)}[\phi'(Xg)\phi'(Xg)^T]$. Due to this Kronecker product representation, the range space of $\Sigma(X)$ is separable. In particular, note that with

$$\widetilde{\mathcal{I}} = \text{Range}\left(\widetilde{\Sigma}(X)\right)$$

we have $\mathcal{I} = I_K \otimes \widetilde{\mathcal{I}}$ which also implies $\text{rank}(\mathcal{I}) = K \cdot \text{rank}\left(\widetilde{\mathcal{I}}\right)$. Hence, this identity allows us to reduce the problem to a single output network. To complete the proof we will prove the following three identities:

- $\widetilde{\mathcal{I}}$ has rank $KC$.

- The nonzero eigenvalues of $\widetilde{\Sigma}(\boldsymbol{X})$ are between $0.5\widetilde{n}s_{\min}(\widetilde{\Sigma}(\boldsymbol{M}))$ to $2\widetilde{n}\|\widetilde{\Sigma}(\boldsymbol{M})\|$.
- The portion of the label vector associated with class $\ell$ i.e. $\boldsymbol{y}^{(\ell)} \in \mathbb{R}^n$ (see (5.4)) lies on $\mathcal{I}$. Hence, the concatenated vector $\boldsymbol{y}$ lies on $\mathcal{I} = \boldsymbol{I}_K \otimes \widetilde{\mathcal{I}}$.

To prove these statements let $\mathcal{J}_\ell(\boldsymbol{X};\boldsymbol{W}_0)$ and $\mathcal{J}_\ell(\boldsymbol{M};\boldsymbol{W}_0)$ be the Jacobian associated with the $\ell$th output of the neural net (see (5.4)) for data matrices $\boldsymbol{X}$ and $\boldsymbol{M}$. Observe that the columns of $\mathcal{J}_\ell(\boldsymbol{X};\boldsymbol{W}_0)$ are chosen from $\mathcal{J}_\ell(\boldsymbol{M};\boldsymbol{W}_0)$ and in particular each column of $\mathcal{J}_\ell(\boldsymbol{M};\boldsymbol{W}_0)$ is repeated between $0.5\widetilde{n}$ to $2\widetilde{n}$ times. To mathematically relate this, define the $KC$ dimensional subspace $\mathcal{S}$ of $\mathbb{R}^n$ where for any $\boldsymbol{v} \in \mathcal{S}$, entries $v_i$ and $v_j$ of $\boldsymbol{v}$ are equal iff data point $\boldsymbol{x}_i$ and $\boldsymbol{x}_j$ are equal (i.e. belong to the same class/cluster pair). Now, we define the orthonormal matrix $\boldsymbol{U}_\mathcal{S} \in \mathbb{R}^{n \times KC}$ as the 0-1 matrix with orthogonal rows that map $\mathbb{R}^{KC}$ to $\mathcal{S}$ as follows. Assume the $i$th data point $\boldsymbol{x}_i$ belongs to the class/cluster pair $(\ell_i, \widetilde{\ell}_i)$. We then set the $i$th row of $\boldsymbol{U}_\mathcal{S}$ as $\mathrm{vect}\left(\boldsymbol{e}_{\ell_i}\boldsymbol{e}_{\widetilde{\ell}_i}^T\right)$. Using $\boldsymbol{U}_\mathcal{S}$ we have

$$\boldsymbol{U}_\mathcal{S}\mathcal{J}_\ell(\boldsymbol{M};\boldsymbol{W}_0) = \mathcal{J}_\ell(\boldsymbol{X};\boldsymbol{W}_0).$$

Now note that using the above identity we have

$$\boldsymbol{U}_\mathcal{S}\widetilde{\Sigma}(\boldsymbol{M})\boldsymbol{U}_\mathcal{S}^T = \widetilde{\Sigma}(\boldsymbol{X}).$$

Since $\boldsymbol{U}_\mathcal{S}$ is tall and orthogonal, the range of $\widetilde{\Sigma}(\boldsymbol{X})$ is exactly the range of $\boldsymbol{U}_\mathcal{S}$ hence $\widetilde{\mathcal{I}} = \mathcal{S}$ which is $KC$ dimensional. Furthermore, nonzero eigenvectors of $\widetilde{\Sigma}(\boldsymbol{X})$ lie on $\mathcal{S}$ and any eigenvector $\boldsymbol{v}$ satisfies

$$\boldsymbol{v}^T\widetilde{\Sigma}(\boldsymbol{X})\boldsymbol{v} \geq s_{\min}(\boldsymbol{U}_\mathcal{S})^2 s_{\min}(\widetilde{\Sigma}(\boldsymbol{M})) \geq 0.5\bar{n}s_{\min}(\widetilde{\Sigma}(\boldsymbol{M}))$$

and similarly

$$\boldsymbol{v}^T\widetilde{\Sigma}(\boldsymbol{X})\boldsymbol{v} \leq 2\bar{n}\|\widetilde{\Sigma}(\boldsymbol{M})\|$$

which follows from the fact that $\ell_2$-norm-squared of columns of $\boldsymbol{U}$ are between $0.5\bar{n}$ to $2\bar{n}$. Finally, we will argue that label vector $\boldsymbol{y}^{(\ell)}$ lies on $\mathcal{S}$. Note that for all samples $i$ that belong to the same cluster $\boldsymbol{y}_i^{(\ell)}$ will be the same (either zero or one), thus $\boldsymbol{y}^{(\ell)} \in \mathcal{S}$. ∎

Next lemma provides a perturbation analysis when there is noise.

**Lemma A.2** *Consider the single-output NTK kernel given by*

$$\widetilde{\Sigma}(\boldsymbol{X}) = \mathbb{E}\left[\phi'(\boldsymbol{X}\boldsymbol{w})\phi'(\boldsymbol{X}\boldsymbol{w})^T\right] \odot \left(\boldsymbol{X}\boldsymbol{X}^T\right),$$

*and assume that this matrix has rank $r$ so that $\lambda_{r+1}\left(\widetilde{\Sigma}(\boldsymbol{X})\right) = \lambda_{r+2}\left(\widetilde{\Sigma}(\boldsymbol{X})\right) = \ldots = \lambda_n\left(\widetilde{\Sigma}(\boldsymbol{X})\right) = 0$. Also assume a noise corrupted version of $\boldsymbol{X}$ given by*

$$\widetilde{\boldsymbol{X}} = \boldsymbol{X} + \frac{\sigma}{\sqrt{d}}\boldsymbol{Z}$$

*with $\boldsymbol{Z}$ a matrix consisting of i.i.d. $\mathcal{N}(0,1)$ entries. Then, $\left\|\widetilde{\Sigma}(\widetilde{\boldsymbol{X}}) - \widetilde{\Sigma}(\boldsymbol{X})\right\| \lesssim \Delta$ where*

$$\Delta := \sigma^2 B^2 \log n \|\boldsymbol{X}\|^2 + \sigma^2 B^2(n/d+1) + \sqrt{\log n} \cdot \sigma B^2 \|\boldsymbol{X}\|^2 + \sigma B^2\sqrt{n/d+1}\|\boldsymbol{X}\| \quad \text{(A.1)}$$

*holds with probability at least $1 - 2n e^{-\frac{d}{2}}$. Whenever $\sigma \leq \frac{1}{\sqrt{\log n}}$, $\Delta$ is upper bounded as*

$$\frac{\Delta}{n} \lesssim B^2\sigma\sqrt{\log n}. \quad \text{(A.2)}$$

*Furthermore, let $\widetilde{\boldsymbol{V}}, \boldsymbol{V} \in \mathbb{R}^{n \times r}$ be orthonormal matrices corresponding to the top $r$ eigenvalues of $\widetilde{\Sigma}(\widetilde{\boldsymbol{X}})$ and $\widetilde{\Sigma}(\boldsymbol{X})$. Then,*

$$\left\|\widetilde{\boldsymbol{V}}\widetilde{\boldsymbol{V}}^T - \boldsymbol{V}\boldsymbol{V}^T\right\| \leq \frac{\Delta}{\lambda_r(\widetilde{\Sigma}(\boldsymbol{X})) - \Delta}$$

**Proof** Note that

$$\mathrm{diag}\left(\phi'\left(\widetilde{\boldsymbol{X}}\boldsymbol{w}\right)\right)\widetilde{\boldsymbol{X}} - \mathrm{diag}\left(\phi'(\boldsymbol{X}\boldsymbol{w})\right)\boldsymbol{X} = \mathrm{diag}\left(\phi'\left(\widetilde{\boldsymbol{X}}\boldsymbol{w}\right)\right)\widetilde{\boldsymbol{X}} - \mathrm{diag}\left(\phi'(\boldsymbol{X}\boldsymbol{w})\right)\boldsymbol{X}$$
$$= \mathrm{diag}\left(\phi'\left(\widetilde{\boldsymbol{X}}\boldsymbol{w}\right) - \phi'(\boldsymbol{X}\boldsymbol{w})\right)\boldsymbol{X}$$
$$+ \mathrm{diag}\left(\phi'\left(\widetilde{\boldsymbol{X}}\boldsymbol{w}\right)\right)\left(\widetilde{\boldsymbol{X}} - \boldsymbol{X}\right)$$

Now define $\widetilde{\boldsymbol{M}} = \mathrm{diag}\left(\phi'(\widetilde{\boldsymbol{X}}\boldsymbol{w})\right)\widetilde{\boldsymbol{X}}$ and $\boldsymbol{M} = \mathrm{diag}\left(\phi'(\boldsymbol{X}\boldsymbol{w})\right)\boldsymbol{X}$ and note that using the above we can conclude that

$$
\begin{aligned}
\left\|\widetilde{\boldsymbol{M}} - \boldsymbol{M}\right\| &\leq \left\|\mathrm{diag}\left(\phi'\left(\widetilde{\boldsymbol{X}}\boldsymbol{w}\right) - \phi'\left(\boldsymbol{X}\boldsymbol{w}\right)\right)\boldsymbol{X}\right\| \\
&\quad + \left\|\mathrm{diag}\left(\phi'\left(\widetilde{\boldsymbol{X}}\boldsymbol{w}\right)\right)\left(\widetilde{\boldsymbol{X}} - \boldsymbol{X}\right)\right\| \\
&\leq B\left\|(\widetilde{\boldsymbol{X}} - \boldsymbol{X})\boldsymbol{w}\right\|_{\ell_\infty}\|\boldsymbol{X}\| + B\left\|\widetilde{\boldsymbol{X}} - \boldsymbol{X}\right\|
\end{aligned}
$$

Now using the fact that

$$
\left\|\widetilde{\boldsymbol{M}}\widetilde{\boldsymbol{M}}^T - \boldsymbol{M}\boldsymbol{M}^T\right\| \leq \left\|\widetilde{\boldsymbol{M}} - \boldsymbol{M}\right\|^2 + 2\left\|\widetilde{\boldsymbol{M}} - \boldsymbol{M}\right\|\|\boldsymbol{M}\|,
$$

we conclude that

$$
\begin{aligned}
\left\|\widetilde{\boldsymbol{\Sigma}}(\widetilde{\boldsymbol{X}}) - \widetilde{\boldsymbol{\Sigma}}(\boldsymbol{X})\right\| &= \left\|\mathbb{E}\left[\widetilde{\boldsymbol{M}}\widetilde{\boldsymbol{M}}^T - \boldsymbol{M}\boldsymbol{M}^T\right]\right\| \\
&\leq \mathbb{E}\left[\left(B\left\|(\widetilde{\boldsymbol{X}} - \boldsymbol{X})\boldsymbol{w}\right\|_{\ell_\infty}\|\boldsymbol{X}\| + B\left\|\widetilde{\boldsymbol{X}} - \boldsymbol{X}\right\|\right)^2\right] \\
&\quad + 2B\|\boldsymbol{X}\|\left(B\|\boldsymbol{X}\|\mathbb{E}\left[\left\|(\widetilde{\boldsymbol{X}} - \boldsymbol{X})\boldsymbol{w}\right\|_{\ell_\infty}\right] + B\left\|\widetilde{\boldsymbol{X}} - \boldsymbol{X}\right\|\right) \\
&\leq 2B^2\|\boldsymbol{X}\|^2\mathbb{E}\left[\left\|(\widetilde{\boldsymbol{X}} - \boldsymbol{X})\boldsymbol{w}\right\|_{\ell_\infty}^2\right] + 2B^2\left\|\widetilde{\boldsymbol{X}} - \boldsymbol{X}\right\|^2 \\
&\quad + 2B^2\|\boldsymbol{X}\|^2\mathbb{E}\left[\left\|(\widetilde{\boldsymbol{X}} - \boldsymbol{X})\boldsymbol{w}\right\|_{\ell_\infty}\right] + 2B^2\left\|\widetilde{\boldsymbol{X}} - \boldsymbol{X}\right\|\|\boldsymbol{X}\|
\end{aligned}
$$

To proceed further, with probability $1 - n\exp(-d/2)$, each row of $\widetilde{\boldsymbol{X}} - \boldsymbol{X}$ is upper bounded by $2\sigma$. Hence, using a standard tail bound over supremum of $n$ Gaussian random variables (which follows by union bounding) we have

$$
\mathbb{E}[\left\|(\widetilde{\boldsymbol{X}} - \boldsymbol{X})\boldsymbol{w}\right\|_{\ell_\infty}^2]^{1/2} \leq 2\sigma\sqrt{2\log n}
$$

holds with the same probability. Furthermore, spectral norm bound on Gaussian random matrix implies that

$$
\left\|\widetilde{\boldsymbol{X}} - \boldsymbol{X}\right\|^2 \leq \left(2(\sqrt{n} + \sqrt{d})\right)^2\frac{\sigma^2}{d} \leq 8(n/d + 1)\sigma^2,
$$

holds with probability at least $1 - \mathrm{e}^{-\frac{1}{2}(n+d)}$. Plugging these two probabilistic bounds into the chain of inequalities we conclude that

$$
\left\|\widetilde{\boldsymbol{\Sigma}}(\widetilde{\boldsymbol{X}}) - \widetilde{\boldsymbol{\Sigma}}(\boldsymbol{X})\right\| \lesssim \sigma^2 B^2\log n\|\boldsymbol{X}\|^2 + \sigma^2 B^2(n/d + 1) + \sqrt{\log n}\cdot\sigma B^2\|\boldsymbol{X}\|^2 + \sigma B^2\sqrt{n/d + 1}\|\boldsymbol{X}\|
$$

To establish (A.2), observe that $B^2\sigma\sqrt{\log n}\|\boldsymbol{X}\|^2$ dominates over other terms in the regime $\sigma\sqrt{\log n}$ is small. The final bound is a standard application of Davis-Kahan Theorem Yu et al. (2014) when we use the fact that $\tilde{\boldsymbol{\Sigma}}(\boldsymbol{X})$ is low-rank. ∎

The following lemma plugs in the critical quantities of Theorem 2.2 for our mixture model to obtain a generalization bound.

**Theorem A.3 (Generalization for Mixture Model)** *Consider a dataset $\{\boldsymbol{x}_i, \boldsymbol{y}_i\}_{i=1}^n$ generated i.i.d. from the Gaussian mixture model in Definition 2.4. Let $\lambda_{\boldsymbol{M}} = \lambda_{\min}(\boldsymbol{\Sigma}(\boldsymbol{M}))$ where $\boldsymbol{M} \in \mathbb{R}^{KC\times d}$ is the matrix of cluster centers. Suppose input noise level $\sigma$ obeys*

$$
\sigma \lesssim \frac{\lambda_{\min}}{B^2 KC\sqrt{\log n}}
$$

*Consider the setup of Theorem 2.2 with quantities $\zeta$ and $\Gamma$. Suppose network width obeys*

$$
k \gtrsim \frac{\Gamma^4 B^8 K^8 C^4\log n}{\zeta^4\lambda_{\min}^4}.
$$

*With probability $1 - n\mathrm{e}^{-d/2} - KC\exp(-\frac{n}{8KC}) - (2K)^{-100} - \delta$, running gradient descent for $T = \frac{2\Gamma K^2 C}{\eta\nu^2 n\lambda_{\min}}$ with learning rate $\eta \leq \frac{1}{\nu^2 B^2\|\boldsymbol{X}\|^2}$, we have that*

$$
Err_{\mathcal{D}}(\boldsymbol{W}_T) \lesssim \sqrt{\frac{\sigma\sqrt{\log n}B^2 KC}{\lambda_{\min}}} + \frac{\Gamma BK\sqrt{C}}{\sqrt{n}\lambda_{\min}} + 12\zeta + 5\sqrt{\frac{\log(2/\delta)}{n}} + 2\mathrm{e}^{-\Gamma}.
$$

**Proof** The proof is an application of Lemma A.2 and Theorem A.1. Let $\mathcal{I}'$ be the information space corresponding to noiseless dataset where input samples are identical to cluster centers. Let $\boldsymbol{P}', \boldsymbol{P}$ correspond to the projection matrices to $\mathcal{I}$ and $\mathcal{I}'$. First, using Lemma A.2 and the bound on $\sigma$, we have

$$\|\boldsymbol{P}' - \boldsymbol{P}\| \le c \frac{\sigma \sqrt{\log n} B^2 KC}{\lambda_{\min}}$$

for some constant $c > 0$. Next we quantify $\Pi_{\mathcal{I}}(\boldsymbol{y})$ using the fact that (i) $\Pi_{\mathcal{I}'}(\boldsymbol{y}) = \boldsymbol{y}$ via Theorem A.1 as follows

$$\|\Pi_{\mathcal{I}}(\boldsymbol{y})\|_{\ell_2} \ge \|\Pi_{\mathcal{I}'}(\boldsymbol{y})\|_{\ell_2} - \|\Pi_{\mathcal{I}}(\boldsymbol{y}) - \Pi_{\mathcal{I}'}(\boldsymbol{y})\|_{\ell_2} \ge \sqrt{n}(1 - c \frac{\sigma \sqrt{\log n} B^2 KC}{\lambda_{\min}}). \tag{A.3}$$

In return, this implies that

$$\|\Pi_{\mathcal{N}}(\boldsymbol{y})\|_{\ell_2} \lesssim \sqrt{\frac{n\sigma \sqrt{\log n} B^2 KC}{\lambda_{\min}}}$$

To proceed, we pick $\alpha_0 = \sqrt{\frac{\lambda_{\min} n}{2KC}}$ and corresponding $\bar{\alpha} = \frac{\alpha_0}{\sqrt[4]{n}\sqrt{K}\|\boldsymbol{X}\|_B} \ge \sqrt{\frac{\lambda_{\min}}{2B^2 K^2 C}}$ and apply (??) to find that, classification error is upper bounded by

$$\text{Err}_{\mathcal{D}}(\boldsymbol{W}_T) \lesssim \sqrt{\frac{\sigma \sqrt{\log n} B^2 KC}{\lambda_{\min}}} + \frac{\Gamma B K \sqrt{C}}{\sqrt{n}\lambda_{\min}} + 12\zeta + 5\sqrt{\frac{\log(2/\delta)}{n}} + 2e^{-\Gamma}.$$

∎

# B  JOINT INPUT-OUTPUT OPTIMIZATION

In this section we wish to provide the ingredients necessary to prove a result for the case where both set of input and output weights $\boldsymbol{W}$ and $\boldsymbol{V}$ are trained. To this aim, we consider the combined neural net Jacobian associated with input and output layers given by

$$\boldsymbol{x} \mapsto f(\boldsymbol{x}; \boldsymbol{V}, \boldsymbol{W}) \coloneqq \boldsymbol{V}\phi(\boldsymbol{W}\boldsymbol{x}). \tag{B.1}$$

Denoting the Jacobian associated with (B.1) by $\mathcal{J}(\boldsymbol{V}, \boldsymbol{W})$ we have that

$$\mathcal{J}(\boldsymbol{V}, \boldsymbol{W}) = [\mathcal{J}(\boldsymbol{V})\ \mathcal{J}(\boldsymbol{W})] \in \mathbb{R}^{Kn \times k(K+d)}$$

Here, $\mathcal{J}(\boldsymbol{W})$ is as before whereas $\mathcal{J}(\boldsymbol{V})$ is the Jacobian with respect to $\boldsymbol{V}$ and is given by

$$\mathcal{J}(\boldsymbol{V}) = [\mathcal{J}(\boldsymbol{v}_1)\ \mathcal{J}(\boldsymbol{v}_2)\ \ldots\ \mathcal{J}(\boldsymbol{v}_K)]. \tag{B.2}$$

where $\mathcal{J}(\boldsymbol{v}_\ell) \in \mathbb{R}^{Kn \times k}$ is so that its $\ell$'th block rows of size $n \times k$ is nonzero for $1 \le \ell \le K$ i.e.

$$\widetilde{\ell}\text{th block row of } \mathcal{J}(\boldsymbol{v}_\ell) = \begin{cases} 0 \text{ if } \ell \ne \widetilde{\ell} \\ \phi(\boldsymbol{X}\boldsymbol{W}^T) \text{ else} \end{cases}.$$

Hence, $\mathcal{J}(\boldsymbol{V})$ is $K \times K$ block diagonal with blocks equal to $\phi(\boldsymbol{X}\boldsymbol{W}^T)$. The following theorem summarizes the properties of the joint Jacobian.

**Theorem B.1 (Properties of the Combined Input/Output Jacobian)** $\mathcal{J}(\boldsymbol{V}, \boldsymbol{W})$ *satisfies the following properties.*

- **Upper bound:** $\|\mathcal{J}(\boldsymbol{V}, \boldsymbol{W})\| \le B\|\boldsymbol{X}\|(\|\boldsymbol{W}\|_F + \sqrt{Kk}\|\boldsymbol{V}\|_{\ell_\infty})$.

- **Row-bound:** *For unit length* $\boldsymbol{u}$*:* $\|mat(\mathcal{J}^T(\boldsymbol{W})\boldsymbol{u})\|_{2,\infty} \le B\sqrt{K}\|\boldsymbol{V}\|_{\ell_\infty}\|\boldsymbol{X}\|$.

- **Entry-bound:** *For unit length* $\boldsymbol{u}$*:* $\|mat(\mathcal{J}^T(\boldsymbol{V})\boldsymbol{u})\|_{\ell_\infty} \le B\|\boldsymbol{W}\|_{2,\infty}\|\boldsymbol{X}\|$.

- **Lipschitzness:** *Given inputs* $\boldsymbol{V}, \boldsymbol{V}'$ *and outputs* $\boldsymbol{W}, \boldsymbol{W}'$

  $\|\mathcal{J}(\boldsymbol{V}, \boldsymbol{W}) - \mathcal{J}(\boldsymbol{V}', \boldsymbol{W}')\| \le B\|\boldsymbol{X}\|(\sqrt{Kk}\|\boldsymbol{V} - \boldsymbol{V}'\|_{\ell_\infty} + \sqrt{K}\|\boldsymbol{V}\|_{\ell_\infty}\|\boldsymbol{W}' - \boldsymbol{W}\|_F + \|\boldsymbol{W} - \boldsymbol{W}'\|_F)$.

**Proof** First, we prove results concerning $\mathcal{J}(\boldsymbol{V})$. First, note that

$$\|\mathcal{J}(\boldsymbol{V})\| \le \|\phi(\boldsymbol{X}\boldsymbol{W}^T)\| \le B\|\boldsymbol{X}\|\|\boldsymbol{W}\|_F.$$

Next, note that for $\boldsymbol{u} = [\boldsymbol{u}_1 \ \ldots \ \boldsymbol{u}_K] \in \mathbb{R}^{Kn}$ we have

$$
\begin{aligned}
\|\mathcal{J}^T(\boldsymbol{V})\boldsymbol{u}\|_{\ell_\infty} &= \max_{1\le\ell\le K} \|\phi(\boldsymbol{W}\boldsymbol{X}^T)\boldsymbol{u}_\ell\|_{\ell_\infty} \\
&= \max_{1\le s\le k} |\phi(\boldsymbol{w}_s\boldsymbol{X}^T)\boldsymbol{u}_\ell| \\
&= B\|\boldsymbol{W}\|_{2,\infty}\|\boldsymbol{X}\|.
\end{aligned}
$$

Let $\mathcal{J}_1, \mathcal{J}_2$ be the Jacobian matrices restricted to $\boldsymbol{V}$ and $\boldsymbol{W}$ of $\mathcal{J}(\boldsymbol{V},\boldsymbol{W})$. To prove Lipschitzness, first observe that

$$\|\mathcal{J}(\boldsymbol{V},\boldsymbol{W}) - \mathcal{J}(\boldsymbol{V}',\boldsymbol{W}')\| \le \|\mathcal{J}_1(\boldsymbol{V},\boldsymbol{W}) - \mathcal{J}_1(\boldsymbol{V}',\boldsymbol{W}')\| + \|\mathcal{J}_2(\boldsymbol{V},\boldsymbol{W}) - \mathcal{J}_2(\boldsymbol{V}',\boldsymbol{W}')\|.$$

Next, observe that

$$\|\mathcal{J}_1(\boldsymbol{V},\boldsymbol{W}) - \mathcal{J}_1(\boldsymbol{V}',\boldsymbol{W}')\| \le \|\phi(\boldsymbol{X}\boldsymbol{W}^T) - \phi(\boldsymbol{X}\boldsymbol{W}'^T)\| \le B\|\boldsymbol{X}\|\|\boldsymbol{W} - \boldsymbol{W}'\|_F.$$

We decompose $\mathcal{J}_2$ via

$$
\begin{aligned}
\|\mathcal{J}_2(\boldsymbol{V},\boldsymbol{W}) - \mathcal{J}_2(\boldsymbol{V}',\boldsymbol{W}')\| &\le \|\mathcal{J}_2(\boldsymbol{V},\boldsymbol{W}) - \mathcal{J}_2(\boldsymbol{V},\boldsymbol{W}')\| + \|\mathcal{J}_2(\boldsymbol{V},\boldsymbol{W}') - \mathcal{J}_2(\boldsymbol{V}',\boldsymbol{W}')\| \\
&\le B\sqrt{K}\|\boldsymbol{V}\|_{\ell_\infty}\|\boldsymbol{X}\|\|\boldsymbol{W}' - \boldsymbol{W}\|_F + \|\mathcal{J}_2(\boldsymbol{V},\boldsymbol{W}') - \mathcal{J}_2(\boldsymbol{V}',\boldsymbol{W}')\|.
\end{aligned}
$$

To address the second term, note that, Jacobian is linear with respect to output layer hence

$$\|\mathcal{J}_2(\boldsymbol{V},\boldsymbol{W}') - \mathcal{J}_2(\boldsymbol{V}',\boldsymbol{W}')\| = \|\mathcal{J}_2(\boldsymbol{V}-\boldsymbol{V}',\boldsymbol{W}')\| \le B\sqrt{Kk}\|\boldsymbol{V} - \boldsymbol{V}'\|_{\ell_\infty}\|\boldsymbol{X}\|.$$

Combining the latter two identities we arrive at

$$\|\mathcal{J}_2(\boldsymbol{V},\boldsymbol{W}) - \mathcal{J}_2(\boldsymbol{V}',\boldsymbol{W}')\| \le B\|\boldsymbol{X}\|(\sqrt{Kk}\|\boldsymbol{V} - \boldsymbol{V}'\|_{\ell_\infty} + \sqrt{K}\|\boldsymbol{V}\|_{\ell_\infty}\|\boldsymbol{W}' - \boldsymbol{W}\|_F),$$

completing the proof. ∎

## C  FURTHER NUMERICAL EXPERIMENTS

We depict the approximate eigenstructure in Section C.1. Complementary to Section 4, in Appendix C.2 we provide the complete set of experiments we performed on the original 10-class CIFAR-10 dataset covering various levels of label corruption. In Appendix C.3 we show numerical results on a sub-sampled 3-class version of CIFAR-10. Moreover, we demonstrate that our theory holds across different datasets by providing experimental results on MNIST in Appendix C.4.

### C.1  EXPERIMENTS ON FULL (10-CLASS) CIFAR-10

As mentioned earlier while calculating all the eigenvalues is not possible, we also verify the bimodal structure of the Jacobian using the entire spectrum by approximating its spectral density using the method recently described in Ghorbani et al. (2019a) and Papyan (2019a) for Hessian eigenvalue density estimation. We detail the parameters of the algorithm in Appendix C.5. Figure 9 depicts the estimated spectrum before and after training. We observe a similar bimodal spectrum with a few outliers. Moreover, in both depictions of the Jacobian eigenstructure we observe increasing separation between the low and high end of the spectrum after training.

### C.2  EXPERIMENTS ON FULL (10-CLASS) CIFAR-10

In addition to what has been described in Section 4 here we disclose the complete set of experiments performed on the CIFAR-10 dataset. We trained the modified ResNet20 model described in Section 4 with SGD as longs as it was necessary to achieve a good fit to the training data. Information subspace

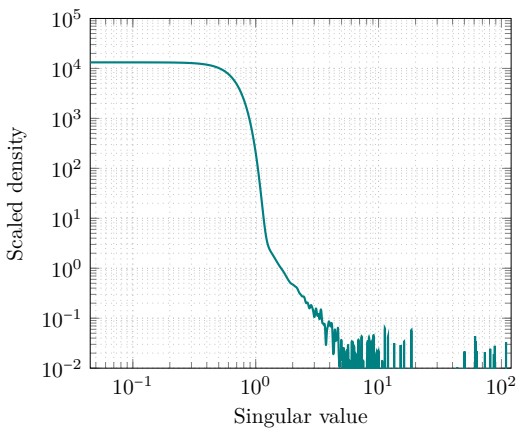

(a) Scaled spectral density of initial train Jacobian

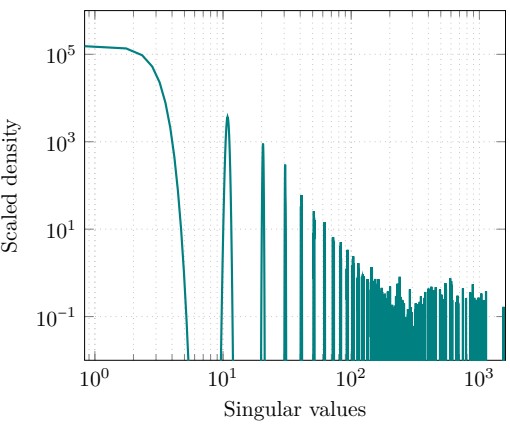

(b) Scaled spectral density of final train Jacobian

Figure 9: Scaled spectral densities of the full train Jacobian at different stages of training

| | $\frac{\|\Pi_{\mathcal{I}}(\boldsymbol{y})\|_{\ell_2}}{\|\boldsymbol{y}\|_{\ell_2}}$ | $\frac{\|\Pi_{\mathcal{N}}(\boldsymbol{y})\|_{\ell_2}}{\|\boldsymbol{y}\|_{\ell_2}}$ | $\frac{\|\boldsymbol{J}_{\mathcal{I}}^{\dagger}\boldsymbol{y}\|_{\ell_2}}{\|\boldsymbol{y}\|_{\ell_2}}$ | $\frac{\|\Pi_{\mathcal{I}}(\boldsymbol{r}_0)\|_{\ell_2}}{\|\boldsymbol{r}_0\|_{\ell_2}}$ | $\frac{\|\Pi_{\mathcal{N}}(\boldsymbol{r}_0)\|_{\ell_2}}{\|\boldsymbol{r}_0\|_{\ell_2}}$ | $\frac{\|\boldsymbol{J}_{\mathcal{I}}^{\dagger}\boldsymbol{r}_0\|_{\ell_2}}{\|\boldsymbol{r}_0\|_{\ell_2}}$ |
|---|---|---|---|---|---|---|
| $\boldsymbol{J}_{init}^{train*}$ | 0.38081 | 0.92465 | 0.027224 | 0.37114 | 0.92858 | 0.027293 |
| $\boldsymbol{J}_{final}^{train*}$ | 0.9869 | 0.16131 | 0.00070893 | 0.98669 | 0.1626 | 0.00070354 |
| $\boldsymbol{J}_{init}^{test}$ | 0.38184 | 0.92423 | 0.060229 | 0.37227 | 0.92812 | 0.06037 |
| $\boldsymbol{J}_{final}^{test}$ | 0.80926 | 0.58746 | 0.0013734 | 0.80912 | 0.58764 | 0.0013716 |

Table 3: Depiction of the alignment of the initial label/residual with the information/nuisance space using uncorrupted CIFAR-10 data.

is spanned by the top 50 singular vectors. We marked figures and table entries also included in Section 4 by asterisk.

**Experiments without label corruption.** We trained the modified ResNet20 model described in Section 4 with SGD for 400 epochs with learning rate 0.1 on the original dataset without any form of data augmentation. The network output after the last layer has been scaled by $s = 0.025$ in all experiments to follow.

**Experiments with label corruption.** We corrupt the labels in the training data by switching each label to a strictly different (incorrect) class. We train the network on the corrupted dataset for 800 epochs with initial step size 0.1 decayed to 0.01 after 760 epochs.

| | $\frac{\|\Pi_{\mathcal{I}}(\boldsymbol{y})\|_{\ell_2}}{\|\boldsymbol{y}\|_{\ell_2}}$ | $\frac{\|\Pi_{\mathcal{N}}(\boldsymbol{y})\|_{\ell_2}}{\|\boldsymbol{y}\|_{\ell_2}}$ | $\frac{\|\boldsymbol{J}_{\mathcal{I}}^{\dagger}\boldsymbol{y}\|_{\ell_2}}{\|\boldsymbol{y}\|_{\ell_2}}$ | $\frac{\|\Pi_{\mathcal{I}}(\boldsymbol{r}_0)\|_{\ell_2}}{\|\boldsymbol{r}_0\|_{\ell_2}}$ | $\frac{\|\Pi_{\mathcal{N}}(\boldsymbol{r}_0)\|_{\ell_2}}{\|\boldsymbol{r}_0\|_{\ell_2}}$ | $\frac{\|\boldsymbol{J}_{\mathcal{I}}^{\dagger}\boldsymbol{r}_0\|_{\ell_2}}{\|\boldsymbol{r}_0\|_{\ell_2}}$ |
|---|---|---|---|---|---|---|
| $\boldsymbol{J}_{init}^{train}$ | 0.35328 | 0.93552 | 0.021592 | 0.36954 | 0.92921 | 0.021568 |
| $\boldsymbol{J}_{final}^{train}$ | 0.92214 | 0.38685 | 0.00087246 | 0.92324 | 0.38423 | 0.00087304 |

Table 4: Depiction of the alignment of the initial label/residual with the information/nuisance space using 25% label corruption on CIFAR-10 data.

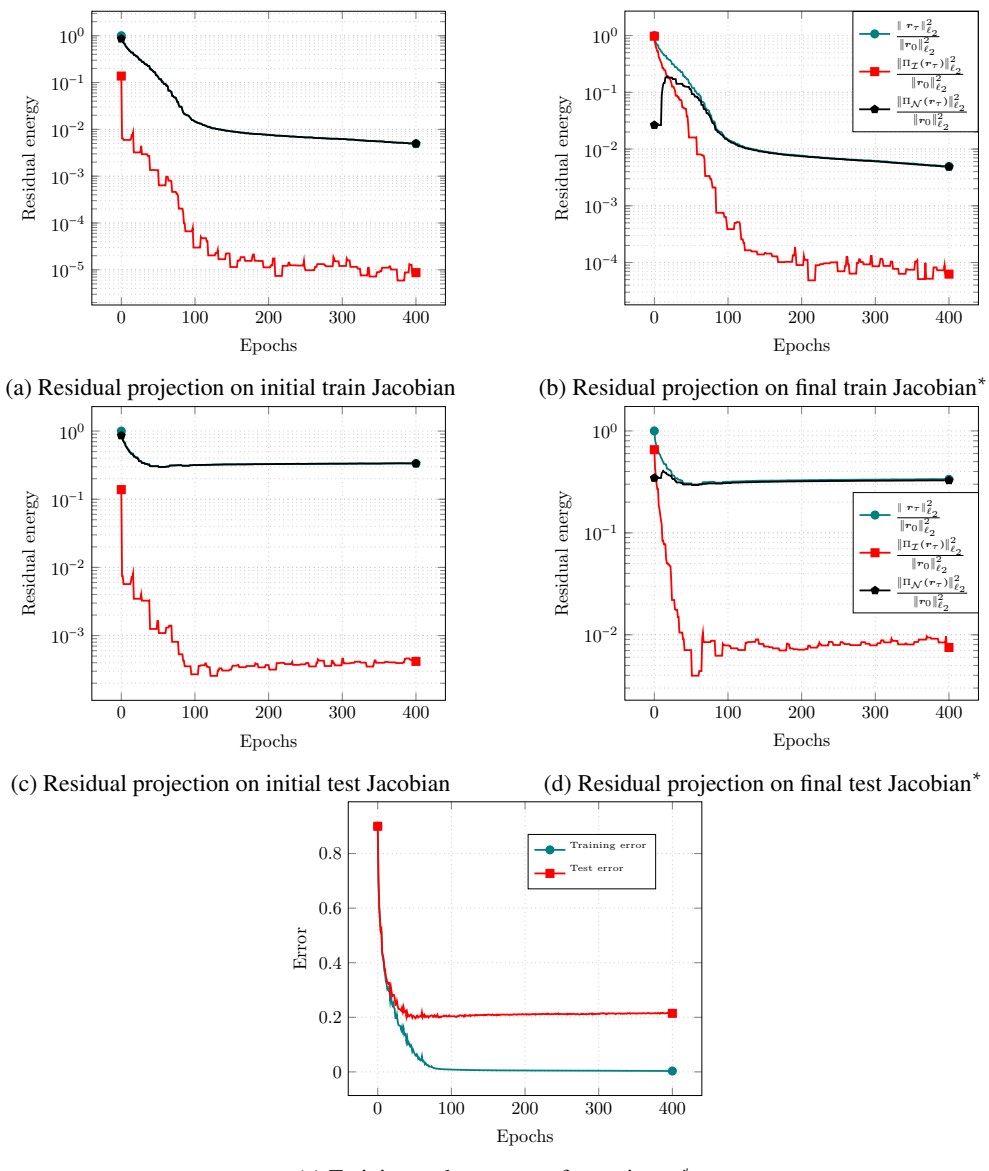

Figure 10: Experiments on the original, uncorrupted CIFAR-10 dataset. $^*$: we discuss these plots in Section 4.

| | $\dfrac{\|\Pi_{\mathcal{I}}(\boldsymbol{y})\|_{\ell_2}}{\|\boldsymbol{y}\|_{\ell_2}}$ | $\dfrac{\|\Pi_{\mathcal{N}}(\boldsymbol{y})\|_{\ell_2}}{\|\boldsymbol{y}\|_{\ell_2}}$ | $\dfrac{\|\boldsymbol{J}_{\mathcal{I}}^{\dagger}\boldsymbol{y}\|_{\ell_2}}{\|\boldsymbol{y}\|_{\ell_2}}$ | $\dfrac{\|\Pi_{\mathcal{I}}(\boldsymbol{r}_0)\|_{\ell_2}}{\|\boldsymbol{r}_0\|_{\ell_2}}$ | $\dfrac{\|\Pi_{\mathcal{N}}(\boldsymbol{r}_0)\|_{\ell_2}}{\|\boldsymbol{r}_0\|_{\ell_2}}$ | $\dfrac{\|\boldsymbol{J}_{\mathcal{I}}^{\dagger}\boldsymbol{r}_0\|_{\ell_2}}{\|\boldsymbol{r}_0\|_{\ell_2}}$ |
|---|---|---|---|---|---|---|
| $\boldsymbol{J}_{init}^{train*}$ | 0.32762 | 0.94481 | 0.017556 | 0.32152 | 0.9469 | 0.017521 |
| $\boldsymbol{J}_{final}^{train*}$ | 0.8956 | 0.44487 | 0.00096413 | 0.89597 | 0.44412 | 0.00096652 |
| $\boldsymbol{J}_{init}^{test}$ | 0.38013 | 0.92493 | 0.080777 | 0.37454 | 0.92721 | 0.080766 |
| $\boldsymbol{J}_{dip}^{test}$ | 0.7041 | 0.7101 | 0.0040147 | 0.70229 | 0.71189 | 0.0040423 |
| $\boldsymbol{J}_{final}^{test}$ | 0.44774 | 0.89416 | 0.0012216 | 0.44409 | 0.89598 | 0.0012157 |

Table 5: Depiction of the alignment of the initial label/residual with the information/nuisance space using 50% label corruption on CIFAR-10 data.

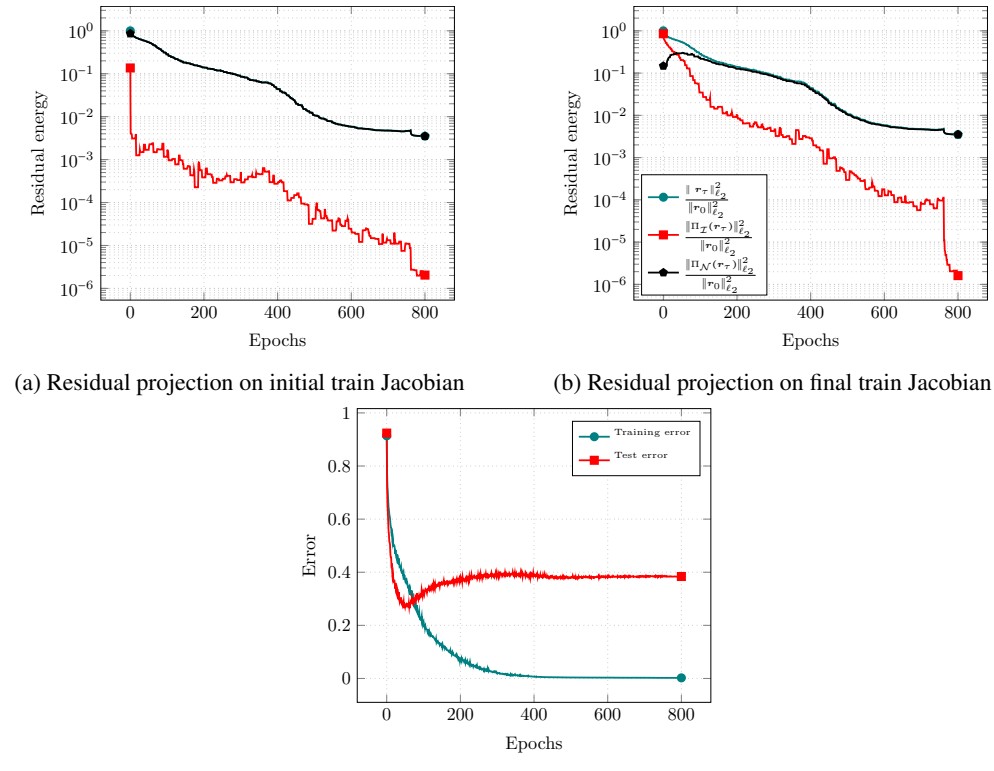

(a) Residual projection on initial train Jacobian

(b) Residual projection on final train Jacobian

(c) Training and test error of experiment

Figure 11: Experiments with 25% label corruption on CIFAR-10 data

|  | $\frac{\|\Pi_{\mathcal{I}}(\boldsymbol{y})\|_{\ell_2}}{\|\boldsymbol{y}\|_{\ell_2}}$ | $\frac{\|\Pi_{\mathcal{N}}(\boldsymbol{y})\|_{\ell_2}}{\|\boldsymbol{y}\|_{\ell_2}}$ | $\frac{\|\boldsymbol{J}_{\mathcal{I}}^{\dagger}\boldsymbol{y}\|_{\ell_2}}{\|\boldsymbol{y}\|_{\ell_2}}$ | $\frac{\|\Pi_{\mathcal{I}}(\boldsymbol{r}_0)\|_{\ell_2}}{\|\boldsymbol{r}_0\|_{\ell_2}}$ | $\frac{\|\Pi_{\mathcal{N}}(\boldsymbol{r}_0)\|_{\ell_2}}{\|\boldsymbol{r}_0\|_{\ell_2}}$ | $\frac{\|\boldsymbol{J}_{\mathcal{I}}^{\dagger}\boldsymbol{r}_0\|_{\ell_2}}{\|\boldsymbol{r}_0\|_{\ell_2}}$ |
|---|---|---|---|---|---|---|
| $\boldsymbol{J}_{init}^{train}$ | 0.31756 | 0.94824 | 0.0056031 | 0.325 | 0.94571 | 0.005592 |
| $\boldsymbol{J}_{final}^{train}$ | 0.50238 | 0.86465 | 0.00047518 | 0.50718 | 0.86184 | 0.00047967 |

Table 6: Depiction of the alignment of the initial label/residual with the information/nuisance space using 75% label corruption on CIFAR-10 data.

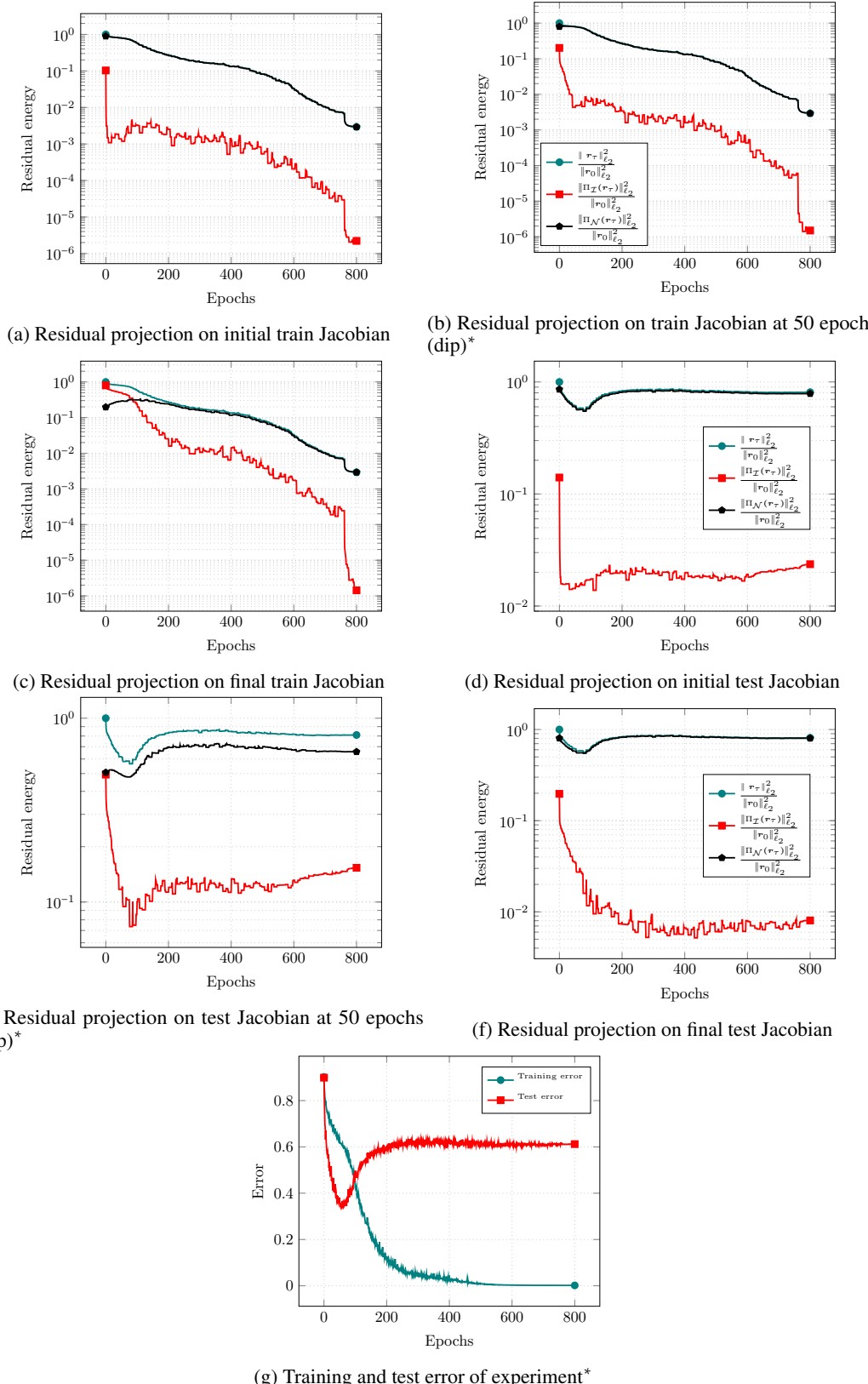

(a) Residual projection on initial train Jacobian

(b) Residual projection on train Jacobian at 50 epochs (dip)*

(c) Residual projection on final train Jacobian

(d) Residual projection on initial test Jacobian

(e) Residual projection on test Jacobian at 50 epochs (dip)*

(f) Residual projection on final test Jacobian

(g) Training and test error of experiment*

Figure 12: Experiments with 50% label corruption on CIFAR-10 data. *: we discuss these plots in Section 4.

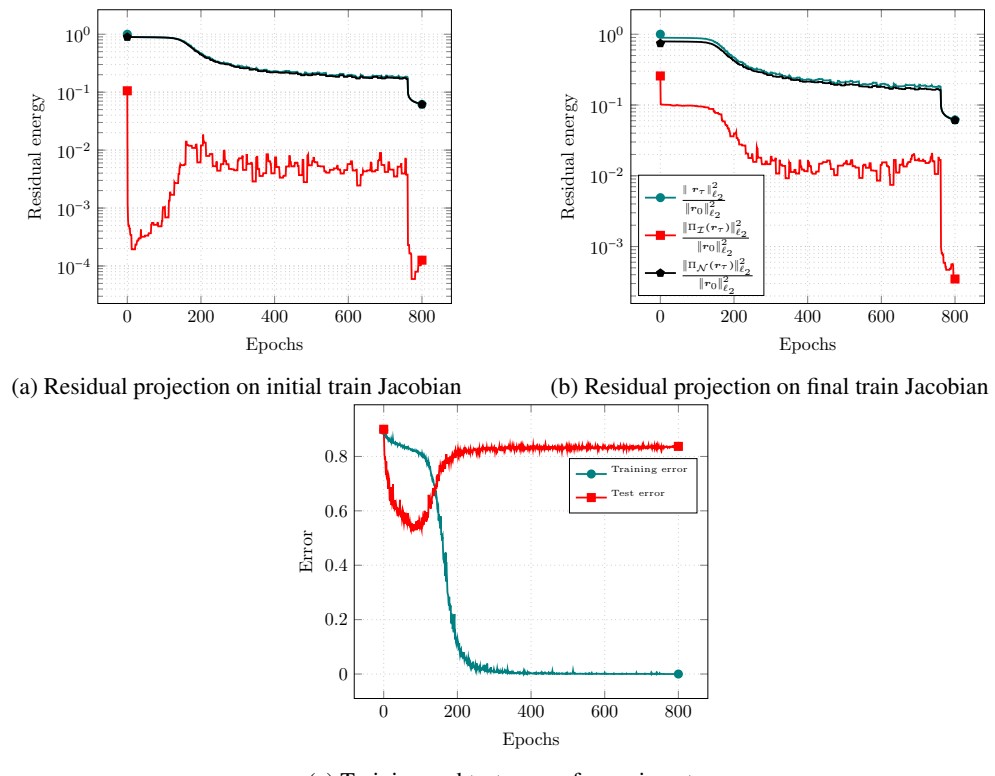

(a) Residual projection on initial train Jacobian      (b) Residual projection on final train Jacobian

(c) Training and test error of experiment

Figure 13: Experiments with 75% label corruption on CIFAR-10 data

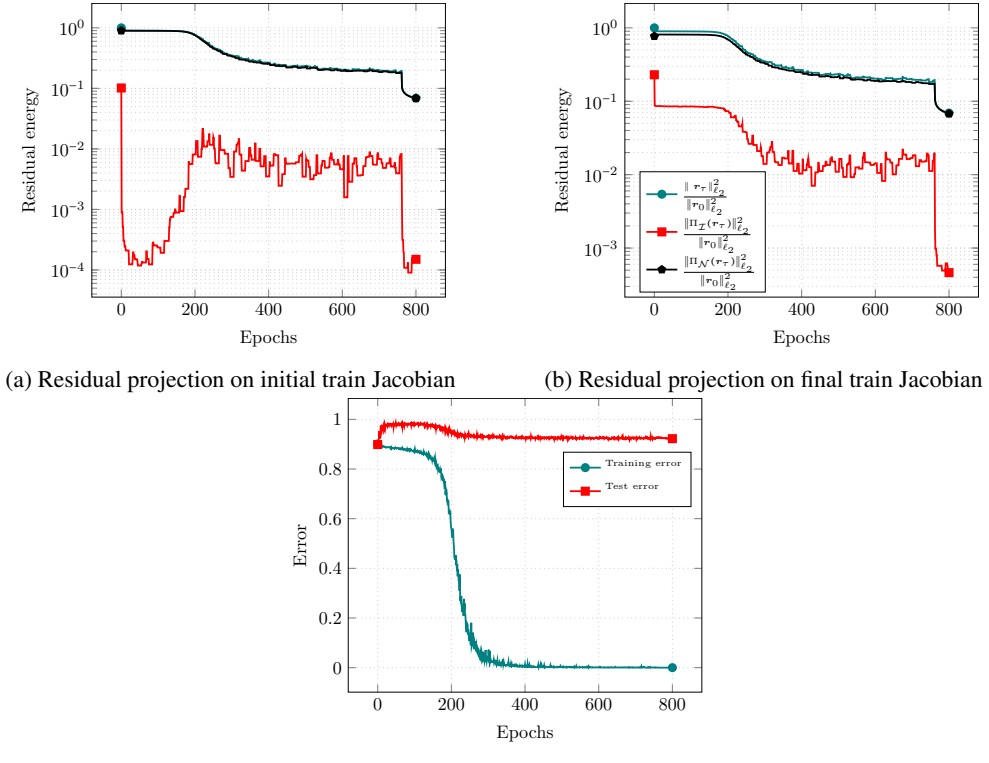

(a) Residual projection on initial train Jacobian      (b) Residual projection on final train Jacobian

(c) Training and test error of experiment

Figure 14: Experiments with 100% label corruption on CIFAR-10 data

| | $\dfrac{\|\Pi_{\mathcal{I}}(\boldsymbol{y})\|_{\ell_2}}{\|\boldsymbol{y}\|_{\ell_2}}$ | $\dfrac{\|\Pi_{\mathcal{N}}(\boldsymbol{y})\|_{\ell_2}}{\|\boldsymbol{y}\|_{\ell_2}}$ | $\dfrac{\|\boldsymbol{J}_{\mathcal{I}}^{\dagger}\boldsymbol{y}\|_{\ell_2}}{\|\boldsymbol{y}\|_{\ell_2}}$ | $\dfrac{\|\Pi_{\mathcal{I}}(\boldsymbol{r}_0)\|_{\ell_2}}{\|\boldsymbol{r}_0\|_{\ell_2}}$ | $\dfrac{\|\Pi_{\mathcal{N}}(\boldsymbol{r}_0)\|_{\ell_2}}{\|\boldsymbol{r}_0\|_{\ell_2}}$ | $\dfrac{\|\boldsymbol{J}_{\mathcal{I}}^{\dagger}\boldsymbol{r}_0\|_{\ell_2}}{\|\boldsymbol{r}_0\|_{\ell_2}}$ |
|---|---|---|---|---|---|---|
| $\boldsymbol{J}_{init}^{train}$ | 0.31581 | 0.94882 | 0.0094479 | 0.31854 | 0.94791 | 0.0094092 |
| $\boldsymbol{J}_{final}^{train}$ | 0.47747 | 0.87865 | 0.00045241 | 0.47921 | 0.8777 | 0.00045344 |

Table 7: Depiction of the alignment of the initial label/residual with the information/nuisance space using 100% label corruption on CIFAR-10 data.

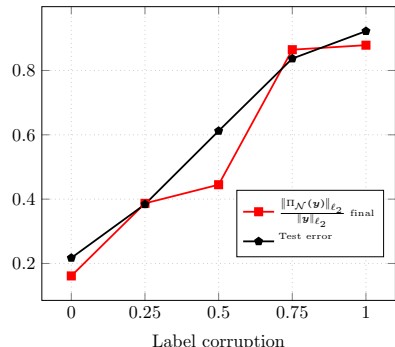

Figure 15: Test error vs. final projection of labels on nuisance subspace for CIFAR-10 experiments. See discussion of this plot in Section 4

### C.3 EXPERIMENTS ON SUBSAMPLED 3-CLASS CIFAR-10

We created a new dataset of 3 classes by sub-sampling the original CIFAR-10 dataset. To do this, we discarded all training and test data except those belonging to classes 0 (airplane), 1 (automobile) and 2 (bird) and sampled 3333 examples of each classes for a total of 9999 training images. We trained the neural network model described in Section 4 with no output scaling using SGD and Adam. We applied standard data augmentation (random crop and flip) to increase generalization. Information subspace is spanned by the top 50 singular vectors.

**Experiments without label corruption.** First we trained the network on the sub-sampled 3-class dataset keeping the original labels. For the SGD experiments, we used initial learning rate 0.01 decreased by a factor of 10 at epochs 260 and 360 for a total of 400 epochs with batch size 128. For the Adam experiment learning rate 0.01 for 400 epochs was sufficient for a good fit to the training data. We observed better Jacobian adaptation using Adam compared to SGD on this dataset (0.98743 from Table 8 vs. 0.9969 from Table 9).

**Experiments with label corruption.** We corrupt the training labels by the corruption model described in C.2. We train the network on the corrupted dataset for 800 epochs with initial step size 0.01 and batch size 128. We decrease the learning rate by a factor of 10 at the following epochs: at 500 epochs for 25% corruption, at 700 epochs for 50% and 75% corruption and at 500 and 700 for 100% corruption.

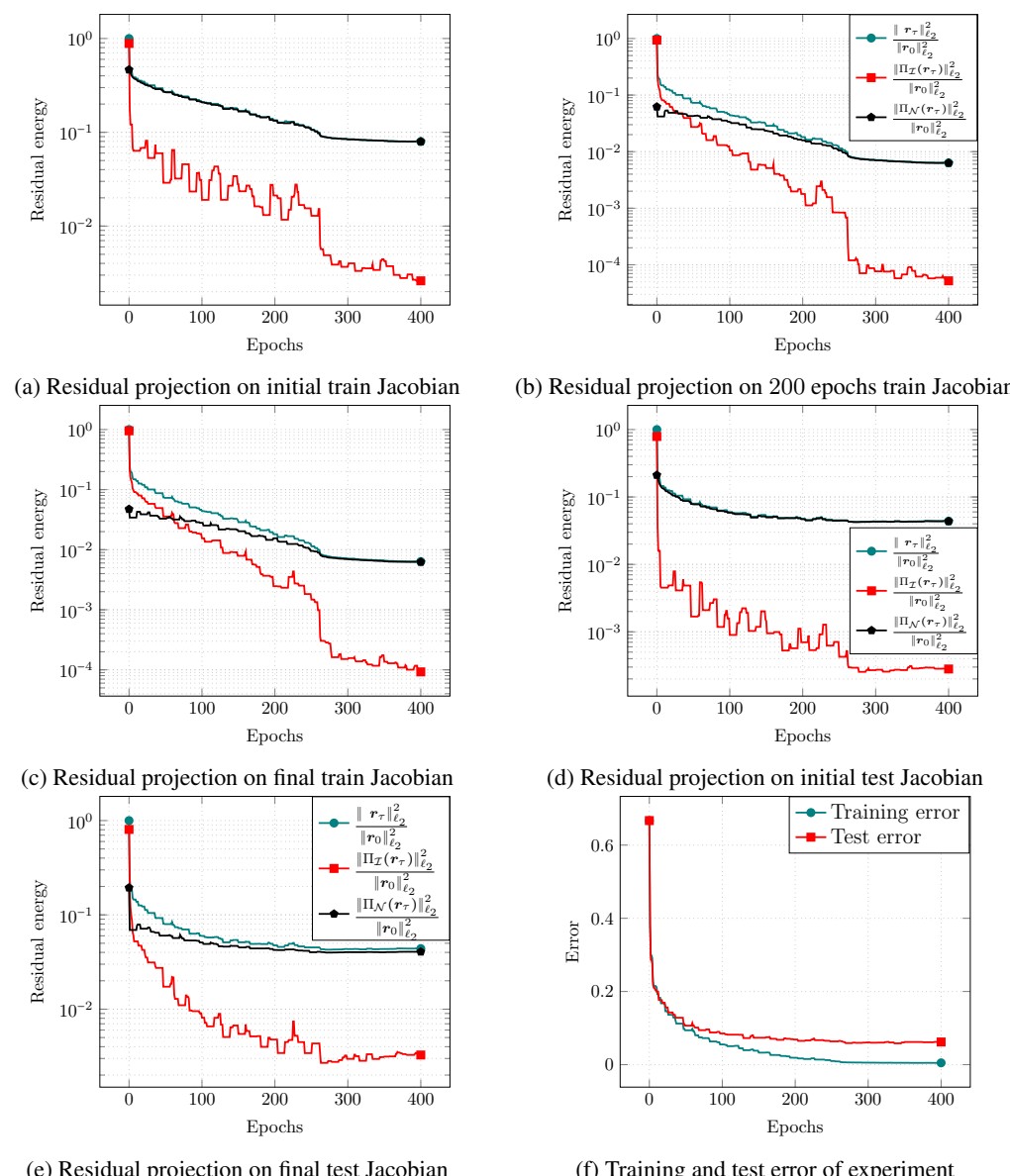

Figure 16: Experiments on 3-class uncorrupted CIFAR-10 dataset

| | $\dfrac{\|\Pi_{\mathcal{I}}(\boldsymbol{y})\|_{\ell_2}}{\|\boldsymbol{y}\|_{\ell_2}}$ | $\dfrac{\|\Pi_{\mathcal{N}}(\boldsymbol{y})\|_{\ell_2}}{\|\boldsymbol{y}\|_{\ell_2}}$ | $\dfrac{\left\|\boldsymbol{J}_{\mathcal{I}}^{\dagger}\boldsymbol{y}\right\|_{\ell_2}}{\|\boldsymbol{y}\|_{\ell_2}}$ | $\dfrac{\|\Pi_{\mathcal{I}}(\boldsymbol{r}_0)\|_{\ell_2}}{\|\boldsymbol{r}_0\|_{\ell_2}}$ | $\dfrac{\|\Pi_{\mathcal{N}}(\boldsymbol{r}_0)\|_{\ell_2}}{\|\boldsymbol{r}_0\|_{\ell_2}}$ | $\dfrac{\left\|\boldsymbol{J}_{\mathcal{I}}^{\dagger}\boldsymbol{r}_0\right\|_{\ell_2}}{\|\boldsymbol{r}_0\|_{\ell_2}}$ |
|---|---|---|---|---|---|---|
| $\boldsymbol{J}_{init}^{train}$ | 0.7239 | 0.68991 | 0.0054426 | 0.88552 | 0.4646 | 0.0040958 |
| $\boldsymbol{J}_{200epochs}^{train}$ | 0.97266 | 0.23224 | 0.0026234 | 0.96849 | 0.24905 | 0.0030069 |
| $\boldsymbol{J}_{final}^{train}$ | 0.98743 | 0.15804 | 0.0031639 | 0.97606 | 0.21749 | 0.0034312 |
| $\boldsymbol{J}_{init}^{test}$ | 0.73366 | 0.67951 | 0.010328 | 0.88827 | 0.45932 | 0.0077364 |
| $\boldsymbol{J}_{final}^{test}$ | 0.8974 | 0.44123 | 0.0027082 | 0.89772 | 0.44057 | 0.0029383 |

Table 8: Depiction of the alignment of the initial label/residual with the information/nuisance space using 3-class uncorrupted CIFAR-10 data.)

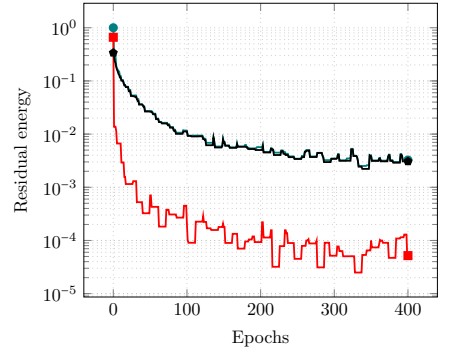
(a) Residual projection on initial train Jacobian

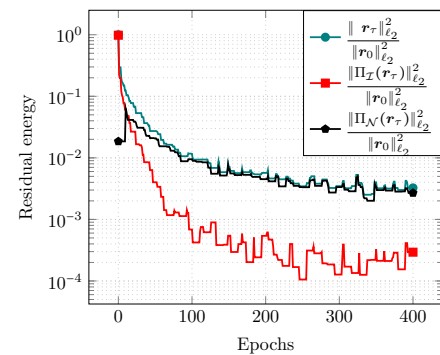
(b) Residual projection on final train Jacobian

Figure 17: Experiments on 3-class uncorrupted CIFAR-10 data, trained with Adam

|  | $\frac{\|\Pi_{\mathcal{I}}(\boldsymbol{y})\|_{\ell_2}}{\|\boldsymbol{y}\|_{\ell_2}}$ | $\frac{\|\Pi_{\mathcal{N}}(\boldsymbol{y})\|_{\ell_2}}{\|\boldsymbol{y}\|_{\ell_2}}$ | $\frac{\|\boldsymbol{J}_{\mathcal{I}}^{\dagger}\boldsymbol{y}\|_{\ell_2}}{\|\boldsymbol{y}\|_{\ell_2}}$ | $\frac{\|\Pi_{\mathcal{I}}(\boldsymbol{r}_0)\|_{\ell_2}}{\|\boldsymbol{r}_0\|_{\ell_2}}$ | $\frac{\|\Pi_{\mathcal{N}}(\boldsymbol{r}_0)\|_{\ell_2}}{\|\boldsymbol{r}_0\|_{\ell_2}}$ | $\frac{\|\boldsymbol{J}_{\mathcal{I}}^{\dagger}\boldsymbol{r}_0\|_{\ell_2}}{\|\boldsymbol{r}_0\|_{\ell_2}}$ |
|---|---|---|---|---|---|---|
| $\boldsymbol{J}_{init}^{train}$ | 0.7025 | 0.71169 | 0.0053554 | 0.8135 | 0.5815 | 0.0044353 |
| $\boldsymbol{J}_{final}^{train}$ | 0.9969 | 0.078332 | 0.0030954 | 0.9907 | 0.1361 | 0.0030632 |

Table 9: Depiction of the alignment of the initial label/residual with the information/nuisance space on 3-class uncorrupted CIFAR-10 data, trained with Adam.

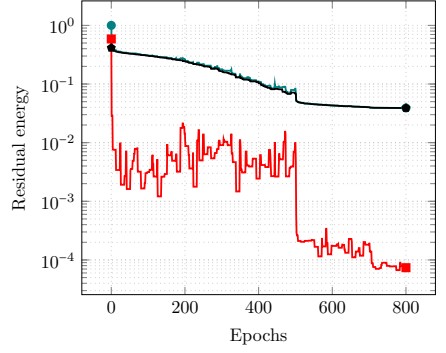
(a) Residual projection on initial train Jacobian

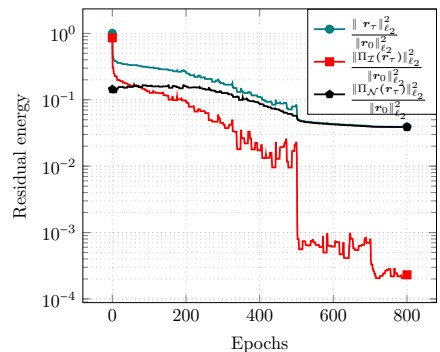
(b) Residual projection on final train Jacobian

Figure 18: Experiments with 25% label corruption on 3-class CIFAR-10 data

|  | $\frac{\|\Pi_{\mathcal{I}}(\boldsymbol{y})\|_{\ell_2}}{\|\boldsymbol{y}\|_{\ell_2}}$ | $\frac{\|\Pi_{\mathcal{N}}(\boldsymbol{y})\|_{\ell_2}}{\|\boldsymbol{y}\|_{\ell_2}}$ | $\frac{\|\boldsymbol{J}_{\mathcal{I}}^{\dagger}\boldsymbol{y}\|_{\ell_2}}{\|\boldsymbol{y}\|_{\ell_2}}$ | $\frac{\|\Pi_{\mathcal{I}}(\boldsymbol{r}_0)\|_{\ell_2}}{\|\boldsymbol{r}_0\|_{\ell_2}}$ | $\frac{\|\Pi_{\mathcal{N}}(\boldsymbol{r}_0)\|_{\ell_2}}{\|\boldsymbol{r}_0\|_{\ell_2}}$ | $\frac{\|\boldsymbol{J}_{\mathcal{I}}^{\dagger}\boldsymbol{r}_0\|_{\ell_2}}{\|\boldsymbol{r}_0\|_{\ell_2}}$ |
|---|---|---|---|---|---|---|
| $\boldsymbol{J}_{init}^{train}$ | 0.63965 | 0.76866 | 0.0047649 | 0.76602 | 0.64282 | 0.0043665 |
| $\boldsymbol{J}_{final}^{train}$ | 0.90862 | 0.41763 | 0.0023974 | 0.9254 | 0.379 | 0.0021731 |

Table 10: Depiction of the alignment of the initial label/residual with the information/nuisance space using 25% label corruption on 3-class CIFAR-10 data.

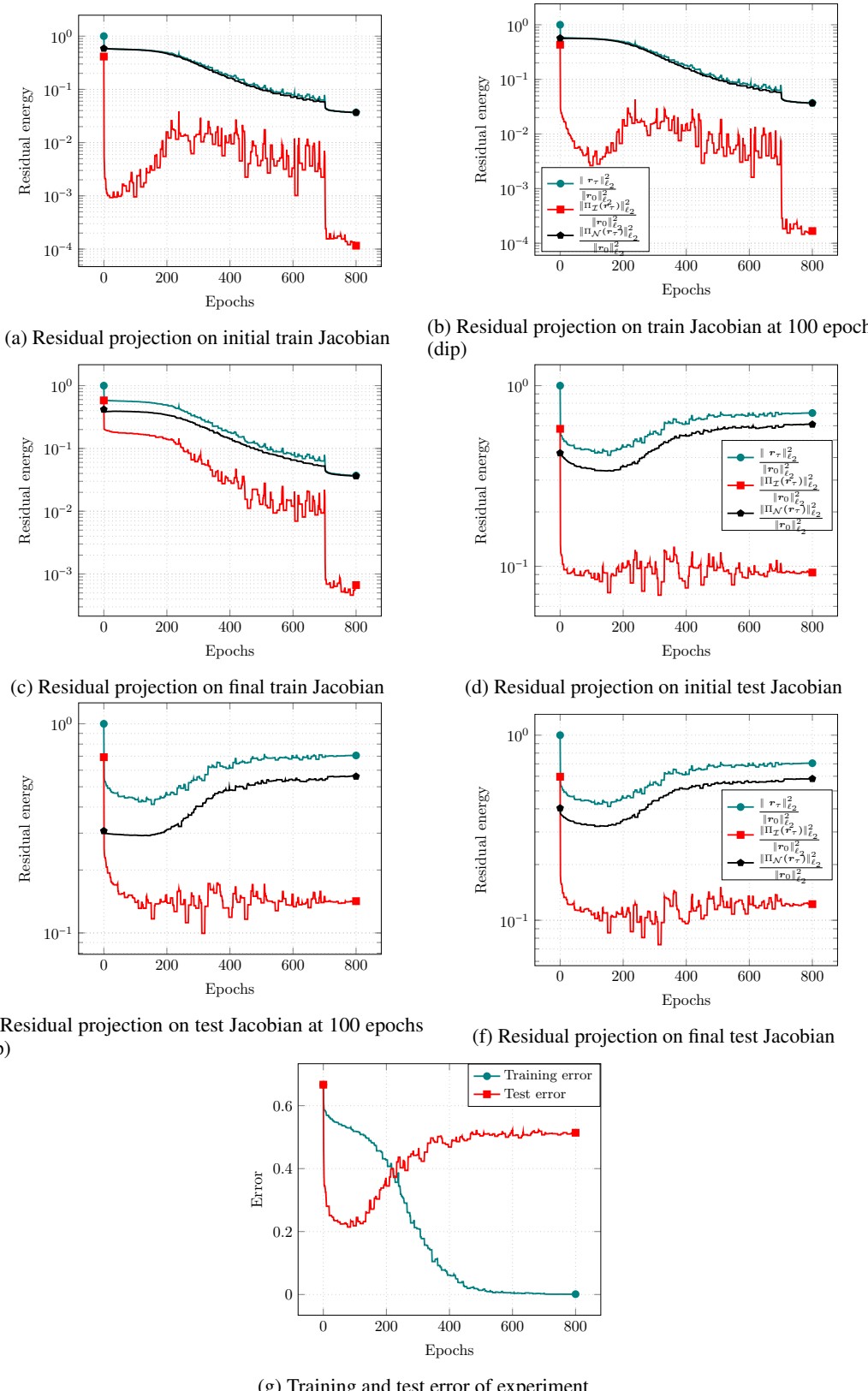

(a) Residual projection on initial train Jacobian

(b) Residual projection on train Jacobian at 100 epochs (dip)

(c) Residual projection on final train Jacobian

(d) Residual projection on initial test Jacobian

(e) Residual projection on test Jacobian at 100 epochs (dip)

(f) Residual projection on final test Jacobian

(g) Training and test error of experiment

Figure 19: Experiments with 50% label corruption on 3-class CIFAR-10 data

| | $\dfrac{\|\Pi_{\mathcal{I}}(\boldsymbol{y})\|_{\ell_2}}{\|\boldsymbol{y}\|_{\ell_2}}$ | $\dfrac{\|\Pi_{\mathcal{N}}(\boldsymbol{y})\|_{\ell_2}}{\|\boldsymbol{y}\|_{\ell_2}}$ | $\dfrac{\|\boldsymbol{J}_{\mathcal{I}}^{\dagger}\boldsymbol{y}\|_{\ell_2}}{\|\boldsymbol{y}\|_{\ell_2}}$ | $\dfrac{\|\Pi_{\mathcal{I}}(\boldsymbol{r}_0)\|_{\ell_2}}{\|\boldsymbol{r}_0\|_{\ell_2}}$ | $\dfrac{\|\Pi_{\mathcal{N}}(\boldsymbol{r}_0)\|_{\ell_2}}{\|\boldsymbol{r}_0\|_{\ell_2}}$ | $\dfrac{\|\boldsymbol{J}_{\mathcal{I}}^{\dagger}\boldsymbol{r}_0\|_{\ell_2}}{\|\boldsymbol{r}_0\|_{\ell_2}}$ |
|---|---|---|---|---|---|---|
| $\boldsymbol{J}_{init}^{train}$ | 0.58664 | 0.80985 | 0.0017197 | 0.64281 | 0.76602 | 0.0019814 |
| $\boldsymbol{J}_{dip}^{train}$ | 0.61125 | 0.79144 | 0.0026809 | 0.65671 | 0.75415 | 0.0031267 |
| $\boldsymbol{J}_{final}^{train}$ | 0.75143 | 0.65981 | 0.0018702 | 0.76311 | 0.64627 | 0.0012039 |
| $\boldsymbol{J}_{init}^{test}$ | 0.7236 | 0.69021 | 0.012765 | 0.7594 | 0.65062 | 0.01235 |
| $\boldsymbol{J}_{dip}^{test}$ | 0.81473 | 0.57984 | 0.009931 | 0.8322 | 0.55447 | 0.010515 |
| $\boldsymbol{J}_{final}^{test}$ | 0.75362 | 0.65731 | 0.0033476 | 0.77225 | 0.63532 | 0.0021479 |

Table 11: Depiction of the alignment of the initial label/residual with the information/nuisance space using 50% label corruption on 3-class CIFAR-10 data.

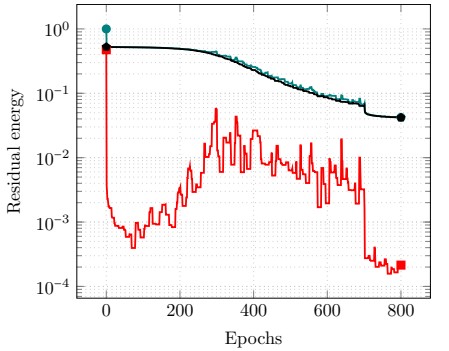

(a) Residual projection on initial train Jacobian

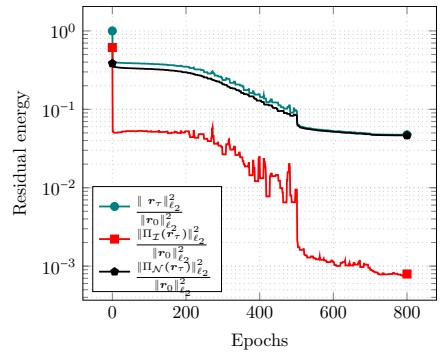

(b) Residual projection on final train Jacobian

Figure 20: Experiments with 75% label corruption on 3-class CIFAR-10 data

| | $\dfrac{\|\Pi_{\mathcal{I}}(\boldsymbol{y})\|_{\ell_2}}{\|\boldsymbol{y}\|_{\ell_2}}$ | $\dfrac{\|\Pi_{\mathcal{N}}(\boldsymbol{y})\|_{\ell_2}}{\|\boldsymbol{y}\|_{\ell_2}}$ | $\dfrac{\|\boldsymbol{J}_{\mathcal{I}}^{\dagger}\boldsymbol{y}\|_{\ell_2}}{\|\boldsymbol{y}\|_{\ell_2}}$ | $\dfrac{\|\Pi_{\mathcal{I}}(\boldsymbol{r}_0)\|_{\ell_2}}{\|\boldsymbol{r}_0\|_{\ell_2}}$ | $\dfrac{\|\Pi_{\mathcal{N}}(\boldsymbol{r}_0)\|_{\ell_2}}{\|\boldsymbol{r}_0\|_{\ell_2}}$ | $\dfrac{\|\boldsymbol{J}_{\mathcal{I}}^{\dagger}\boldsymbol{r}_0\|_{\ell_2}}{\|\boldsymbol{r}_0\|_{\ell_2}}$ |
|---|---|---|---|---|---|---|
| $\boldsymbol{J}_{init}^{train}$ | 0.58032 | 0.81439 | 0.0014306 | 0.68898 | 0.72478 | 0.0015956 |
| $\boldsymbol{J}_{final}^{train}$ | 0.64704 | 0.76246 | 0.0015808 | 0.78391 | 0.62088 | 0.00091715 |

Table 12: Depiction of the alignment of the initial label/residual with the information/nuisance space using 75% label corruption on 3-class CIFAR-10 data.

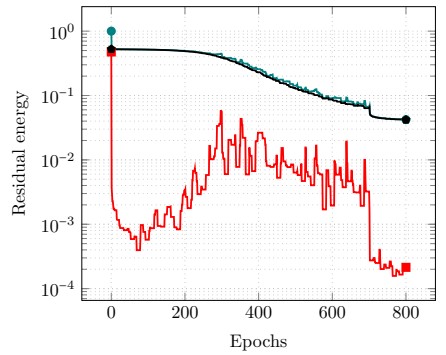

(a) Residual projection on initial train Jacobian

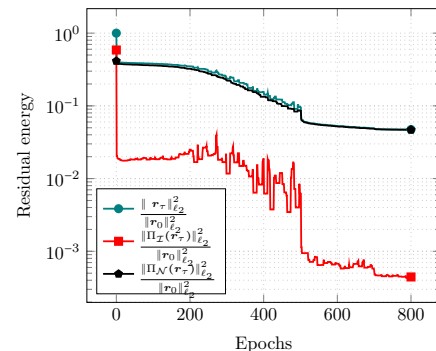

(b) Residual projection on final train Jacobian

Figure 21: Experiments with 100% label corruption on 3-class CIFAR-10 data

|  | $\frac{\|\Pi_{\mathcal{I}}(\boldsymbol{y})\|_{\ell_2}}{\|\boldsymbol{y}\|_{\ell_2}}$ | $\frac{\|\Pi_{\mathcal{N}}(\boldsymbol{y})\|_{\ell_2}}{\|\boldsymbol{y}\|_{\ell_2}}$ | $\frac{\|\boldsymbol{J}_{\mathcal{I}}^{\dagger}\boldsymbol{y}\|_{\ell_2}}{\|\boldsymbol{y}\|_{\ell_2}}$ | $\frac{\|\Pi_{\mathcal{I}}(\boldsymbol{r}_0)\|_{\ell_2}}{\|\boldsymbol{r}_0\|_{\ell_2}}$ | $\frac{\|\Pi_{\mathcal{N}}(\boldsymbol{r}_0)\|_{\ell_2}}{\|\boldsymbol{r}_0\|_{\ell_2}}$ | $\frac{\|\boldsymbol{J}_{\mathcal{I}}^{\dagger}\boldsymbol{r}_0\|_{\ell_2}}{\|\boldsymbol{r}_0\|_{\ell_2}}$ |
|---|---|---|---|---|---|---|
| $\boldsymbol{J}_{init}^{train}$ | 0.614 | 0.78931 | 0.0024274 | 0.78608 | 0.61813 | 0.0022444 |
| $\boldsymbol{J}_{final}^{train}$ | 0.60094 | 0.7993 | 0.001845 | 0.7657 | 0.6432 | 0.00098981 |

Table 13: Depiction of the alignment of the initial label/residual with the information/nuisance space using 100% label corruption on 3-class CIFAR-10 data.

## C.4 EXPERIMENTS ON MNIST

The MNIST dataset contains handwritten digits in 10 classes divided into 60k training and 10k test images. To demonstrate our theoretical findings on a dataset different from CIFAR-10 we repeat all experiments on MNIST under various levels of label corruption. We run SGD with batch size 128 on least-squares loss using the modified ResNet20 model from Section 4. In all of the following MNIST experiments the model output after the last layer has been scaled by 0.1. Information subspace is spanned by the top 50 singular vectors

**Experiments without label corruption.** We train the network on the original dataset for 100 epochs with initial step size 0.1 decayed to 0.01 after 60 epochs.

**Experiments with label corruption.** We corrupt various portions of the labels in the training data by switching the labels to a random incorrect class. We train the network on the corrupted dataset for 400 epochs with initial step size 0.1 decayed to 0.01 after 360 epochs.

|  | $\frac{\|\Pi_{\mathcal{I}}(\boldsymbol{y})\|_{\ell_2}}{\|\boldsymbol{y}\|_{\ell_2}}$ | $\frac{\|\Pi_{\mathcal{N}}(\boldsymbol{y})\|_{\ell_2}}{\|\boldsymbol{y}\|_{\ell_2}}$ | $\frac{\|\boldsymbol{J}_{\mathcal{I}}^{\dagger}\boldsymbol{y}\|_{\ell_2}}{\|\boldsymbol{y}\|_{\ell_2}}$ | $\frac{\|\Pi_{\mathcal{I}}(\boldsymbol{r}_0)\|_{\ell_2}}{\|\boldsymbol{r}_0\|_{\ell_2}}$ | $\frac{\|\Pi_{\mathcal{N}}(\boldsymbol{r}_0)\|_{\ell_2}}{\|\boldsymbol{r}_0\|_{\ell_2}}$ | $\frac{\|\boldsymbol{J}_{\mathcal{I}}^{\dagger}\boldsymbol{r}_0\|_{\ell_2}}{\|\boldsymbol{r}_0\|_{\ell_2}}$ |
|---|---|---|---|---|---|---|
| $\boldsymbol{J}_{init}^{train}$ | 0.43598 | 0.89996 | 0.028665 | 0.48168 | 0.87635 | 0.027921 |
| $\boldsymbol{J}_{final}^{train}$ | 0.9946 | 0.10375 | 0.00066394 | 0.99321 | 0.11633 | 0.00076602 |
| $\boldsymbol{J}_{init}^{test}$ | 0.43964 | 0.89817 | 0.071577 | 0.48474 | 0.87466 | 0.069724 |
| $\boldsymbol{J}_{final}^{test}$ | 0.99188 | 0.12716 | 0.0016387 | 0.99044 | 0.13796 | 0.0018745 |

Table 14: Depiction of the alignment of the initial label/residual with the information/nuisance space using uncorrupted MNIST data.

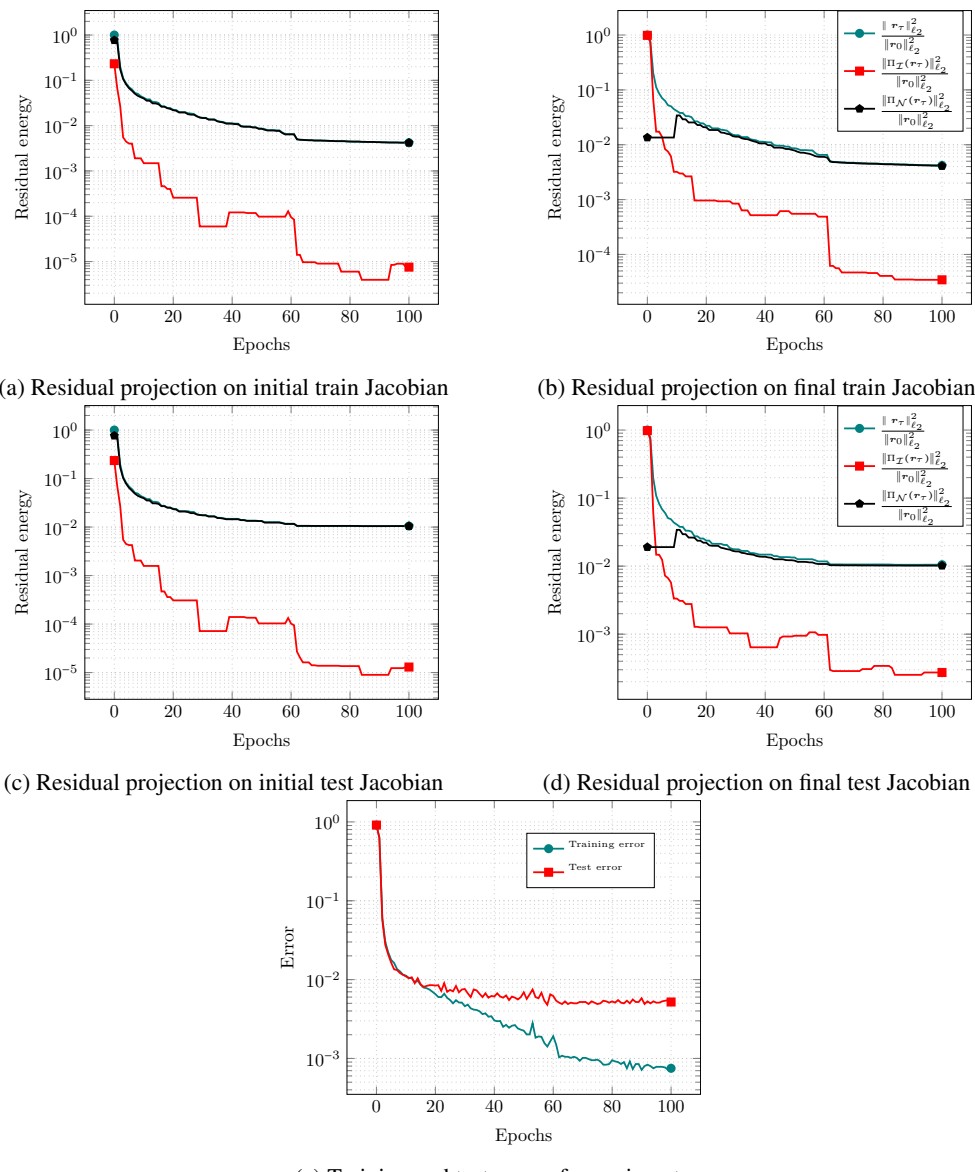

(a) Residual projection on initial train Jacobian

(b) Residual projection on final train Jacobian

(c) Residual projection on initial test Jacobian

(d) Residual projection on final test Jacobian

(e) Training and test error of experiment

Figure 22: Experiments on uncorrupted MNIST data

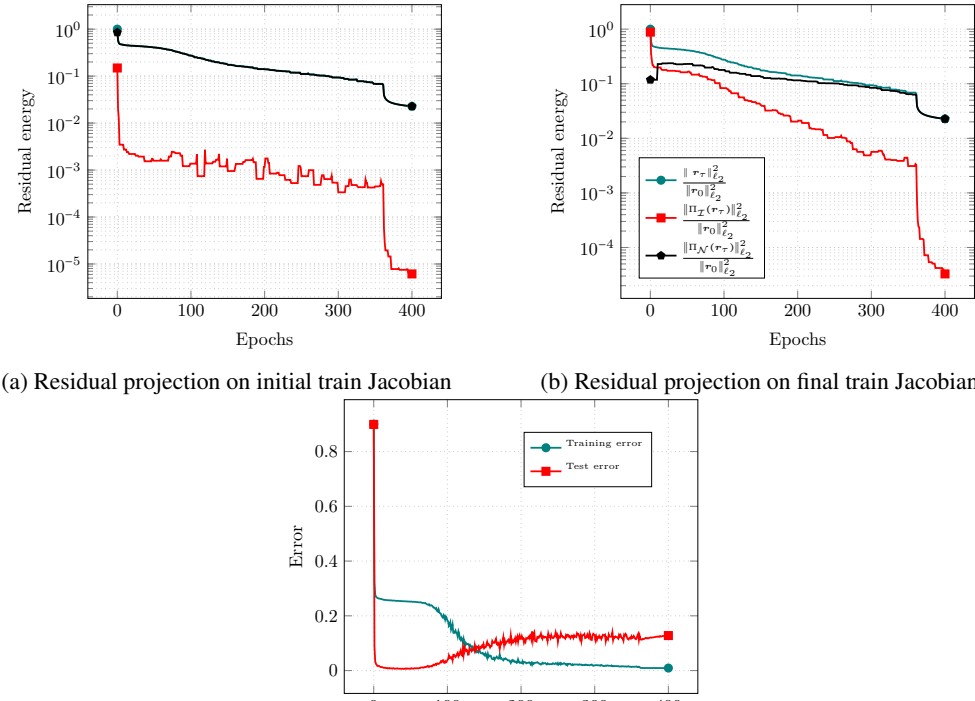

(a) Residual projection on initial train Jacobian    (b) Residual projection on final train Jacobian

(c) Training and test error of experiment

Figure 23: Experiments with 25% label corruption on MNIST data

|  | $\frac{\|\Pi_{\mathcal{I}}(\boldsymbol{y})\|_{\ell_2}}{\|\boldsymbol{y}\|_{\ell_2}}$ | $\frac{\|\Pi_{\mathcal{N}}(\boldsymbol{y})\|_{\ell_2}}{\|\boldsymbol{y}\|_{\ell_2}}$ | $\frac{\|\boldsymbol{J}_{\mathcal{I}}^{\dagger}\boldsymbol{y}\|_{\ell_2}}{\|\boldsymbol{y}\|_{\ell_2}}$ | $\frac{\|\Pi_{\mathcal{I}}(\boldsymbol{r}_0)\|_{\ell_2}}{\|\boldsymbol{r}_0\|_{\ell_2}}$ | $\frac{\|\Pi_{\mathcal{N}}(\boldsymbol{r}_0)\|_{\ell_2}}{\|\boldsymbol{r}_0\|_{\ell_2}}$ | $\frac{\|\boldsymbol{J}_{\mathcal{I}}^{\dagger}\boldsymbol{r}_0\|_{\ell_2}}{\|\boldsymbol{r}_0\|_{\ell_2}}$ |
|---|---|---|---|---|---|---|
| $\boldsymbol{J}_{init}^{train}$ | 0.38906 | 0.92121 | 0.02197 | 0.38586 | 0.92256 | 0.022028 |
| $\boldsymbol{J}_{final}^{train}$ | 0.94093 | 0.33861 | 0.0010507 | 0.9391 | 0.34365 | 0.0010331 |

Table 15: Depiction of the alignment of the initial label/residual with the information/nuisance space using 25% label corruption on MNIST data.

|  | $\frac{\|\Pi_{\mathcal{I}}(\boldsymbol{y})\|_{\ell_2}}{\|\boldsymbol{y}\|_{\ell_2}}$ | $\frac{\|\Pi_{\mathcal{N}}(\boldsymbol{y})\|_{\ell_2}}{\|\boldsymbol{y}\|_{\ell_2}}$ | $\frac{\|\boldsymbol{J}_{\mathcal{I}}^{\dagger}\boldsymbol{y}\|_{\ell_2}}{\|\boldsymbol{y}\|_{\ell_2}}$ | $\frac{\|\Pi_{\mathcal{I}}(\boldsymbol{r}_0)\|_{\ell_2}}{\|\boldsymbol{r}_0\|_{\ell_2}}$ | $\frac{\|\Pi_{\mathcal{N}}(\boldsymbol{r}_0)\|_{\ell_2}}{\|\boldsymbol{r}_0\|_{\ell_2}}$ | $\frac{\|\boldsymbol{J}_{\mathcal{I}}^{\dagger}\boldsymbol{r}_0\|_{\ell_2}}{\|\boldsymbol{r}_0\|_{\ell_2}}$ |
|---|---|---|---|---|---|---|
| $\boldsymbol{J}_{init}^{train}$ | 0.34434 | 0.93885 | 0.013931 | 0.35864 | 0.93347 | 0.013967 |
| $\boldsymbol{J}_{final}^{train}$ | 0.83136 | 0.55573 | 0.00081502 | 0.83235 | 0.55425 | 0.00081616 |
| $\boldsymbol{J}_{init}^{test}$ | 0.44458 | 0.89574 | 0.076224 | 0.45366 | 0.89117 | 0.076101 |
| $\boldsymbol{J}_{dip}^{test}$ | 0.97007 | 0.24284 | 0.0043166 | 0.9705 | 0.24109 | 0.0043041 |
| $\boldsymbol{J}_{final}^{test}$ | 0.71536 | 0.69876 | 0.0018554 | 0.71632 | 0.69777 | 0.0018564 |

Table 16: Depiction of the alignment of the initial label/residual with the information/nuisance space using 50% label corruption on MNIST data.

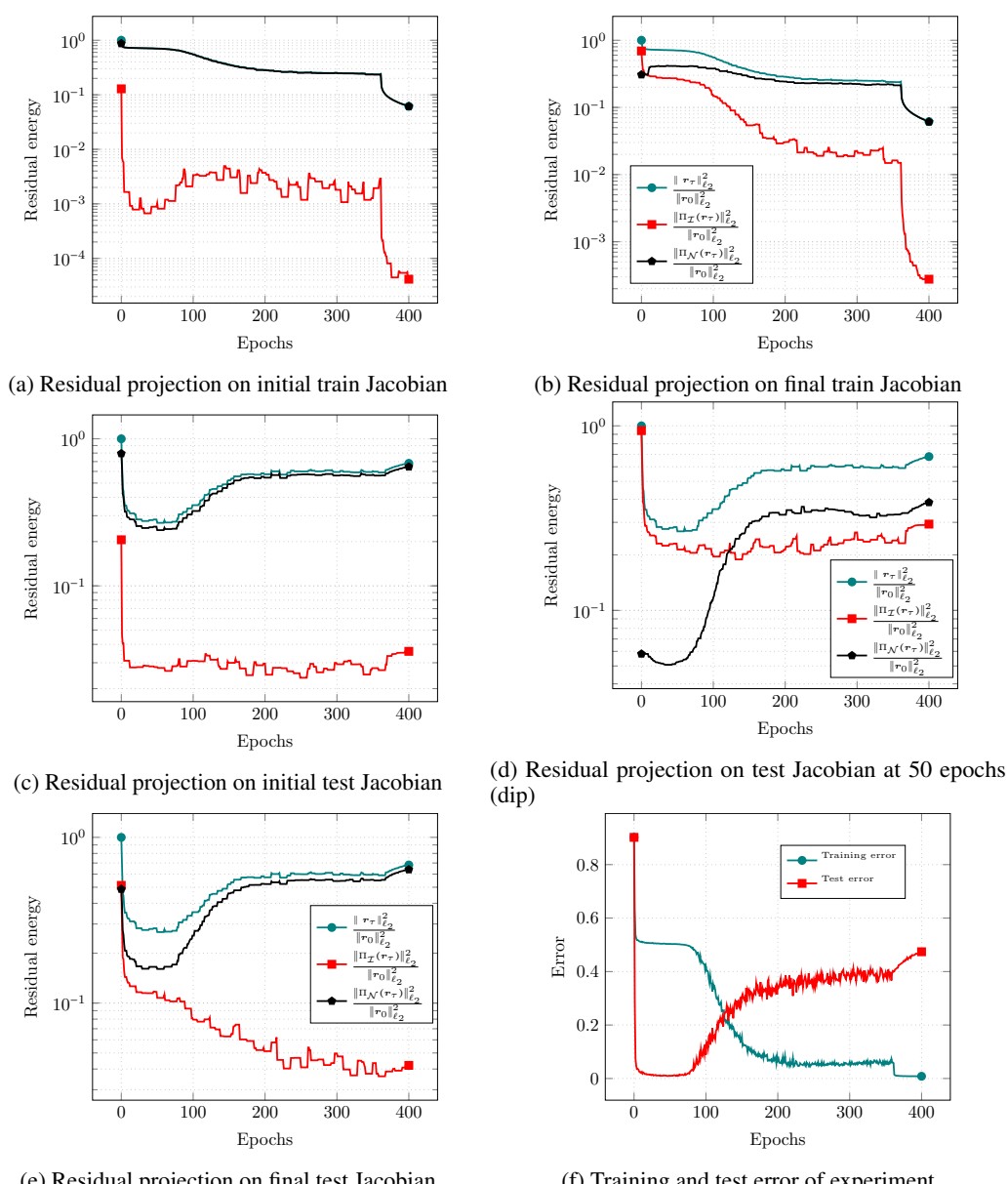

Figure 24: Experiments with 50% label corruption on MNIST data

| | $\dfrac{\|\Pi_{\mathcal{I}}(\boldsymbol{y})\|_{\ell_2}}{\|\boldsymbol{y}\|_{\ell_2}}$ | $\dfrac{\|\Pi_{\mathcal{N}}(\boldsymbol{y})\|_{\ell_2}}{\|\boldsymbol{y}\|_{\ell_2}}$ | $\dfrac{\|\boldsymbol{J}_{\mathcal{I}}^{\dagger}\boldsymbol{y}\|_{\ell_2}}{\|\boldsymbol{y}\|_{\ell_2}}$ | $\dfrac{\|\Pi_{\mathcal{I}}(\boldsymbol{r}_0)\|_{\ell_2}}{\|\boldsymbol{r}_0\|_{\ell_2}}$ | $\dfrac{\|\Pi_{\mathcal{N}}(\boldsymbol{r}_0)\|_{\ell_2}}{\|\boldsymbol{r}_0\|_{\ell_2}}$ | $\dfrac{\|\boldsymbol{J}_{\mathcal{I}}^{\dagger}\boldsymbol{r}_0\|_{\ell_2}}{\|\boldsymbol{r}_0\|_{\ell_2}}$ |
|---|---|---|---|---|---|---|
| $\boldsymbol{J}_{init}^{train}$ | 0.31995 | 0.94744 | 0.0061168 | 0.32072 | 0.94718 | 0.0061618 |
| $\boldsymbol{J}_{final}^{train}$ | 0.56007 | 0.82844 | 0.00063427 | 0.55869 | 0.82938 | 0.00062509 |

Table 17: Depiction of the alignment of the initial label/residual with the information/nuisance space using 75% label corruption on MNIST data.

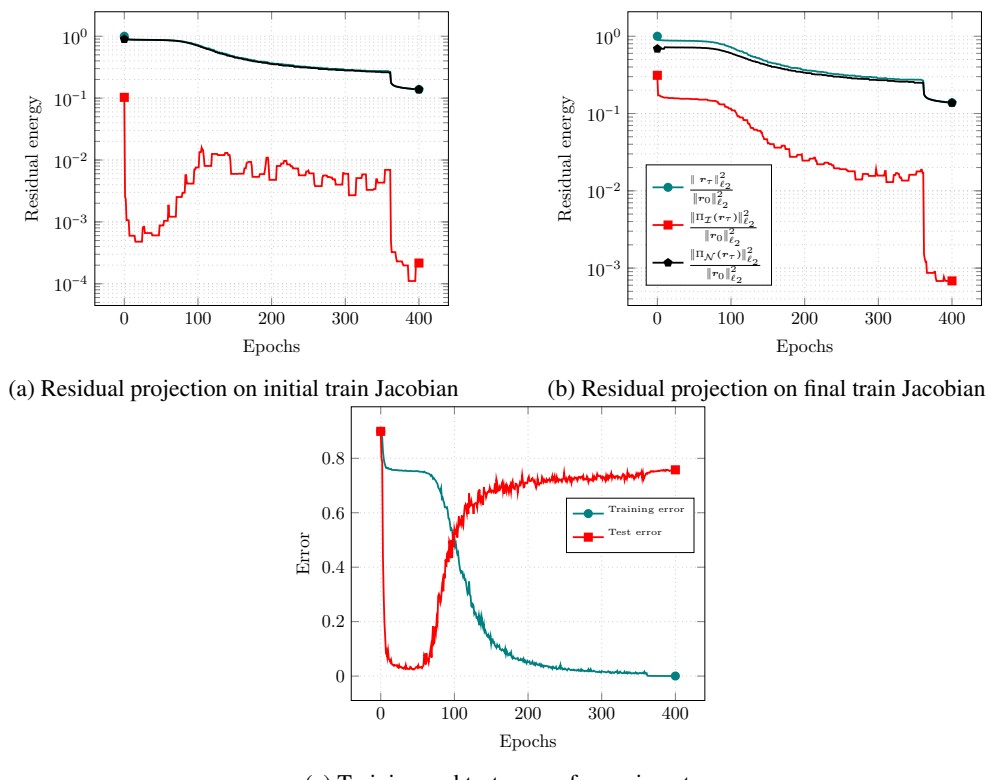

(a) Residual projection on initial train Jacobian

(b) Residual projection on final train Jacobian

(c) Training and test error of experiment

Figure 25: Experiments with 75% label corruption on MNIST data

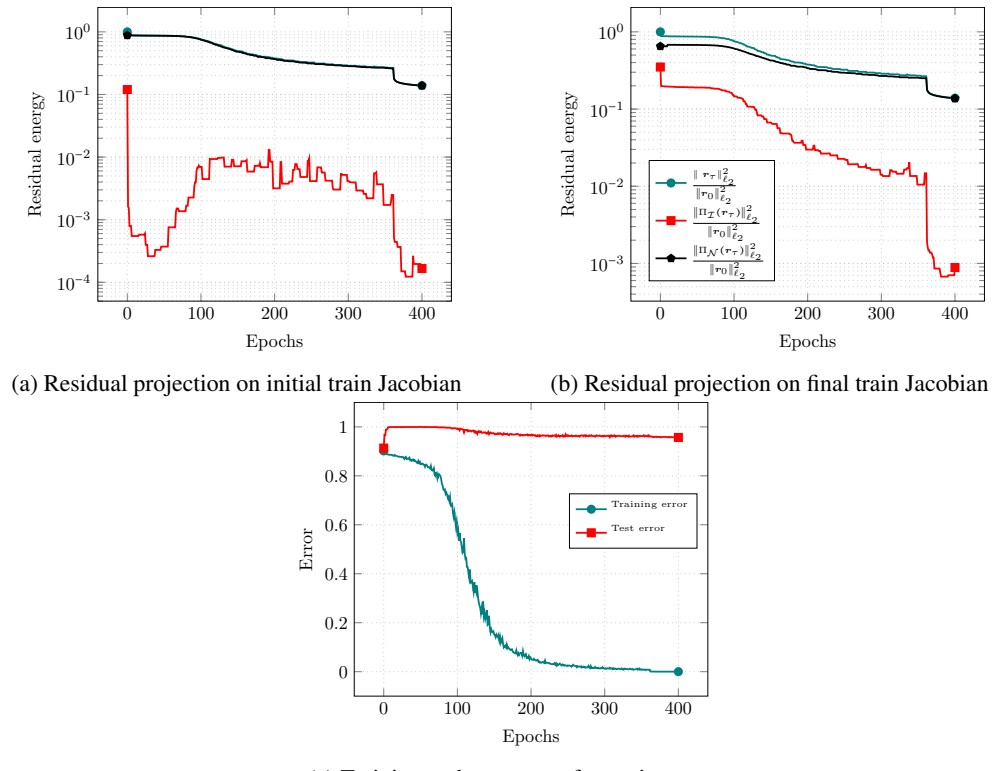

(a) Residual projection on initial train Jacobian

(b) Residual projection on final train Jacobian

(c) Training and test error of experiment

Figure 26: Experiments with 100% label corruption on MNIST data

| | $\dfrac{\|\Pi_{\mathcal{I}}(\boldsymbol{y})\|_{\ell_2}}{\|\boldsymbol{y}\|_{\ell_2}}$ | $\dfrac{\|\Pi_{\mathcal{N}}(\boldsymbol{y})\|_{\ell_2}}{\|\boldsymbol{y}\|_{\ell_2}}$ | $\dfrac{\|\boldsymbol{J}_{\mathcal{I}}^{\dagger}\boldsymbol{y}\|_{\ell_2}}{\|\boldsymbol{y}\|_{\ell_2}}$ | $\dfrac{\|\Pi_{\mathcal{I}}(\boldsymbol{r}_0)\|_{\ell_2}}{\|\boldsymbol{r}_0\|_{\ell_2}}$ | $\dfrac{\|\Pi_{\mathcal{N}}(\boldsymbol{r}_0)\|_{\ell_2}}{\|\boldsymbol{r}_0\|_{\ell_2}}$ | $\dfrac{\|\boldsymbol{J}_{\mathcal{I}}^{\dagger}\boldsymbol{r}_0\|_{\ell_2}}{\|\boldsymbol{r}_0\|_{\ell_2}}$ |
|---|---|---|---|---|---|---|
| $\boldsymbol{J}_{init}^{train}$ | 0.31793 | 0.94811 | 0.0050114 | 0.34633 | 0.93811 | 0.0050949 |
| $\boldsymbol{J}_{final}^{train}$ | 0.57769 | 0.81626 | 0.00060636 | 0.59174 | 0.80613 | 0.00062812 |

Table 18: Depiction of the alignment of the initial label/residual with the information/nuisance space using 100% label corruption on MNIST data.

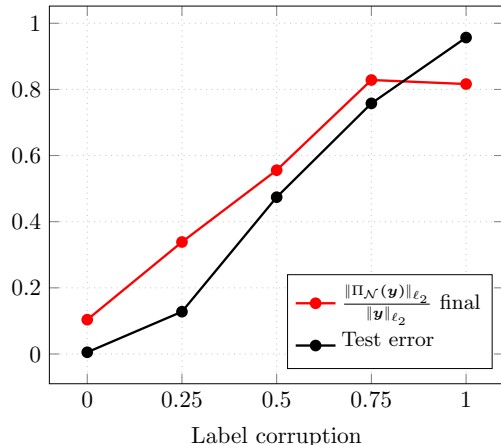

Figure 27: Test error vs. final projection of labels on nuisance subspace for MNIST experiments

## C.5    SINGULAR VALUE DECOMPOSITION OF THE JACOBIAN

The matrix representation of the Jacobian arising in the numerical experiments is in general very large and pose a computational challenge. In particular, the Jacobian matrix for a ResNet20 model is $500k \times 270k$ (or 540 GB memory with float32 entries, more than 1 TB with float64) on the CIFAR-10 training data. Storing these matrices in memory for singular value decomposition is not feasible and therefore other methods are necessary to approximate the singular values and singular vectors.

The Lanczos algorithm Lanczos (1950) is an adaptation of power methods, which guarantees fast convergence to the largest eigenvalues of a hermitian matrix. More importantly, it only requires matrix-vector products to estimate the top eigenvectors and therefore resolves the memory issues with large Jacobians. Using the ARPACK implementation of the implicitly restarted Lanczos method Lehoucq et al. (1998) it takes about 1 hour on 64 cores to find the top 50 eigenvalues and eigenvectors of the $500k \times 500k$ symmetric matrix $\boldsymbol{J}\boldsymbol{J}^T$ for CIFAR-10 training data. The same algorithm has been used to calculate the histogram of top 1000 singular values.

The required computation time for Lanczos method is dominated by large Jacobian-vector products, which increases linearly with the number of calculated eigenvalues. Therefore it is necessary to depart from deterministic power iteration type algorithms and take a different approach. Stochastic Lanczos quadrature algorithm has been recently introduced to the machine learning community in Ghorbani et al. (2019a) and Papyan (2019a) to approximate the eigenvalue density of large neural network Hessians. Finding the exact spectral density might be difficult therefore first the problem is relaxed by approximating the spectrum convolved with a Gaussian density of variance $\sigma^2 = 10^{-5}$. As described in Ghorbani et al. (2019a), we draw $k = 10$ i.i.d. Gaussian random vectors and run $m = 80$ steps of the Lanczos algorithm starting from each random vector to estimate the parameters of the spectral density. We modified the algorithm by replacing Hessian-vector products by $\boldsymbol{J}\boldsymbol{J}^T$-vector products. Finally, we normalize the density and scale it by the dimension of the Jacobian output space.

# D    Jacobian Adaptation for Linear Data

In this section, we will (non-rigorously) argue that Jacobian and its singular value spectrum indeed adapts to the data for simple linear datasets. Specifically, we will show that the very first gradient iteration from a random initialization is already powerful enough to encourage an approximately low-rank Jacobian.

**Dataset:** Let $(\boldsymbol{x}_i)_{i=1}^n \overset{\text{i.i.d.}}{\sim} \mathcal{N}(0, \boldsymbol{I}_d)$ and fix $\boldsymbol{\theta} \in \mathbb{R}^d$. Fix labels $y_i = \boldsymbol{\theta}^T \boldsymbol{x}_i$ and the dataset $(\boldsymbol{x}_i, y_i)_{i=1}^n$.

**Neural net:** Let $\boldsymbol{W}_0 \in \mathbb{R}^{k \times d} \overset{\text{i.i.d.}}{\sim} \mathcal{N}(0, 1)$ and fix $\boldsymbol{v} \in \mathbb{R}^k$ with half $+\sigma$ and half $-\sigma$ entries. Set activation to be ReLU.

**(Non-rigorous) Statement:** With small initialization ($\sigma \approx 0$) and a single large gradient step, $\boldsymbol{W}_1 = \boldsymbol{W}_0 - \eta \nabla \mathcal{L}(\boldsymbol{W}_0)$ has a (properly scaled) Jacobian matrix with **approximately rank** $2d$. Instead, at random initialization $\boldsymbol{W}_0$, Jacobian has **full rank** $n$ (assuming $kd \gg n$).

Specifically, at $\boldsymbol{W}_1$, we have

$$\sigma_{2d+1}(\mathcal{J}(\boldsymbol{X}, \boldsymbol{W}_1)) \lesssim \sqrt{d/n} \quad \text{and} \quad \sigma_{2d}(\mathcal{J}(\boldsymbol{X}, \boldsymbol{W}_1)) \gtrsim 1. \tag{D.1}$$

where $\sigma_i$ is the $i$th largest singular value. In words, $\mathcal{J}(\boldsymbol{X}, \boldsymbol{W}_1)$ is approximately rank $2d$ when $n \gtrsim d$.

This is in stark contrast to the properties of Jacobian at random initialization. For instance Li & Liang (2018); Zou et al. (2018); Oymak & Soltanolkotabi (2019) actually show that Jacobian is well conditioned with minimum singular value lower bounded by the minimum separation between the samples (implying full rank of $n$). Such separation (independent of $n$) holds for Gaussian data even when $n$ is polynomially larger than $d$ via standard concentration / packing bounds (also see Oymak & Soltanolkotabi (2019); Li & Liang (2018)). Hence, as further formalized below, we argue that even a single step of gradient descent can lead to an approximately low-rank Jacobian.

## D.1    Approach

Fixing weights of $\boldsymbol{W}_0$, we study the gradient of the random data. Let $\boldsymbol{w}_r$ be $r$th row of $\boldsymbol{W}_0$. Fixing hidden node $r$, let $\boldsymbol{x}_i' = \boldsymbol{x}_i$ if $\boldsymbol{x}_i^T \boldsymbol{w}_r \geq 0$ (i.e. the $r$th node is active) and $0$ else. The gradient of the node $r$ at $\boldsymbol{W}_0$ is given by

$$\frac{\nabla \mathcal{L}(\boldsymbol{W})}{\partial \boldsymbol{w}_r} = \frac{v_r}{n} \sum_{i=1}^n (f(\boldsymbol{x}_i, \boldsymbol{W}_0) - \boldsymbol{y}_i) \boldsymbol{x}_i'.$$

If initialization is small enough ($\sigma \approx 0$), we can approximate $f(\boldsymbol{x}_i, \boldsymbol{W}_0) - y_i \approx y_i$ hence we study

$$\frac{\nabla \mathcal{L}(\boldsymbol{W})}{\partial \boldsymbol{w}_k} \approx \frac{v_r}{n} \sum_{i=1}^n \boldsymbol{y}_i \boldsymbol{x}_i'.$$

We will show that gradients of each row are $\pm \sigma \boldsymbol{\theta}/2$ in expectation. Decompose a sample $\boldsymbol{x} \sim \mathcal{N}(0, \boldsymbol{I}_d)$ as

$$\boldsymbol{x} = g \frac{\boldsymbol{w}_r}{\|\boldsymbol{w}_r\|_{\ell_2}} + \boldsymbol{g}$$

where $\boldsymbol{g}$ is standard normal vector over the orthogonal complement of vector $\boldsymbol{w}_r$ and $g \sim \mathcal{N}(0, 1)$. Denoting label associated to $\boldsymbol{x}$ by $y$, consequently, we find

$$\mathbb{E}[y\boldsymbol{x}'] = \mathbb{E}[y\boldsymbol{x} \mid \boldsymbol{x}^T \boldsymbol{w}_r \geq 0] \mathbb{P}(\boldsymbol{x}^T \boldsymbol{w}_r \geq 0) \tag{D.2}$$

$$= \frac{1}{2} \mathbb{E}[(g \frac{\boldsymbol{w}_r}{\|\boldsymbol{w}_r\|_{\ell_2}} + \boldsymbol{g})(g\boldsymbol{w}_r + \boldsymbol{g})^T \boldsymbol{\theta} \mid g \geq 0] \tag{D.3}$$

$$= \frac{1}{2} \mathbb{E}[\boldsymbol{g}\boldsymbol{g}^T] \boldsymbol{\theta} + \mathbb{E}[g^2 \mid g \geq 0] \boldsymbol{w}_r \boldsymbol{w}_r^T \boldsymbol{\theta} \tag{D.4}$$

$$= \frac{\boldsymbol{\theta}}{2}. \tag{D.5}$$

Hence

$$\mathbb{E}[\frac{\nabla \mathcal{L}(\boldsymbol{W})}{\partial \boldsymbol{w}_k}] \approx v_i \boldsymbol{\theta}/2.$$

This implies that, in population ($n \to \infty$), gradient vector has an extremely simple rank-one form:

$$\mathbb{E}_{\boldsymbol{X}}[\nabla \mathcal{L}(\boldsymbol{W})\big|_{\boldsymbol{W}_0}] = \frac{1}{2}\boldsymbol{v}\boldsymbol{\theta}^T.$$

Standard results on non-asymptotic statistics then yield that with exponentially high-probability (i.e. $1 - \exp(-cn)$),

$$\|\frac{\nabla \mathcal{L}(\boldsymbol{W})}{\partial \boldsymbol{w}_k} - \mathbb{E}_{\boldsymbol{X}}[\frac{\nabla \mathcal{L}(\boldsymbol{W})}{\partial \boldsymbol{w}_k}]\|_{\ell_2} \lesssim \sqrt{d/n}$$

As long as $k \lesssim \mathrm{poly}(n)$, union bounding over all rows, we find the spectral norm bound

$$\|\nabla \mathcal{L}(\boldsymbol{W})\big|_{\boldsymbol{W}_0} - \mathbb{E}_{\boldsymbol{X}}[\nabla \mathcal{L}(\boldsymbol{W})\big|_{\boldsymbol{W}_0}]\| \le \sigma\sqrt{kd/n}$$

On the other hand, $\mathbb{E}_{\boldsymbol{X}}[\nabla \mathcal{L}(\boldsymbol{W})\big|_{\boldsymbol{W}_0}]$ is rank 1 with nonzero singular value $\sigma\sqrt{k}/2$.

Consequently, a large gradient step will ensure that

$$\boldsymbol{W}_1 = \boldsymbol{W}_0 - \eta\nabla\mathcal{L}(\boldsymbol{W})\big|_{\boldsymbol{W}_0}$$

is approximately rank one as well. Specifically, set the rank one matrix

$$\bar{\boldsymbol{W}}_1 = \mathbb{E}_{\boldsymbol{X}}[\eta\nabla\mathcal{L}(\boldsymbol{W})\big|_{\boldsymbol{W}_0}].$$

Then the tail of the spectrum is at most $\|\boldsymbol{W}_1 - \bar{\boldsymbol{W}}_1\| \lesssim \sqrt{d/n}\|\bar{\boldsymbol{W}}_1\|$ (i.e. $\sqrt{d/n}$ as large as the spectral norm).

Finally, we need to move from showing low-rankness of the weight matrix to low-rankness of Jacobian. Define the Jacobians associated with data at $\bar{\boldsymbol{W}}_1$ and $\boldsymbol{W}_1$ by

$$\bar{\mathcal{J}} = \mathcal{J}(\boldsymbol{X}, \bar{\boldsymbol{W}}_1) = \mathrm{diag}(\boldsymbol{v})\phi'(\bar{\boldsymbol{W}}_1\boldsymbol{X}^T) * \boldsymbol{X}^T, \quad \mathcal{J} = \mathcal{J}(\boldsymbol{X}, \boldsymbol{W}_1) = \mathrm{diag}(\boldsymbol{v})\phi'(\boldsymbol{W}_1\boldsymbol{X}^T) * \boldsymbol{X}^T$$

The Jacobian associated with $\bar{\mathcal{J}}$ is exactly rank $2d$ since matrix $\mathrm{diag}(\boldsymbol{v})\phi'(\bar{\boldsymbol{W}}_1\boldsymbol{X}^T)$ has exactly two distinct rows and kronecker producting with $\boldsymbol{X}^T$ will yield $d$ rank for each (adding up to $2d$). Since $\boldsymbol{X}$ is well-conditioned with high probability (as it is $\mathcal{N}(0,1)$ and fat with $n \gtrsim d$), top $2d$ singular values and $\sigma_{2d}(\bar{\mathcal{J}})$ are strictly positive with a lower bound independent of $n$ and $\sigma$ (after scaling by $1/\sigma\|\boldsymbol{X}\|$).

Finally, using smoothness of $\phi'$ (which excludes ReLU but includes smoother versions of ReLU such as softplus), using Jacobian perturbation bound (6.31), we have (after scaling by $1/\sigma\|\boldsymbol{X}\|$).

$$\|\mathcal{J} - \bar{\mathcal{J}}\| \lesssim \|\boldsymbol{W}_1 - \bar{\boldsymbol{W}}_1\| \lesssim \sqrt{d/n}.$$

Together, this leads to the desired conclusion (D.1) since singular value perturbation inequality yields $\sigma_{2d+1}(\mathcal{J}) \le \|\mathcal{J} - \bar{\mathcal{J}}\| \lesssim \sqrt{d/n}$ (as $\sigma_{2d+1}(\bar{\mathcal{J}}) = 0$).

