# OpenReview forum: "GENERALIZATION GUARANTEES FOR NEURAL NETS VIA HARNESSING THE LOW-RANKNESS OF JACOBIAN"
_ICLR.cc/2020/Conference — Reject_

### Official Review · AnonReviewer3 · 2019-10-23
**Official Blind Review #3**

**Rating:** 3

**Review:**

Note: The template used in this paper is of ICLR 2019, not ICLR 2020.

This paper identifies the information space and nuisance space by thresholding the singular values of the network's jacobian and shows that generally the residuals projected to the information space can be effectively optimized to zero, thus leading to efficient optimization and good generalization.

I believe this paper should be rejected because its motivation and technical framework are not novel enough in that 1) the motivation of decomposition along gradient matrix is already well-founded by a series of paper related to neural tangent kernel 2) the techniques used here also fall in a similar framework. The following is the detailed comments.

First, this paper's motivation is to employ the singular decomposition of the jacobian. Actually, the motivation is essentially the same as (Arora et al. 2019) and many other works. The neural tangent kernel matrix defined there is exactly the inner product of two jacobian (or gradient) described here and to employ the singular decomposition is actually corresponding to employing the eigendecomposition of the neural tangent kernel, which appears first in (Arora et al. 2019). The logic behind dividing the singular space into information space and nuisance space is that gradient descending along different directions has different speeds, determined by the eigenvalues.

The framework presented in this paper is based on the assumption that the parameters will not be far away from the starting point. Such an assumption further guarantees the trajectory won't be far away from the linearized trajectory, leading to an optimization guarantee. This approach is widely used by many works, and well-known for a considerably long time. Also, the paper's proof is complicated and lengthy which hinders its clarity.

To summarize, this paper definitely contains some rigorous analysis which I appreciate, but it doesn't provide new insights into optimization and generalization for deep nets. The motivation and logic behind are not novel enough, the main theorem neither. So I suggest a weak rejection to this paper in its current form.

[1] Arora, Sanjeev, et al. "Fine-grained analysis of optimization and generalization for over-parameterized two-layer neural networks." arXiv preprint arXiv:1901.08584 (2019).

****** Post-rebuttal response ******

Thanks to the authors' response. I have read the rebuttal and unfortunately, I feel it is still not strong enough to  justify this paper's novelty issue and I will keep my rating unchanged.

**Experience Assessment:**

I have read many papers in this area.

**Review Assessment: Checking Correctness Of Derivations And Theory:**

I assessed the sensibility of the derivations and theory.

**Review Assessment: Checking Correctness Of Experiments:**

I assessed the sensibility of the experiments.

**Review Assessment: Thoroughness In Paper Reading:**

I read the paper at least twice and used my best judgement in assessing the paper.

---

> ### Author Response · Authors · 2019-11-15
> **Response to Reviewer #3**
>
> Thank you for your time and efforts in reviewing our paper. First, many thanks for pointing out the template. Great catch! We fixed it.
>
> We agree that a few of the high-level motivations of the paper is related to Arora et. al. and Jacot et al. However, our paper contains many new insights, studies completely new phenomena (adaptation, harnessing low-rank, rigorous early stopping analysis etc), new experiments and theoretical justifications. We have stated the similarities and differences between our paper and this work in various places in the paper. Let us point to a few (of many) of the novelties of our work w.r.t. the Arora et. al. paper.
>
> (1)	The central premise of this paper is how one can utilize the low-rank nature of the Jacobian to provide generalization guarantees. Arora et. al. does not utilize low-rank structure at all. In particular, as the minimum eigenvalue of the NTK kernel goes to zero the required width would go to infinity. In the limit where the Jacobian is exactly low-rank the results of Arora et at are vacuous (requires $k\tendsto +inf) for instance when sigma tends to zero in the simple case study of Section 2.1 or the simplest binary classification problem where (x,y) samples have the discrete distribution (1,1) or (-1,-1). The advantage of this becomes clear during the network size analysis: Our network width is data-dependent, and it can be as small as constant (or logarithmic) if the data is low-dimensional. In contrast, results of Arora et al. don’t even apply if the kernel matrix is rank deficient. Observe that kernel matrix is expected to have bad condition number for structured data (in contrast to random data).
> (2)	The concept of information and nuisance space also does not seem to appear in Arora et al.
> (3)	Another key aspect of the results of this paper is the use of early stopping. The results of Arora et al are based on iterating to convergence and do not have early stopping which is important for the results of this paper. In fact, earlier versions of Arora et al seemed to advocated that early stopping is completely unnecessary!
> (4)	We also note that as mentioned the paper in the extreme case where alpha_0=sqrt{lambda}=sqrt{lambda_min(Sigma(X))) with with K=1 we require k>= cn^4/lambda^2 whereas Arora et al requires k>= cn^8/lambda^6. So that in this very special case our results can prove the results of Arora et al with less stringent width requirements.
> (5)	Our contributions go beyond the NTK regime as we do not require all the eigen directions of the Jacobian to be fixed across iterations rather we only need the very top ones to remain fixed.
> (6)	We provide a detailed experimental study of how neural networks learn better low-rank representations over time and provide theoretical justification for it (see Appendix D). Hence, our contributions go even beyond the NTK regime on the top eigen directions mentioned in (5).
> (7)	We also notably do not require random initialization (see Theorem 2.3). We are not familiar with any comparable generalization bound for such deterministic initialization which can be used for pretrained models.
> (8)	Finally of minor importance we study K>1.
>
>
> In summary we believe that some similarities in the high-level motivations should not overshadow the novel new insights, phenomena, experiments and theoretical results in this paper and we hope that you reconsider your score.

---

### Official Review · AnonReviewer2 · 2019-10-23
**Official Blind Review #1**

**Rating:** 3

**Review:**

The authors use the empirically supported assumption of the low-rank nature of the neural network Jacobian to provide new elements of data-dependent optimization and generalization theory. By modelling the data as a low-rank object itself, they analytically study the evolution of the train and test errors. The paper divides the space of weights and biases into the “information” and “nuisance” subspaces, spanned by the top largest and the remaining singular vectors of the Jacobian respectively. They use this division and its alignment with the low-rank structure of the data to talk about convergence speed. Finally, they provide numerical experiments to back their claims.

I enjoyed the paper, however, there were many points where I was unclear on the precise nature of the assumptions used / the strength of the results.

Disclaimer: I didn’t manage to read through the proofs in the appendix and cannot therefore vouch for its correctness.

-- Point 1 --
Leveraging the data structure

I am unclear on how exactly you were modelling the structure of the data. From your proofs, it seems that you have been dealing with the matrix X comprising the concatenated flatted vectors of the raw input features (e.g. pixels) of the input data [x1,x2,...,x_datasetsize]^T. In particular, the only place where I see data explicitly enter is in Definition 2.1, where you look at the X X^T and fi’(w X) fi’(w X)^T.

If the data is linearly separable in the raw input space on its own, then I see that the matrix X X^T will be low-rank (related to the number of classes). I also see your point about the connection of the y to the relevant (semantic) clusters. The same argument could by applied to fi’(w X) fi’(w X)^T -- provided that the features produced are again linearly separable, we will observe this object to have a low (number of classes - 1) rank.

What is unclear to me is whether these assumptions are warranted. I understand that some simple datasets, e.g. MNIST, are essentially linearly separable in the raw pixels, and therefore the X X^T indeed is low rank. However, I doubt anything like that is true for big datasets, such as ImageNet. For deep networks that are used on these big datasets, such a modelling assumptions would likely not be true. I wonder how this relates to your results, since the low-rank nature of the Hessian is observed even for those networks, which is in turn related to the low rank nature of the Jacobian.

-- Point 2 --
Data implicitly present in the Jacobian tensor.

I wonder how you modelled the Jacobian tensor that you started using on page 3. Since the Jacobian -- the derivative of the output logits with respect to the weights, has to be evaluated at a particular input X, the assumptions you make on the data are in turn having an effect on the Jacobian, and vice versa.

I am unclear on exactly what assumptions you make about the object, and whether you are actually saying that its low rank structure comes from the data, is empirical observed and therefore assumed, or due to the network regardless of the data.

I recently saw a new arXiv submission that seems to be looking into this on real networks: https://arxiv.org/pdf/1910.05929.pdf Their model explicitly assumes that logit gradients cluster in the weight space in a particular way.

-- Point 3 --
Square loss vs softmax

You are using the square loss |f(X) - y|^2 throughout your work. Many of the empirical low-rank observations of the Hessian (related to the JJ^T) are performed on real networks with the cross-entropy loss. While the Hessian with the square loss is of the form JJ^T + terms, the softmax in the cross-entropy loss introduces an additional cross-term (let us call it P for now), which in turn makes it JP(PJ)^T + terms. Do you know how this relates to your results?

More generally, does the square loss you use make the results significantly different from what we would get for a softmax?

-- Point 4 --
Neural Tangent Kernel (NTK) -- assumptions

Under the NTK assumption, you still need to model the derivatives of each logit with respect to each weight on each input in order to obtain the Jacobian matrix and in turn JJ^T. I am therefore very confused by “Based on our simulations the M-NTK indeed haslow-rank structure with a few large eigenvalues and many smaller ones. “ on page 5. What assumptions exactly do you use in your model?

-- Point 5 --
Neural Tangent Kernel (NTK) -- validity

By assuming the NTK holding, do you limit the validity of your results? I think it is believed that NTK might not, generally speaking, be enough to capture the complexity of DNNs, and therefore assuming it might limit the range of applicability of any results derived assuming it.

-- Conclusion --
In general, my points of confusion often stem from being unsure as to what parts of the argument were assumed and based on what empirical / theoretical evidence, and what parts were generically true. While the paper seems interesting, I am not sure what its novel contribution is and how broad the claims made actually are in their applicability.

Appendix: I was not able to judge the proofs in the appendix.


**Experience Assessment:**

I have read many papers in this area.

**Review Assessment: Checking Correctness Of Derivations And Theory:**

I assessed the sensibility of the derivations and theory.

**Review Assessment: Checking Correctness Of Experiments:**

I assessed the sensibility of the experiments.

**Review Assessment: Thoroughness In Paper Reading:**

I made a quick assessment of this paper.

---

> ### Author Response · Authors · 2019-11-15
> **Response to Reviewer #1**
>
> Thank you for your time and efforts in reviewing our paper.
>
> Re Point 1: Our theory yields data-dependent bounds and does not require any assumptions on the data or the Jacobian. In short: If dataset has nice properties (quantified by theory e.g. low-rankness), bounds become stronger and theory works for any cutoff value of alpha as soon as the network is sufficiently wide. However, our result shows that when the Jacobian exhibits such low-rank structure one can use a large alpha and in turn small width networks to achieve good generalization (proportional to how aligned the labels are with top eigen-vectors of the Jacobian). As case study, we rigorously proved that the Jacobian indeed becomes low-rank when the input data obeys a mixture of Gaussian model. However, we note that for low-rank behavior to happen it is not necessary for the input data to be linearly separable or low rank. Indeed, we empirically demonstrated this low-rank/bimodal structure on CIFAR-10, a dataset that is clearly not linearly separable (unlike MNIST). The low-rankedness of the Jacobian originates from the representation power of the architecture for a given data set. It seems that when the network architecture can learn good data representations, the Jacobian becomes low rank. Stated differently, if the data features at the last hidden layer are approximately separable then we expect the Jacobian to be approximately low rank. It is difficult to see how a network can generalize without such low-rank or clusterable representations. Hence, we expect low-rank behavior to be prevalent and expect these observations to hold on larger datasets such as CIFAR 100 and ImageNet for typical networks that achieve good generalization performance. In fact, preliminary simulations on a subset of this data sets confirm this.
>
> Re Point 2: The low-rank structure of the Jacobian is strongly suggested by theory as demonstrated in Sec 2.3 when the data comes from a Gaussian Mixtures. Specifically, Thm 2.5 shows that if the noise level is small, the effective rank of the M-NTK is approximately K^2 C. To verify the effective low-rank property of the Jacobian on real networks and practical datasets (not generated by GMM) we empirically show that the Jacobian has bimodal structure on CIFAR-10 and MNIST. That being said, our generalization results hold for any spectrum cut-off and we don’t need any assumptions on the data distribution. Please see response to point 1 for further discussion about this. Thanks for the reference (https://arxiv.org/pdf/1910.05929.pdf). This is in line with our intuition and gives further credence to the observation that the Jacobian is low rank.
>
> Re Point 3: Great question! Cross-entropy term P will emphasize the points that are close to the decision boundary (small margin) and shrink the other ones (large margin). Hence, during training JP will be low-rank where low-rankness is not only due to J but also related to the number of points in the class boundaries. We used square loss to keep the theoretical analysis simpler, however we did make similar observations empirically on other loss functions.  In particular, we indeed found that for both squared loss with softmax and cross-entropy loss the Jacobian has also low-rank structure and observed Jacobian adaptation on CIFAR10 with cross-entropy. Roughly stated for cross-entropy theory one has to replace the residual with the derivative of the loss and we expect that the general ideas presented in this paper will hold.
>
> Re Point 4: We are not completely sure if we understand your question on modeling the derivatives w.r.t each logit. As you pointed out, in the multi-class NTK case we have to differentiate each output w.r.t. each input feature and concatenate them to obtain J. In M-NTK, we still have a single Jacobian whose dimensions grow with number of classes and this is exactly what we assumed in theory and calculated in the numerical experiments.
>
> Re Point 5: The theoretical analysis on convergence and generalization in Jacot, 2018 hold for the infinite-width limit NTK, which we agree is restrictive in practice. Our theory does not make this assumption, and provides generalization results for finite width networks, where the width k can be even constant in some cases (in particular when the Jacobian is sufficiently low-rank). In particular, our results go beyond the NTK regime as we do not require all the eigenvectors of the Jacobian to remain approximately fixed. Rather we show that it is sufficient if only the few top ones remain approximately fixed. This is exactly why we can handle rather small widths not possible in the NTK analysis. Please see the case study in Section 2.3 which clearly demonstrates the advantages of our result with respect to the NTK regime. Furthermore, in Appendix D we go even further than this and begin to demystify the more mysterious adaptation behavior observed in our experiments which is clearly outside the purview of NTK style analysis.

---

### Official Review · AnonReviewer4 · 2019-11-03
**Official Blind Review #4**

**Rating:** 3

**Review:**

This paper proposes new (data dependent) generalization guarantees based on the Jacobian of the model. The authors suggest that if the desired outputs lie into the information space (the subspace spanned by the largest eigenvectors of the NTK), the model will train faster and better generalization will be achieved.

The faster convergence of the model in the information space is not surprising and was observed by Jacot 2018. The authors make improvements over this result:
 - They present a generalization result, whereas Jacot 2018 focuses only on the convergence on the training set. It is also formulated as a classification problem instead of a regression one.
- It doesn’t need JJ^t to stay constant during the training.

However, the setting considered by the authors to derive their theoretical contributions is too restrictive. The model exposed in section 1 is extremely simplified, as only W can be learned and V is fixed. As a result, the model is in essence completely linear: the goal is, for a given V, to learn a “good” hidden layer using a linear model and the loss L : h -> ||V phi(h) - y ||.

The experiment on cifar10 is interesting, especially the section regarding label corruption. A more extensive empirical investigation is this direction would be of great value. The results uncovered are not surprising and predicted by Jacot 2018 (granted, it is interesting to see that the result holds for finite width and non-continuous gradient flow).

I think this paper in its current state is not good enough for two reasons. First, the major contribution is a generalization bound that is derived for a model that is too simple (even simpler than a standard one hidden layer network). Beside this result, the rest of the paper is  incremental, as the link between convergence rate and the projection of the desired outputs on the information space was already made in Jacot 2018

NB: I did not check the derivation of the results in annexes.

Nitpick:

Page 2: “our results may shed light on the generalization capabilities of networks initialized with pre-trained models commonly used in meta/transfer learning” seems like a bit of a stretch. While I agree that theories that requires random initialization won’t work for transfer learning, the results presented by the authors don’t really leverage anything particular about pre-training.


**Experience Assessment:**

I have read many papers in this area.

**Review Assessment: Checking Correctness Of Derivations And Theory:**

I assessed the sensibility of the derivations and theory.

**Review Assessment: Checking Correctness Of Experiments:**

I assessed the sensibility of the experiments.

**Review Assessment: Thoroughness In Paper Reading:**

I read the paper at least twice and used my best judgement in assessing the paper.

---

> ### Author Response · Authors · 2019-11-15
> **Response to Reviewer #4**
>
> Thank you for your effort and time for reviewing our paper.
>
> Re “The faster convergence of the model in the information space is not surprising and was observed by Jacot 2018” and “The results uncovered are not surprising and predicted by Jacot 2018 (granted, it is interesting to see that the result holds for finite width and non-continuous gradient flow).”
>
> First let us agree that we should have discussed Jacot 2018 in further detail. We were not aware that Jacot et al. contained similar insights to Arora et al. (which we compared to). We revised the paper and now properly refer to Jacot et al (see Prior Art section). Compared to Jacot et al., paper contains many new insights (adaptation, harnessing low-rank, early stopping generalization analysis etc), provides new experiments and theoretical justifications. Below we outline some of these novelties
>
> (1)	The central premise of this paper (as clear by the title) is how one can utilize the low-rank nature of the Jacobian to provide generalization guarantees. Jacot et al. does not utilize low-rank structure at all. Jacot et al. does mention principal components and lower eigenvectors in one paragraph (see their page 7). This is obviously related but much closer to Arora et al.’s analysis (which is also followup on Jacot et al.) than ours. In contrast this paper quantifies the low-rankness of features, demonstrates real datasets exhibit Jacobian low-rankness,  and states many provable benefits (generalization, small network width, convergence…) via low-rankness.
> (2)	The concept of info and nuisance space (which helps quantify low-rankness) is also new to this paper and does not seem to appear in Jacot et al.
> (3)	The result of Jacot et al. are for gradient flow on infinite width networks. Our results hold for gradient descent with finite width neural networks. In fact, in some cases we can handle constant width thanks to carefully quantifying the benefit of low-rankness. Any simple discrete variant of Jacot et al. would require the width to scale polynomially in the size of the training data (e.g. the Arora et al. paper). It will also get much worse for badly conditioned NTK see Section 2.3 on mixture models.
> (4)	Our contributions go beyond the NTK regime as we do not require all the eigendirections of the Jacobian to be fixed across iterations rather we only need the very top ones to remain fixed.
> (5)	We provide a detailed experimental study of how neural nets learn better low-rank representations over time and provide theoretical justification for it (see Appendix D). Hence, our contributions go even beyond the NTK regime on the top eigen directions mentioned in (5).
> (6)	As also noted by the reviewer we focus on generalization and classification unlike Jacot et. al. 2018.
>
>
> Re “the setting of theoretical contributions is too restrictive. The model exposed in section 1 is extremely simplified, as only W can be learned, and V is fixed. As a result, the model is in essence completely linear...”
>
> We respectfully disagree. First re “the model is linear”. If W was fixed, then yes, the model would be linear but as a function of W the model is definitely not linear. We note that we have only focused on learning the first layer of the network for clarity of exposition. That said most of the arguments in the paper have been written in a way which extension to training of both layers is possible (including the Radamecher complexity arguments). Also, in Appendix B we already sketched how one would go about handling joint optimization of both layers.
>
> Re Nitpick “our results may shed light on the generalization with pre-trained models” seems stretch. While I agree that theories that require random initialization won’t work for transfer learning, the results of the authors don’t leverage anything about pre-training.
>
> Our contribution re pretraining is that we do not need random initialization unlike related works. Any initial Jacobian can be your NTK (applies to finite networks under smooth activations). Hence guarantees hold from arbitrary initialization. Numerical section also demonstrates network learns better low-rank representation over time. When you put these together, it is reasonable to highlight possible benefits on transfer learning. That said, we will tone down the statement.
>
> Re Model is too simple (even simpler than a standard one hidden layer network).
>
> Please see response earlier above which demonstrates that the proof does extend to joint optimization of both layers.
>
> Re The paper is incremental, as the link between convergence rate and the projection of the desired outputs on the information space was already made in Jacot 2018
>
> Please see response to question above clarifying the novelties with respect to Jacot et al.
>
>
> In conclusion we believe that some similarities in the high-level motivations should not overshadow the novel new insights, phenomena, experiments and theoretical results in this paper and we hope that you reconsider your score.

---

### Decision · Program_Chairs · 2019-12-19

**Decision:**

Reject

**Comment:**

This submission investigates the properties of the Jacobian matrix in deep learning setup. Specifically, it splits the spectrum of the matrix into information (large singulars) and ``nuisance (small singulars) spaces. The paper shows that over the information space learning is fast and achieves zero loss. It also shows that generalization relates to how well labels are aligned with the information space.

While the submission certainly has encouraging analysis/results, reviewers find these contributions limited and it is not clear how some of the claims in the paper can be extended to more general settings. For example, while the authors claim that low-rank structure is suggested by theory, the support of this claim is limited to a case study on mixture of Gaussians. In addition, the provided analysis only studies two-layer networks. As elaborated by R4, extending these arguments to more than two layers does not seem straighforward using the tools used in the submission. While all reviewers appreciated author's response, they were not convinced and maintained their original ratings.